# OBCache: Optimal Brain KV Cache Pruning for Efficient Long-Context LLM Inference

**Yuzhe Gu** [1]   **Xiyu Liang** [2]   **Jiaojiao Zhao** [3]   **Enmao Diao** [4]

## Abstract

Large language models (LLMs) with extended context windows enable powerful applications but impose significant memory overhead, as caching all key-value (KV) states scales linearly with sequence length and batch size. Existing cache eviction methods address this by exploiting attention sparsity, yet they typically rank tokens heuristically using accumulated attention weights without considering their true impact on attention outputs. We propose Optimal Brain Cache (OBCache), a principled framework that formulates cache eviction as a layer-wise structured pruning problem. Building upon the Optimal Brain Damage (OBD) theory, OBCache quantifies token saliency by measuring the perturbation in attention outputs induced by pruning tokens, with closed-form scores derived for isolated keys, isolated values, and joint key-value pairs. Our scores account not only for attention weights but also for information from value states and attention outputs, thereby enhancing existing eviction strategies with output-aware signals. Experiments on LLaMA and Qwen models demonstrate that replacing the heuristic scores in existing works, which estimate token saliency across different query positions, with OBCache's output-aware scores consistently improves long-context accuracy. Code is available at https://github.com/DreamSoul-AI/OBCache.

## 1. Introduction

Large language models (LLMs) (Touvron et al., 2023a;b; Bai et al., 2023; OpenAI, 2023) have recently revolution-ized a wide range of natural language processing tasks, including document summarization (Zhang et al., 2024a), question answering (Kamalloo et al., 2023), code generation (Roziere et al., 2023), and dialogue systems (Taori et al., 2023; Chiang et al., 2023). Despite their impressive capabilities, many of these applications require processing long sequences, which poses substantial challenges for efficient deployment. A major bottleneck in LLM inference stems from their autoregressive nature, which necessitates caching all key-value (KV) states across the context window. Because the cache size scales linearly with both sequence length and batch size, it leads to substantial memory and latency overheads. For instance, running a LLaMA-3.1-8B model (Grattafiori et al., 2024) with a 1M-token context window requires storing over 120GB of KV cache, which exceeds the memory capacity of most GPUs.

A promising line of research addresses this challenge via *KV cache eviction*, a training-free technique that reduces inference costs by selectively discarding unimportant KV tokens (Zhang et al., 2023; Xiao et al., 2024). These methods are motivated by the observation that only a small subset of tokens significantly influences model predictions. By evicting redundant tokens during inference, such approaches can substantially reduce memory and computational costs while incurring moderate performance degradation. Early methods such as $H_2O$ (Zhang et al., 2023) evict tokens based on accumulated attention weights, a simple yet effective heuristic for estimating token saliency. More recent techniques, including TOVA (Oren et al., 2024) and SnapKV (Li et al., 2024), refine this strategy by introducing more sophisticated attention-based scoring mechanisms to better preserve accuracy. However, these methods rely primarily on attention weights and often overlook the contribution of value states in shaping the final model outputs. Heuristically accumulating attention scores fails to fully capture the true impact of token removal on attention outputs, which directly affects the hidden states and downstream predictions. Consequently, these methods may mistakenly retain tokens with negligible influence or discard ones whose importance is not evident from attention weights alone.

To tackle these limitations, we propose Optimal Brain Cache (OBCACHE), a principled framework that formulates KV

[1]University of Pennsylvania, Philadelphia, USA [2]University of Electronic Science and Technology of China, Chengdu, China [3]Duke Kunshan University, Kunshan, China [4]DreamSoul, China. Correspondence to: Yuzhe Gu <tracygu@seas.upenn.edu>, Enmao Diao <enmao.diao@dreamsoul.com>.

*Proceedings of the $43^{rd}$ International Conference on Machine Learning*, Seoul, South Korea. PMLR 306, 2026. Copyright 2026 by the author(s).

cache eviction as a layer-wise structured pruning problem. The key insight behind OBCACHE is that the actual impact of removing KV pairs, namely, their influence on future model outputs, can be effectively approximated by analyzing local perturbations in historical attention outputs when the corresponding KV vectors are pruned. This approximation enables us to estimate the contribution of each token to the model output without requiring access to future states at inference time. These perturbation estimates therefore serve as token-wise saliency scores to inform output-aware cache eviction and retention decisions.

In parallel, some recent works have also explored token saliency through output perturbation analysis and observed that value-state information, such as value norms, can provide useful signals for improved cache eviction. However, these approaches either lack a formal theoretical framework (Guo et al., 2024) or base their analysis on objective bounding and relaxation techniques that require additional assumptions on attention distributions (Feng et al., 2025). In contrast, our analysis is grounded in the Optimal Brain Damage (OBD) theory (LeCun et al., 1989), originally proposed for pruning static model weights. By treating cached KV states as pruning variables, we define three types of pruning units: isolated value vectors, isolated key vectors, and joint key-value pairs at the same token position. In each case, we derive closed-form expressions for pruning-induced output perturbations via second-order Taylor approximation.

Under this OBD-based formulation, we show that prior value-aware eviction scores (Guo et al., 2024; Feng et al., 2025) correspond to the isolated-value case within our framework. Moreover, our framework offers flexibility in the choice of pruning objectives and pruning units. This flexibility allows it to extend beyond value-only criteria by capturing the contributions of key states and key-value interactions, and to naturally support both static eviction in the prefill stage and dynamic eviction in the decoding stage.

Furthermore, we show that existing attention-based scoring methods, which accumulate attention weights across different query positions, emerge as special cases under our framework. Specifically, the pruning objective is simplified to preserving the attention matrix within different perturbation windows, and the pruning units are reduced to individual attention columns. As such, OBCACHE generalizes and complements existing attention-based eviction methods, and can be seamlessly integrated into any score-based cache eviction pipeline to improve token selection.

In summary, our key contributions are as follows:

- We introduce OBCACHE, a principled scoring framework for KV cache eviction that estimates token saliency via eviction-induced perturbations in attention outputs, which can be seamlessly integrated into

existing eviction pipelines to improve token selection.

- We provide the first theoretical formulation of KV cache eviction as a layer-wise structured pruning problem based on the Optimal Brain Damage theory, deriving closed-form saliency scores for isolated values, isolated keys, and joint key-value pairs, and showing that prior attention-based heuristic scores arise as special cases of our more general formulation.

- We conduct extensive experiments showing that replacing heuristic attention scores in existing eviction methods with OBCACHE's output-aware scores consistently improves long-context performance on LLaMA-3.1 and Qwen-2.5 models across long retrieval, language modeling, and LongBench benchmarks.

## 2. Related Work

### 2.1. KV Cache Compression

Motivated by the observation that only a sparse subset of key-value (KV) tokens contributes to model predictions, *cache eviction* methods directly reduce the memory footprint by eliminating redundant KV states. For example, StreamingLLM (Xiao et al., 2024) discovers the *attention sink* phenomenon and retains only the initial and most recent tokens to enable infinite-context decoding. $H_2O$ (Zhang et al., 2023) proposes accumulating attention weights across all query positions to dynamically identify salient tokens. TOVA (Oren et al., 2024) simplifies this approach by considering only the attention distribution of the most recent query. A more refined strategy, SnapKV (Li et al., 2024), aggregates attention scores within a small observation window and applies a pooling-based clustering mechanism, which is effective in retrieval-centric tasks. To refine the attention-only saliency in earlier methods, VATP (Guo et al., 2024) and CriticalKV (Feng et al., 2025) show that scaling accumulated attention weights by the norm of the corresponding value states leads to improved long-context performance.

Orthogonal to eviction-based methods, *sparse attention* approaches (Tang et al., 2024; Sun et al., 2025) reduce inference cost by dynamically selecting a subset of KV pairs during attention computation while retaining the full KV cache. *Cache merging* methods (Zhang et al., 2024b; Wan et al., 2025) seek to mitigate the information loss caused by irreversible eviction by merging evicted KV states into the retained cache. Other works exploit distinctive sparsity patterns across different layers (Qin et al., 2025) or attention heads (Fu et al., 2025) to design adaptive KV cache allocation strategies. For example, PyramidKV (Cai et al., 2025) observes higher redundancy in deeper layers and statically assigns larger cache budgets to them, while AdaKV (Feng et al., 2026) dynamically allocates head-wise budgets based on global attention statistics across all heads.

## 2.2. Model Pruning

Another line of research for reducing LLM inference costs focuses on pruning model parameters. Classical pruning frameworks (LeCun et al., 1989; Hassibi & Stork, 1992) quantify the saliency of each pruning unit by estimating the perturbation it induces in a task-specific loss, often approximated to second order by a Taylor series. However, computing second-order statistics globally is still inefficient for LLM-scale models (Ma et al., 2023). To reduce complexity, recent approaches adopt local formulations that operate at the level of individual layers (Frantar & Alistarh, 2023; Sun et al., 2024), where the objective becomes minimizing the change in layer outputs. By varying the pruning units, such methods can structurally remove redundant components such as attention layers, heads, or hidden channels. In this work, we extend the layer-wise pruning paradigm from static weight pruning to the eviction of the dynamic KV cache. By treating KV states as pruning units, we introduce a theoretically grounded framework to more accurately quantify token saliency, thereby connecting KV cache compression with the model-pruning literature.

## 3. Notation and Preliminaries

We review the KV caching mechanism in transformer-based LLMs, focusing on the prefill, decoding, and cache eviction phases. Bold capital letters denote matrices, and bold lower-case letters with subscripts denote row vectors. For clarity, we omit the batch, head, and layer indices.

**Prefill.** Let $X \in \mathbb{R}^{l \times d}$ be the prompt embeddings, and let $\mathbf{Q}, \mathbf{K}, \mathbf{V} \in \mathbb{R}^{l \times d}$ be the projected query, key, and value matrices, where $l$ is the prompt length and $d$ is the hidden size. The attention output $\mathbf{O} \in \mathbb{R}^{l \times d}$ is computed as follows:

$$\mathbf{O} = \mathbf{A}\mathbf{V}, \quad \mathbf{A} = \sigma(\mathbf{Z}), \quad \mathbf{Z} = \frac{\mathbf{Q}\mathbf{K}^\top}{\sqrt{d}} \in \mathbb{R}^{l \times l},$$

where $\sigma(\cdot)$ is the row-wise softmax function, $\mathbf{Z}$ denotes the pre-softmax attention logits, and $\mathbf{A} \in \mathbb{R}^{l \times l}$ is the attention weight matrix. In this phase, $\mathbf{K}$ and $\mathbf{V}$ are cached to avoid recomputation in subsequent decoding steps.

**Decoding.** At decoding step $t$, let $\boldsymbol{x}_t \in \mathbb{R}^d$ be the newly generated token embedding, and let $\mathbf{q}_t, \mathbf{k}_t, \mathbf{v}_t \in \mathbb{R}^d$ be its projected query, key, and value vectors. After appending $\mathbf{k}_t, \mathbf{v}_t$ to the cache, the key and value matrices extend to shape $s \times d$, where $s = l + t$. The step-$t$ attention output is

$$\mathbf{o}_s = \mathbf{a}_s \mathbf{V}, \quad \mathbf{a}_s = \sigma(\mathbf{z}_s), \quad \mathbf{z}_s = \frac{\mathbf{q}_t \mathbf{K}^\top}{\sqrt{d}} \in \mathbb{R}^s.$$

For the remainder of this paper, we use $\mathbf{Q}, \mathbf{K}, \mathbf{V}, \mathbf{A}, \mathbf{Z}, \mathbf{O}$ to denote the full matrices of sequence length $s$. Under this notation, the prefill phase corresponds to the special case $t = 0$, while the decoding phase corresponds to $t > 0$.

**Cache Eviction.** Eviction is triggered when the sequence length $s$ of the KV cache exceeds a predefined budget $N$.

After each forward pass, an eviction algorithm selects $s - N$ rows from $\mathbf{K}$ and $\mathbf{V}$ for permanent deletion. By enforcing a fixed cache budget, the model can decode arbitrarily long sequences while maintaining a bounded memory footprint.

## 4. Optimal Brain Cache (OBCache)

In this section, we present the details of OBCACHE. In Section 4.1, we first formulate cache eviction as a layer-wise structured pruning problem, with the objective of minimizing perturbations in attention outputs. Section 4.2 details our analytical solution to this perturbation minimization problem based on second-order Taylor approximations. In Section 4.3, we show that our framework recovers existing attention-based scoring methods as special cases. We then provide a qualitative example to highlight the effectiveness of our formulation in Section 4.4. An overview of the OBCACHE scoring mechanism is shown in Figure 1.

### 4.1. Cache Eviction via Perturbation Minimization

Cache eviction can be viewed as a form of layer-wise structured pruning, where the saliency of each pruning unit (i.e., a key or a value vector) is quantified by the error it induces in the layer output. However, cache eviction introduces a unique challenge not encountered in classical model pruning: due to the autoregressive nature of generation, the actual error caused by removing a key–value pair at step $s$ affects only future attention outputs, $\mathbf{o}_{s+1}, \mathbf{o}_{s+2}, \ldots$, which are inaccessible at eviction time. We refer to this unobservable quantity as the *true eviction error*.

Although the true eviction error is not directly available, we observe that it can be effectively approximated by measuring perturbations in recent historical attention outputs, specifically $\mathbf{o}_s, \mathbf{o}_{s-1}, \ldots$, when the corresponding KV vectors are pruned. We refer to this measurable surrogate as the *pruning-induced eviction error*, and use it as a proxy objective to estimate token saliency. Building on this insight, we formulate cache eviction as a layer-wise structured pruning problem by treating $\mathbf{V}$ and $\mathbf{K}$ as pruning variables.

**Definition 4.1** (Token Saliency via Pruning-Induced Error). Let $\widehat{\mathbf{V}} = \mathbf{V} + \delta\mathbf{V}$ and $\widehat{\mathbf{K}} = \mathbf{K} + \delta\mathbf{K}$ denote the perturbed value and key matrices after pruning, with resulting perturbed attention weights $\widehat{\mathbf{A}}$ and outputs $\widehat{\mathbf{O}}$. The saliency score of a token position $p$ is defined as the change in recent historical attention outputs $\mathbf{O}$ when $\mathbf{v}_p$ and $\mathbf{k}_p$ are pruned:

$$\boldsymbol{S}_p := \mathcal{L}_{\boldsymbol{e}_p^\top [\widehat{\mathbf{V}} \ \widehat{\mathbf{K}}] = \mathbf{0}}(\widehat{\mathbf{V}}, \widehat{\mathbf{K}})$$

$$= f\left( \sigma\left( \frac{\mathbf{Q}\widehat{\mathbf{K}}^\top}{\sqrt{d}} \right) \widehat{\mathbf{V}} \Big|_{\boldsymbol{e}_p^\top [\widehat{\mathbf{V}} \ \widehat{\mathbf{K}}] = \mathbf{0}} - \sigma\left( \frac{\mathbf{Q}\mathbf{K}^\top}{\sqrt{d}} \right) \mathbf{V} \right), \quad (1)$$

where $\boldsymbol{e}_p$ is a unit vector selecting the $p$-th row of $\widehat{\mathbf{V}}$ and $\widehat{\mathbf{K}}$, and $f(\cdot)$ is a norm function. Following prior works on

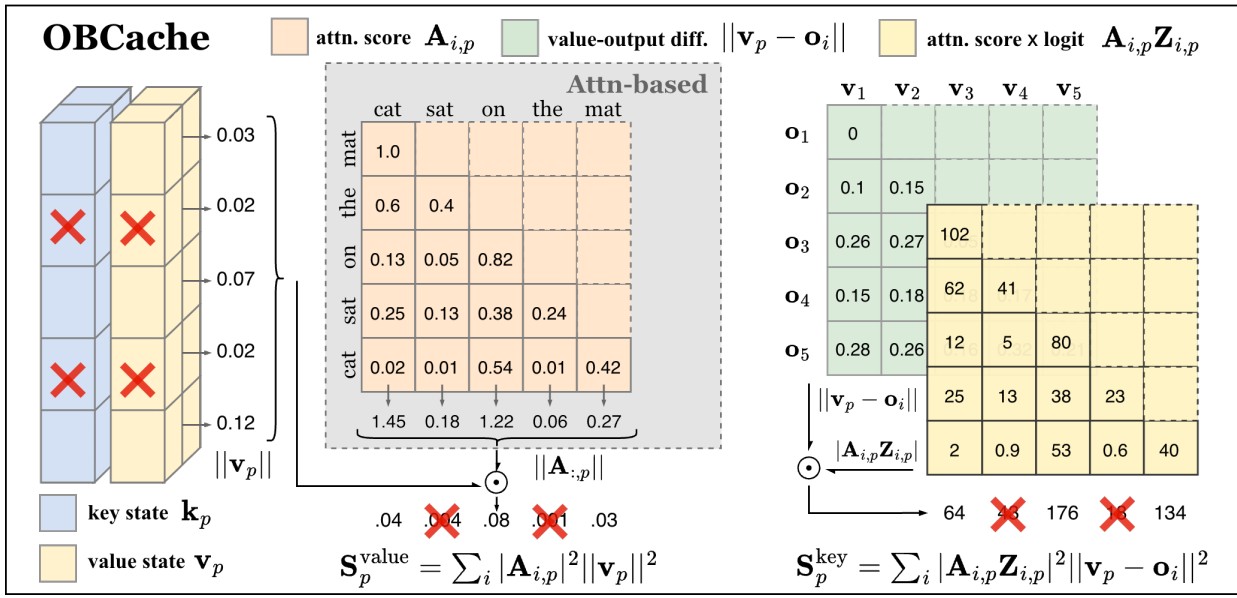

*Figure 1.* Overview of the OBCACHE scoring mechanism. The diagram shows the eviction process using value-pruning (left) and key-pruning scores (right). Unlike prior methods based solely on attention statistics (gray region), OBCACHE further incorporates value states, attention logits, and outputs to estimate token saliency, explicitly targeting the minimization of eviction-induced errors.

layer-wise LLM weight pruning (Frantar & Alistarh, 2023; Sun et al., 2024), we adopt the squared Frobenius norm $\|\cdot\|_F^2$ due to its well-defined gradients and Hessians.

To illustrate the effectiveness of the pruning-induced error as a proxy, we present an empirical example in Section 4.3.

### 4.2. Cache Pruning Scores

Directly recomputing Equation 1 for every position $p$ requires repeated attention operations and is computationally infeasible. Inspired by Optimal Brain Damage (OBD) (LeCun et al., 1989), we approximate it via a second-order Taylor expansion around the unperturbed point $(\mathbf{V}, \mathbf{K})$.

**Theorem 4.2.** *Let $\mathbf{H}^{vv}, \mathbf{H}^{kk}, \mathbf{H}^{vk}$ be the Hessians of $\mathcal{L}$ with respect to $\widehat{\mathbf{V}}, \widehat{\mathbf{K}}$, and their cross-term. The pruning-induced eviction error $\mathcal{L}$ expanded around $(\mathbf{V}, \mathbf{K})$ up to second order is given by:*

$$\mathcal{L}(\widehat{\mathbf{V}}, \widehat{\mathbf{K}}) = \frac{1}{2}\delta\mathbf{V}^\top\mathbf{H}^{vv}\delta\mathbf{V} + \frac{1}{2}\delta\mathbf{K}^\top\mathbf{H}^{kk}\delta\mathbf{K}$$
$$+ \delta\mathbf{V}^\top\mathbf{H}^{vk}\delta\mathbf{K} + \mathcal{O}\big(\|(\delta\mathbf{V}, \delta\mathbf{K})\|^3\big). \quad (2)$$

*When only $\mathbf{v}_p$ and $\mathbf{k}_p$ are pruned with others unchanged, i.e., $\mathbf{e}_p^\top[\widehat{\mathbf{V}}\ \widehat{\mathbf{K}}] = \mathbf{0}$, the above pruning-induced eviction error further simplifies, yielding the saliency score for token position $p$ approximated to second order:*

$$S_p \overset{second}{\underset{order}{=}} \frac{1}{2}\mathbf{v}_p^\top\mathbf{H}_{pp}^{vv}\mathbf{v}_p + \frac{1}{2}\mathbf{k}_p^\top\mathbf{H}_{pp}^{kk}\mathbf{k}_p + \mathbf{v}_p^\top\mathbf{H}_{pp}^{vk}\mathbf{k}_p, \quad (3)$$

In Equation 2, the first-order terms vanish at the expansion point since $\widehat{\mathbf{O}} - \mathbf{O} = \mathbf{0}$. The simplification to Equation 3 uses the fact that the off-diagonal blocks of $\mathbf{H}^{vv}, \mathbf{H}^{kk}, \mathbf{H}^{vk}$ do not contribute to $\mathcal{L}$. This mirrors the diagonal assumption adopted in OBD. The full proof is provided in Appendix A.

By explicitly evaluating the Hessian sub-blocks $\mathbf{H}_{pp}^{vv}, \mathbf{H}_{pp}^{kk}$, and $\mathbf{H}_{pp}^{vk}$, we next derive closed-form eviction scores that are both interpretable and efficient to compute. Currently, Equation 3 captures the joint impact of perturbing both $\mathbf{V}$ and $\mathbf{K}$. We also consider simplified variants where either $\mathbf{V}$ or $\mathbf{K}$ is perturbed independently while the other remains fixed. As a result, OBCACHE features three output-aware saliency scores, as demonstrated in Propositions 4.3–4.5.

**Proposition 4.3** (Value-Pruning Score). *When only $\mathbf{V}$ is the pruning unit, i.e., $\mathbf{e}_p^\top\widehat{\mathbf{V}} = \mathbf{0}$, the pruning-induced eviction error reduces to the first term in Equation 3, which is:*

$$S_p^{\text{value}} = \frac{1}{2}\mathbf{v}_p^\top\mathbf{H}_{pp}^{vv}\mathbf{v}_p = \sum_i |\mathbf{A}_{i,p}|^2\|\mathbf{v}_p\|^2. \quad (4)$$

This score computes the squared $\ell_2$-norm of the $p$-th column of the attention weight matrix, scaled by the squared $\ell_2$-norm of the value vector $\mathbf{v}_p$. It corresponds to the value-aware score proposed in VATP (Guo et al., 2024) and CriticalKV (Feng et al., 2025). Note that these methods use the $\ell_1$-norm of the attention weights and value states. Our choice of a smooth $\ell_2$ objective is required to derive key-pruning scores in closed form. If we instead adopt an $\ell_1$ objective for value pruning, the resulting score reduces to

the same form. The full proof, including the derivation of the Hessian sub-block $\mathbf{H}_{pp}^{vv}$ is provided in Appendix A.2.

**Proposition 4.4** (Key-Pruning Score). *When only* $\mathbf{K}$ *is the pruning unit, i.e.,* $\boldsymbol{e}_p^\top \widehat{\mathbf{K}} = \mathbf{0}$*, the pruning-induced eviction error reduces to the second term in Equation 3, which is:*

$$S_p^{\text{key}} = \frac{1}{2}\mathbf{k}_p^\top \mathbf{H}_{pp}^{kk}\mathbf{k}_p = \sum_i |\mathbf{A}_{i,p}\mathbf{Z}_{i,p}|^2 \|\mathbf{v}_p - \mathbf{o}_i\|^2. \quad (5)$$

This score captures the deviation between the value vector and attention output, weighted by both the attention weights and the pre-softmax logits. Key pruning generally incurs larger errors than value pruning, as it alters the entire attention distribution. The proof is provided in Appendix A.3.

**Proposition 4.5** (Joint-Pruning Score). *When* $\mathbf{V}$ *and* $\mathbf{K}$ *are treated as a combined pruning unit, the pruning-induced eviction error, as given by Equation 3, has the form:*

$$\begin{aligned} S_p^{\text{joint}} &= S_p^{\text{value}} + S_p^{\text{key}} + \mathbf{v}_p^\top \big[\mathbf{H}^{vk}\big]_{pp}\mathbf{k}_p \\ &= S_p^{\text{value}} + S_p^{\text{key}} + 2\sum_i |\mathbf{A}_{i,p}|^2 \mathbf{Z}_{i,p}(\|\mathbf{v}_p\|^2 - \mathbf{v}_p^\top \mathbf{o}_i). \end{aligned} \quad (6)$$

This score captures both the individual and interactive effects of pruning the key and value states, providing the most comprehensive estimate of the pruning-induced eviction error (Equation 1). The derivation of the cross-term is provided in in Appendix A.4.

We note that OBCACHE scores can be applied for both prefill and decoding. In the prefill phase, the saliency score $S_p$ can be used to greedily evict multiple tokens to achieve a desired sparsity in a one-shot manner. During decoding, by accumulating $S_p$ over time, OBCACHE can support real-time updates to token-wise saliency, enabling dynamic KV cache eviction as generation progresses. For models using Grouped-Query Attention (Ainslie et al., 2023), we include an additional score derivation in Appendix A.5.

### 4.3. Connection to Existing Methods

Our framework naturally recovers existing attention-based eviction strategies as special cases. To show this, we introduce an additional index $w \in [1, s]$ and use $\mathbf{A}_{w:s}$ to denote the rows of the attention matrix $\mathbf{A}$ corresponding to the query positions from $w$ to $s$. Consider an alternative formulation that minimizes perturbations not in the attention outputs $\mathbf{O}$, but in the historical attention rows $\mathbf{A}_{w:s}$. In this case, the pruning-induced error reduces to:

$$S_p^{\text{attn}} := \mathcal{L}_{\widehat{\mathbf{A}}_{:,p}=\mathbf{0}}(\widehat{\mathbf{A}}) = \left\|\widehat{\mathbf{A}}_{w:s}\Big|_{\widehat{\mathbf{A}}_{:,p}=\mathbf{0}} - \mathbf{A}_{w:s}\right\|_{1,1}, \quad (7)$$

where the pruning unit also simplifies to an attention matrix column. We refer to query positions from $w$ to $s$ as the *perturbation window* in order to connect and formalize the

accumulation strategies used in prior methods. Equation 7 can be directly simplified to the form:

$$S_p^{\text{attn}} = \sum_{i=w}^{s} |\mathbf{A}_{i,p}|, \quad (8)$$

which corresponds to an $\ell_1$-norm variant of our value-pruning score $S_p^{\text{value}}$, but without incorporating any value-state information. This formulation connects directly to several recent methods. Specifically, $H_2O$ (Zhang et al., 2023) sets $w = 1$, accumulating attention weights over the entire sequence history. TOVA (Oren et al., 2024) sets $w = s$, targeting only the most recent query position. SnapKV (Li et al., 2024) adopts a short window (i.e., $w \gg 1$), emphasizing recent attentions. By varying the choice of $w$, these methods effectively target different perturbation windows.

In OBCACHE, when we also relax the objective to the output error within the perturbation window:

$$S_p = \left\| \sigma\Big(\frac{\mathbf{Q}_{w:s}\widehat{\mathbf{K}}^\top}{\sqrt{d}}\Big)\widehat{\mathbf{V}} \Big|_{\boldsymbol{e}_p^\top[\widehat{\mathbf{V}}\ \widehat{\mathbf{K}}]=\mathbf{0}} - \sigma\Big(\frac{\mathbf{Q}_{w:s}\mathbf{K}^\top}{\sqrt{d}}\Big)\mathbf{V} \right\|_F^2, \quad (9)$$

the resulting scores also become localized to the same set of query positions $w$ through $s$. Therefore, by accumulating the pruning-induced eviction error within different perturbation windows, OBCACHE scores generalize these prior approaches by further introducing output-aware signals, enabling more informed eviction decisions that go beyond raw attention statistics.

### 4.4. Effectiveness of Pruning-Induced Eviction Error

To further support our pruning-based formulation for cache eviction, we present a qualitative analysis on a Needle-In-A-Haystack (Kamradt, 2023) passkey retrieval task, as shown in Figure 2. We begin by establishing an oracle baseline based on the *true eviction error*, which is measured as the perturbation in the first decoding-step output $\mathbf{o}_{l+1}$ caused by evicting each cached token during the prefill phase. The top-$k$ tokens with the largest oracle errors are treated as ground-truth important positions.

To evaluate the effectiveness of the *pruning-induced eviction error* as a proxy, we compute Equation 1 exactly for each candidate position in the prefill phase and select the top-$k$ accordingly. As shown in the figure, when the perturbation window is appropriately chosen, the proxy achieves up to 85% recall of the oracle top-$k$ selections. Next, we demonstrate the superiority of OBCACHE. Although derived via second-order Taylor approximation, OBCACHE scores achieve nearly identical ranking performance to the exact proxy, while being significantly more efficient due to their closed-form expression. Compared to attention-based methods, our scores consistently yield higher oracle recall, showcasing the benefit of incorporating output-aware signals into saliency estimation.

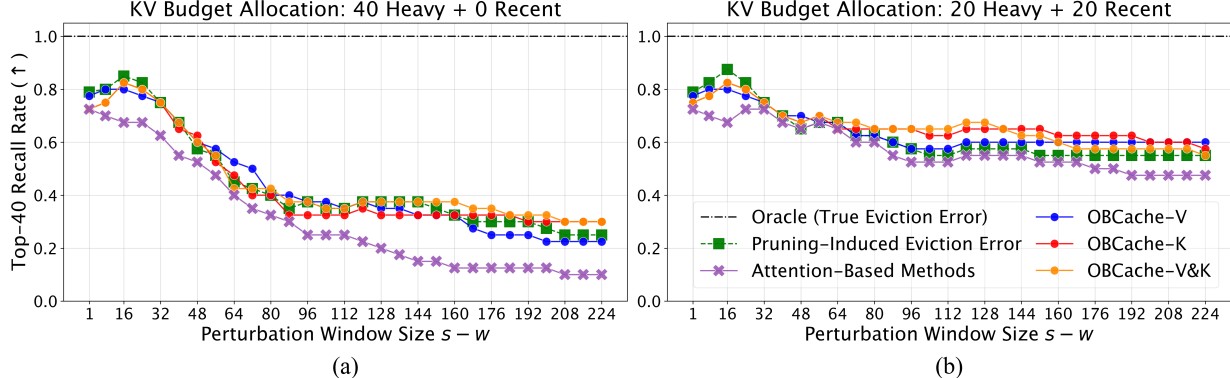

*Figure 2.* Recall rate of the top-40 salient tokens identified by the oracle eviction error. Results are collected from a 4K-context passkey retrieval task using LLaMA-3.2-3B-Instruct. The left plot (a) shows recall of top-40 tokens selected by different scoring methods. The right plot (b) demonstrates how allocating a fixed recent window improves oracle recall when perturbation windows are large, mitigating the structural bias that disproportionately favors earlier tokens.

We observe that recall degrades when the perturbation window becomes larger. This issue stems from attention causality: earlier tokens attend to more queries and thus accumulate disproportionately higher saliency scores, introducing a structural bias that favors retaining initial tokens. To mitigate this, $H_2O$ reserves a portion of the cache budget for a recent window, ensuring that the most recent tokens are never evicted. This policy is also complementary to OB-CACHE, and we find that recall further improves when a fixed 20-token window is reserved, as shown in Figure 2b.

## 5. Experiments

In this section, we conduct comprehensive experiments to evaluate the effectiveness of OBCACHE in improving existing score-based KV cache eviction methods across diverse long-context benchmarks. We evaluate OBCACHE in both static cache eviction in the prefill stage (Section 5.2) and dynamic cache eviction in the decoding stage (Section 5.3). We additionally report efficiency results to demonstrate the negligible computational overhead introduced by our output-aware scoring mechanism compared to attention-based methods in both prefill and decoding (Appendix D.1).

### 5.1. Experimental Setup

#### 5.1.1. DATASETS.

For prefill-stage cache eviction, we evaluate on two widely used long-context benchmarks: RULER (Hsieh et al., 2024) and LongBench (Bai et al., 2024). RULER is a synthetic benchmark suite consisting of 13 tasks designed to evaluate long-context retrieval, multi-hop reasoning, aggregation, and positional robustness under controlled context lengths and compression settings. Following prior KV cache eviction works, we evaluate under 4K (denoted as RULER-4K) and 32K (denoted as RULER-32K) context lengths.

LongBench evaluates long-context understanding through 16 real-world datasets spanning six task categories: single-document QA, multi-document QA, summarization, few-shot learning, synthetic reasoning, and code completion. The average input length across all datasets is 6,711 words, making KV cache optimization critical for efficient inference. For decoding-stage cache eviction, we follow the setup in StreamingLLM (Xiao et al., 2024) and report language modeling perplexity on PG19 (Rae et al., 2019), a dataset of 100 books with an average length of 70K tokens.

#### 5.1.2. BASELINES.

We compare OBCACHE with four representative KV cache eviction methods: $H_2O$ (Zhang et al., 2023), TOVA (Oren et al., 2024), SnapKV (Li et al., 2024), and AdaKV (Feng et al., 2026), all of which rely on attention statistics as token importance indicators but differ in their eviction and budget allocation strategies. $H_2O$ accumulates historical attention weights while retaining a fixed recent window. TOVA selects tokens based on the most recent query attention and does not require a recent window. SnapKV, designed for prefill-stage eviction, scores tokens based on a recent attention window and applies pooling to smooth token importance. AdaKV further introduces adaptive head-wise budget allocation, globally redistributing cache budgets across attention heads based on their cumulative attention weights. We integrate OBCACHE into these baselines by replacing their original attention-based scores with our output-aware scores while preserving their original eviction strategies. We refer to our eviction methods using the value-pruning score as OBCACHE-V, the key-pruning score as OBCACHE-K, and the joint pruning score as OBCACHE-V&K.

To compare against existing value-aware scoring methods, we additionally include comparisons with VATP (Guo et al., 2024) and CriticalKV (Feng et al., 2025) in Section 5.2.4.

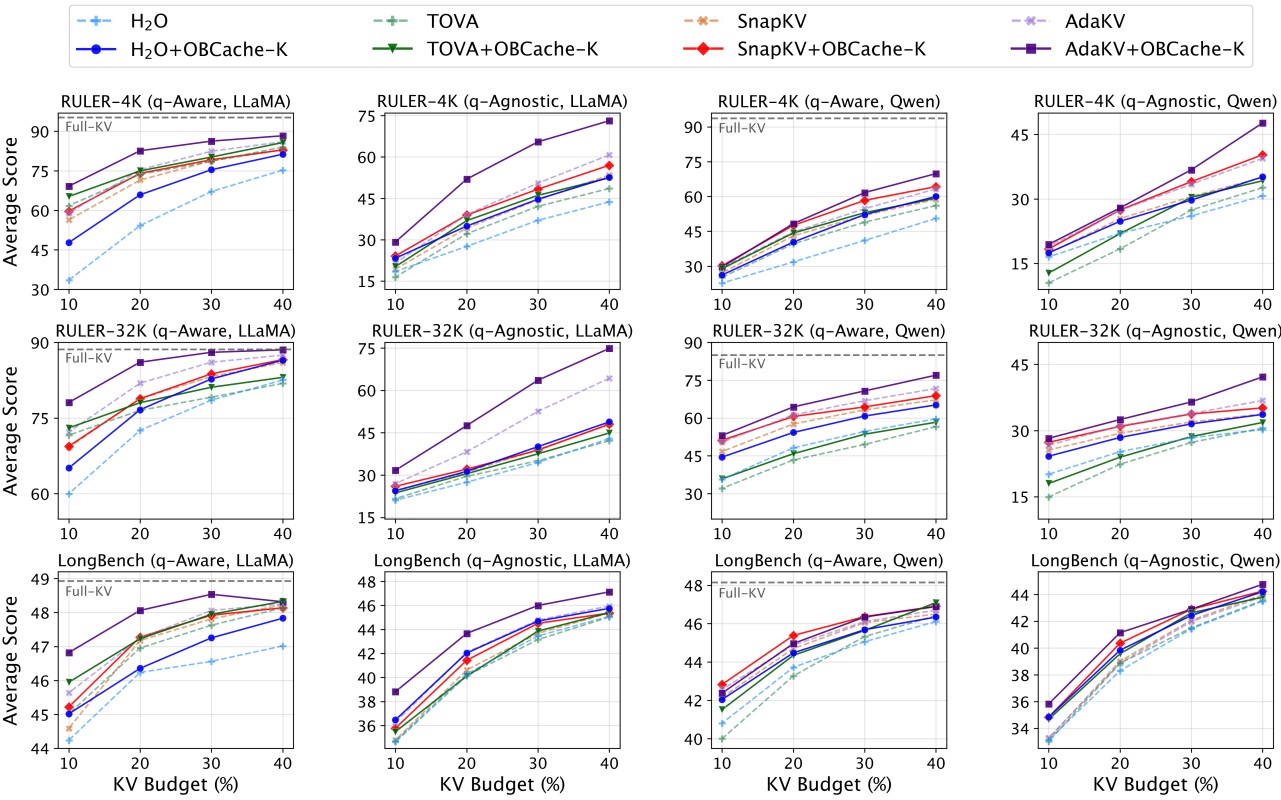

*Figure 3.* Evaluation of prefill-stage cache eviction on LLaMA-3.1-8B and Qwen-2.5-7B. When integrated with OBCACHE scores (solid lines), existing attention-only baselines (dashed lines) consistently achieve superior compression-performance trade-offs on RULER and LongBench benchmarks under both query-aware and query-agnostic settings.

VATP uses an $\ell_1$-norm value-based importance score, while CriticalKV combines an attention-based preselection stage with a secondary value-aware scoring stage.

### 5.1.3. IMPLEMENTATION DETAILS.

We use LLaMA-3.1-8B-Instruct (Grattafiori et al., 2024) and Qwen-2.5-7B-Instruct (Bai et al., 2023) as our backbone LLMs, both of which natively support 128K context windows and adopt Grouped-Query Attention (Ainslie et al., 2023) to reduce KV cache memory. All experiments are implemented using the Transformers library (Wolf et al., 2020) based on the KVPress repository (Devoto et al., 2025). To improve efficiency, the prefill-stage attention operation is implemented via FlashAttention-2 (Dao, 2024), which reduces memory overhead for long-context inputs.

In static cache eviction, KV compression is performed only once before decoding begins. This reflects realistic deployment scenarios where prefill-stage KV cache dominates memory usage compared to decoding-stage cache. We evaluate all methods under cache budget ratios of 10%, 20%, 30%, and 40%. The cache budget is determined as a fixed percentage of the input prompt length. For decoding-stage eviction, we follow $H_2O$ and maintain a fixed recent window

while evicting tokens at every decoding step to evaluate the dynamic impact of different saliency scores.

Following KVPress and AdaKV, we evaluate under both *query-aware* and *query-agnostic* cache compression settings. In the query-aware setting, cache eviction is performed after observing the downstream query, following the standard setup used in prior methods such as SnapKV. In the more challenging query-agnostic setting (Feng et al., 2026; Kim et al., 2026), the context KV cache must be compressed before future queries are observed, which better reflects real-world cache reuse scenarios such as prefix caching and multi-turn dialogue. Full implementation details are provided in Appendix C.

### 5.2. Evaluation of Prefill Cache Eviction

We present RULER and LongBench performance curves under varying cache budgets in Figure 3. We report results for OBCACHE-K, since it consistently outperforms OBCACHE-V and mostly matches OBCACHE-V&K's performance while requiring lower complexity. Additionally, Table 1 reports detailed average RULER accuracy for LLaMA-3.1-8B, comparing baselines with their three OBCACHE-enhanced counterparts. Full results across all 13 RULER tasks and 16

*Table 1.* Accuracy results on RULER for LLaMA-3.1-8B. Each value corresponds to an average exact match score over 13 tasks from the RULER benchmark. For each baseline method, context length, and compression setting, the best average accuracy of all compression rates is in **bold** and the second best is underlined.

| KV Budget (%) | RULER-4K (Q-Aware) | | | | | RULER-4K (Q-Agnostic) | | | | | RULER-32K (Q-Aware) | | | | | RULER-32K (Q-Agnostic) | | | | |
|---|---|---|---|---|---|---|---|---|---|---|---|---|---|---|---|---|---|---|---|---|
| | 10 | 20 | 30 | 40 | Avg. | 10 | 20 | 30 | 40 | Avg. | 10 | 20 | 30 | 40 | Avg. | 10 | 20 | 30 | 40 | Avg. |
| Full KV | | | 95.3 | | | | | | | | | | 88.6 | | | | | | | |
| H$_2$O | 33.5 | 54.2 | 67.1 | 75.3 | 57.5 | 18.5 | 27.6 | 37.0 | 43.7 | 31.7 | 60.0 | 72.5 | 78.5 | 82.5 | 73.4 | 21.1 | 27.5 | 34.4 | 42.9 | 31.5 |
| + OBCACHE-V | 37.7 | 57.1 | 68.6 | 76.0 | 59.9 | 19.6 | 28.2 | 36.5 | 44.9 | 32.3 | 62.7 | 73.5 | 80.4 | 84.3 | 75.2 | 20.0 | 27.7 | 36.7 | 46.0 | 32.6 |
| + OBCACHE-K | 47.7 | 65.9 | 75.4 | 81.3 | 67.6 | 23.3 | 34.9 | 44.7 | 52.7 | 38.9 | 65.1 | 76.6 | 82.7 | 86.5 | 77.7 | 24.4 | 31.3 | 40.1 | 48.8 | 36.2 |
| + OBCACHE-V&K | 46.3 | 67.5 | 75.6 | 82.0 | **67.8** | 23.3 | 35.4 | 46.4 | 54.7 | **40.0** | 66.0 | 76.8 | 83.0 | 86.7 | **78.1** | 24.9 | 31.9 | 40.6 | 49.5 | **36.7** |
| TOVA | 61.8 | 73.6 | 78.6 | 84.0 | 74.5 | 16.4 | 32.2 | 42.1 | 48.6 | 34.8 | 71.6 | 76.5 | 79.1 | 81.9 | 77.3 | 21.6 | 29.5 | 35.1 | 42.2 | 32.1 |
| + OBCACHE-V | 64.9 | 74.8 | 80.0 | 85.4 | 76.3 | 18.8 | 34.6 | 44.3 | 50.7 | 37.1 | 72.3 | 76.9 | 79.9 | 83.2 | 78.1 | 22.3 | 30.9 | 36.8 | 44.0 | 33.5 |
| + OBCACHE-K | 65.3 | 75.0 | 80.2 | 85.7 | 76.5 | 20.3 | 36.9 | 46.1 | 52.5 | 39.0 | 73.0 | 78.0 | 81.1 | 83.1 | 78.8 | 23.7 | 30.5 | 37.5 | 44.9 | 34.1 |
| + OBCACHE-V&K | 65.5 | 75.3 | 80.1 | 85.8 | **76.7** | 21.5 | 37.0 | 46.0 | 52.3 | **39.2** | 72.7 | 77.9 | 81.4 | 83.2 | **78.8** | 23.7 | 30.9 | 37.8 | 44.7 | **34.3** |
| SnapKV | 56.4 | 71.5 | 78.8 | 83.0 | 72.4 | 19.4 | 34.2 | 44.5 | 53.4 | 37.9 | 69.2 | 78.9 | 83.2 | 86.0 | 79.3 | 24.5 | 31.1 | 39.0 | 47.9 | 35.6 |
| + OBCACHE-V | 55.8 | 72.2 | 77.6 | 82.0 | 71.9 | 18.6 | 32.1 | 41.7 | 50.7 | 35.8 | 68.5 | 78.0 | 83.7 | 86.6 | 79.2 | 24.3 | 30.3 | 37.7 | 46.3 | 34.7 |
| + OBCACHE-K | 59.7 | 74.0 | 79.2 | 82.9 | **73.9** | 24.1 | 38.9 | 48.3 | 57.0 | **42.1** | 69.4 | 78.8 | 83.8 | 86.6 | 79.7 | 26.0 | 32.1 | 38.8 | 48.0 | 36.2 |
| + OBCACHE-V&K | 59.0 | 73.6 | 78.9 | 82.9 | 73.6 | 22.4 | 38.5 | 48.8 | 58.0 | 41.9 | 69.7 | 79.3 | 83.6 | 86.5 | **79.8** | 26.2 | 31.9 | 39.3 | 48.2 | **36.4** |
| AdaKV | 58.9 | 75.4 | 82.4 | 86.0 | 75.7 | 21.5 | 39.1 | 50.5 | 60.7 | 43.0 | 72.2 | 81.9 | 86.1 | 87.5 | 81.9 | 26.9 | 38.2 | 52.6 | 64.3 | 45.5 |
| + OBCACHE-V | 66.2 | 81.4 | 86.0 | 87.6 | 80.3 | 23.9 | 49.0 | 61.9 | 70.9 | 51.4 | 76.5 | 85.4 | 87.7 | 88.6 | 84.6 | 30.3 | 46.1 | 60.9 | 72.4 | 52.4 |
| + OBCACHE-K | 69.2 | 82.6 | 86.3 | 88.3 | 81.6 | 29.2 | 52.0 | 65.5 | 73.2 | 55.0 | 78.1 | 86.1 | 88.0 | 88.5 | **85.2** | 31.7 | 47.5 | 63.6 | 74.8 | 54.4 |
| + OBCACHE-V&K | 70.1 | 82.7 | 86.3 | 88.4 | **81.9** | 30.0 | 52.2 | 65.1 | 73.6 | **55.2** | 78.2 | 86.1 | 88.1 | 88.6 | **85.2** | 32.6 | 48.1 | 64.2 | 75.4 | **55.1** |

LongBench tasks are provided in Appendix D.2.

### 5.2.1. RESULTS ON RULER

Across all baselines, compression settings, cache budgets, context lengths, and model families, integrating OBCACHE scores consistently improves task accuracy. As illustrated in Table 1, when applied to H$_2$O, OBCACHE yields large average accuracy gains across all compression ratios, achieving more than 10% accuracy increase on RULER-4K and 5% on RULER-32K. TOVA is also consistently improved by 2%∼5% in average accuracy. For SnapKV, the improvement is relatively smaller in query-aware settings. This is likely because its heuristic 1D-pooling step is tailored to noisy attention-only scores. Since OBCACHE produces output-aware saliency scores that are already more accurate, the marginal benefit of such smoothing may be reduced. Notably, when integrated with AdaKV, our strongest baseline, OBCACHE yields substantial additional gains. For example, we observe nearly 15% accuracy increase on the query-agnostic RULER-4K setting with a 30% cache budget. These results demonstrate the consistent effectiveness of OBCACHE scores across diverse settings. Overall, all four baselines benefit from integrating OBCACHE, highlighting the advantage of our output-aware scoring mechanism over attention-based heuristics.

### 5.2.2. COMPARISON OF OBCACHE SCORES

In addition to the comparison with baselines, Table 1 also provides an ablation study of the three OBCACHE score variants derived under different pruning unit assumptions. We observe that eviction scores derived from key pruning, i.e., OBCACHE-K and OBCACHE-V&K, consistently out-

perform the value-only variant OBCACHE-V. This aligns with our expectation that pruning keys has a larger impact due to their role in shaping attention distributions and ultimately the model predictions. Accounting for key sensitivity therefore yields more accurate saliency estimation and better performance. Although the OBCACHE-V variant performs slightly worse, it remains appealing for its simplicity: it requires only an additional scaling factor based on value-state norms, making it nearly as efficient as attention-only scores while still delivering substantial performance gains.

### 5.2.3. RESULTS ON LONGBENCH

As shown in Figure 3, we similarly observe consistent Long-Bench performance improvements when integrating OB-CACHE into all four baselines for both LLaMA-3.1 and Qwen-2.5 models. The gains tend to become more pronounced under higher compression ratios, where output-aware saliency estimation becomes increasingly important. Notably, for AdaKV, our strongest baseline, OBCACHE-K improves performance by +1.2 in the query-aware setting and +2.6 in the query-agnostic setting under a 10% cache budget on LLaMA. This provides strong evidence that OB-CACHE scores remain effective on real-world tasks and further advance the state of the art. Overall, these results confirm that replacing attention-only scores with OBCACHE scores enables more effective preservation of task-critical tokens under diverse compression settings.

### 5.2.4. COMPARISON TO VALUE-AWARE BASELINES

As discussed in Section 4.2, OBCACHE-V reduces to the scores proposed in VATP and CriticalKV under different norm choices. Since our OBD-based formulation flexibly

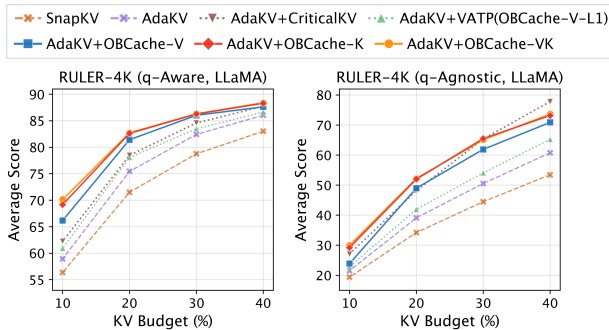

*Figure 4.* Comparison between three OBCACHE scores and existing value-aware score baselines, VATP (denoted as OBCACHE-V-L1) and CriticalKV, when integrated into AdaKV. Performance is reported on the RULER-4K benchmark with LLaMA-3.1-8B.

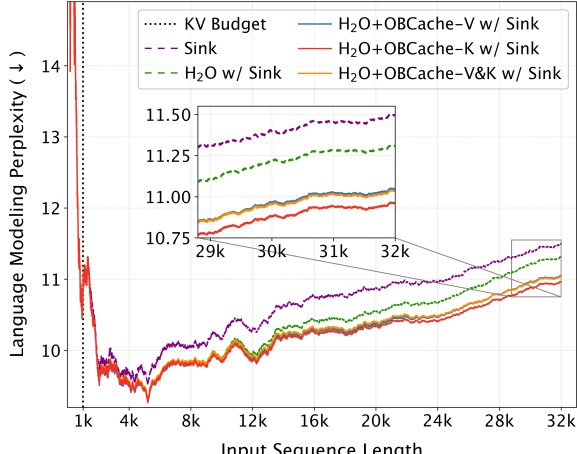

*Figure 5.* Evaluation of dynamic cache eviction on PG19 test set. We prompt Llama-3.1-8B with 1 to 32K tokens and measure the perplexity of output tokens at varying context lengths. The KV cache budget (dotted line) for all methods is fixed at 1024 tokens.

defines pruning units and objectives, it additionally yields two key-pruning-oriented scores that existing value-aware baselines do not consider. To demonstrate the effectiveness of our general framework, we reproduce these value-aware baselines when integrated into AdaKV on RULER-4K. As shown in Figure 4, in the query-aware setting, OBCACHE-K and OBCACHE-V&K achieve the strongest performance among all methods. OBCACHE-V follows and outperforms both CriticalKV and VATP, especially at higher compression ratios. In the query-agnostic setting, CriticalKV is competitive at the lowest compression ratio (40% budget); however, it still underperforms OBCACHE-K and OBCACHE-V&K at higher compression ratios. Overall, VATP provides the smallest improvements over AdaKV, CriticalKV underperforms our OBCACHE scores in most settings, and OBCACHE-K and OBCACHE-V&K achieve the best performance-compression trade-offs. These results demonstrate that the gains of OBCACHE do not arise merely from rederiving a known value-aware score within a cleaner framework, but from the broader OBD-based formulation and, importantly, from the additional key-aware scoring terms that more effectively capture output sensitivity.

### 5.3. Evaluation of Decoding Cache Eviction

Beyond static prefill-stage cache eviction, OBCACHE is also applicable to decoding scenarios, where eviction decisions must be made dynamically. To evaluate this setting, we measure cumulative language modeling perplexity at varying sequence lengths on the PG19 dataset using a fixed KV cache budget of 1024 tokens. Following Xiao et al. (2024), we retain 4 fixed initial tokens for all methods, as these tokens act as attention sinks and are crucial for long-sequence generation quality. As shown in Figure 5, the Sink baseline, which statically allocates the cache to fixed initial and recent tokens, exhibits the highest perplexity. The $H_2O$ baseline, which dynamically selects important tokens based on accumulated attention statistics, performs moderately

better. In contrast, all OBCACHE variants, which instead accumulate output-aware saliency scores over time, consistently outperform $H_2O$ across all sequence lengths. We also observe that OBCACHE-V&K does not outperform OBCACHE-K, suggesting that an improved combination of key-pruning and value-pruning scores may exist beyond the current plain additive formulation. SnapKV and AdaKV are not evaluated, since they are specifically designed for static prefill-stage eviction and do not inherently support dynamic decoding eviction. Overall, these results demonstrate that our output-aware saliency scores can improve memory utilization while better preserving long-range dependencies critical for continuous high-quality generation.

## 6. Conclusion

In this work, we introduce OBCACHE, a principled scoring framework for KV cache eviction in language model inference grounded in the Optimal Brain Damage theory. By casting cache eviction as a layer-wise structured pruning problem, we derive token saliency scores that aim to minimize the impact of removal on attention outputs. Experiments across both prefilling and decoding scenarios on long-context benchmarks show that OBCACHE consistently improves state-of-the-art baselines, achieving superior performance–compression trade-offs with negligible overhead. Beyond the proposed scores, OBCACHE provides a flexible foundation for extending to additional KV cache compression tasks: by modifying the pruning objective or pruning units, the framework can be naturally applied to settings such as channel-wise KV pruning or cache merging via relaxed diagonal approximations. These directions open promising paths toward more effective and theoretically grounded KV cache management for long-context LLMs.

## Impact Statement

This paper presents work whose goal is to advance the field of Machine Learning. There are many potential societal consequences of our work, none of which we feel must be specifically highlighted here.

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

# A. Theoretical Analysis

In this section, we present the full proof and derivations of the OBCACHE scores in Equation 4, Equation 5 and Equation 6. We use the notation previously defined in Section 3.

## A.1. OBCache Objective Function

As demonstrated in Section 4.1 and Section 4.3, we formulate the saliency score for a candidate key-value token in Equation 1 and Equation 9, which is the *pruning-induced eviction error* within a *perturbation window* for query positions $w$ through $s$:

$$\mathcal{L}(\widehat{\mathbf{V}}, \widehat{\mathbf{K}}) := \left\| \widehat{\mathbf{O}}_{w:s} - \mathbf{O}_{w:s} \right\|_F^2 = \left\| \text{softmax}\left(\frac{\mathbf{Q}_{w:s}\widehat{\mathbf{K}}^\top}{\sqrt{d}}\right)\widehat{\mathbf{V}} - \text{softmax}\left(\frac{\mathbf{Q}_{w:s}\mathbf{K}^\top}{\sqrt{d}}\right)\mathbf{V} \right\|_F^2. \tag{10}$$

In what follows, we decompose the objective $\mathcal{L}$ into element-wise form. We let $\mathbf{O}_{i,j} = \mathbf{a}_i \mathbf{V}_{:,j}$ denote the $j$-th feature element in the $i$-th row vector of the attention output matrix $\mathbf{O}$, where $\mathbf{a}_i$ is the $i$-th row vector of the attention weight matrix. The squared Frobenius norm objective in Equation 10 can be explicitly decomposed into a summation form across all element-wise squared errors:

$$\begin{aligned}
\mathcal{L}(\widehat{\mathbf{V}}, \widehat{\mathbf{K}}) &= \sum_{i=w}^{s} \sum_{j=1}^{d} |\widehat{\mathbf{O}}_{i,j} - \mathbf{O}_{i,j}|^2 \\
&= \sum_{i=w}^{s} \sum_{j=1}^{d} |\text{softmax}\left(\frac{\mathbf{q}_i\widehat{\mathbf{K}}^\top}{\sqrt{d}}\right)\widehat{\mathbf{V}}_{:,j} - \text{softmax}\left(\frac{\mathbf{q}_i\mathbf{K}^\top}{\sqrt{d}}\right)\mathbf{V}_{:,j}|^2 \\
&\stackrel{\triangle}{=} \sum_{i=w}^{s} \sum_{j=1}^{d} \boldsymbol{E}_{i,j}.
\end{aligned} \tag{11}$$

For convenience in the following derivations, we define $\boldsymbol{E}_{i,j} := |\widehat{\mathbf{O}}_{i,j} - \mathbf{O}_{i,j}|^2$ as the squared difference of the output element $\mathbf{O}_{i,j}$ before and after applying a perturbation $\delta\mathbf{V}$ and $\delta\mathbf{K}$. Expanding explicitly, the *element-wise output perturbation* for $\mathbf{O}_{i,j}$ is:

$$\boldsymbol{E}_{i,j}(\widehat{\mathbf{V}}_{:,j}, \widehat{\mathbf{K}}) = |\text{softmax}\left(\frac{\mathbf{q}_i\widehat{\mathbf{K}}^\top}{\sqrt{d}}\right)\widehat{\mathbf{V}}_{:,j} - \text{softmax}\left(\frac{\mathbf{q}_i\mathbf{K}^\top}{\sqrt{d}}\right)\mathbf{V}_{:,j}|^2. \tag{12}$$

Given this element-wise objective, we follow the Optimal Brain Damage (OBD) theory (LeCun et al., 1989) to apply a second-order Taylor expansion of $\boldsymbol{E}_{i,j}$ around $(\mathbf{V}_{:,j}, \mathbf{K})$, and analytically approximate the element-wise output perturbation:

$$\begin{aligned}
\boldsymbol{E}_{i,j}(\widehat{\mathbf{V}}_{:,j}, \widehat{\mathbf{K}}) \stackrel{\text{second}}{\underset{\text{order}}{=}} \ &\boldsymbol{E}_{i,j}(\mathbf{V}_{:,j}, \mathbf{K}) + \delta\mathbf{V}_{:,j}^\top \frac{\partial \boldsymbol{E}_{i,j}(\mathbf{V}_{:,j}, \mathbf{K})}{\partial \widehat{\mathbf{V}}_{:,j}} + \frac{1}{2}\delta\mathbf{V}_{:,j}^\top \frac{\partial^2 \boldsymbol{E}_{i,j}(\mathbf{V}_{:,j}, \mathbf{K})}{\partial \widehat{\mathbf{V}}_{:,j}^2}\delta\mathbf{V}_{:,j} \\
&+ \delta\mathbf{K}^\top \frac{\partial \boldsymbol{E}_{i,j}(\mathbf{V}_{:,j}, \mathbf{K})}{\partial \widehat{\mathbf{K}}} + \frac{1}{2}\delta\mathbf{K}^\top \frac{\partial^2 \boldsymbol{E}_{i,j}(\mathbf{V}_{:,j}, \mathbf{K})}{\partial \widehat{\mathbf{K}}^2}\delta\mathbf{K} \\
&+ \delta\mathbf{V}_{:,j}^\top \frac{\partial^2 \boldsymbol{E}_{i,j}(\mathbf{V}_{:,j}, \mathbf{K})}{\partial \widehat{\mathbf{V}}_{:,j}\partial \widehat{\mathbf{K}}}\delta\mathbf{K} + \mathcal{O}(\|(\delta\mathbf{V}_{:,j}, \delta\mathbf{K})\|^3).
\end{aligned}$$

Through substitution into Equation 12, the constant term $\boldsymbol{E}_{i,j}(\mathbf{V}_{:,j}, \mathbf{K})$ vanishes because we have $\widehat{\mathbf{O}}_{i,j} - \mathbf{O}_{i,j} = 0$ at the expansion point $(\mathbf{V}_{:,j}, \mathbf{K})$. Therefore, the full matrix-wise objective, *pruning-induced eviction error*, approximated via a second-order Taylor series, becomes:

$$\mathcal{L}(\widehat{\mathbf{V}}, \widehat{\mathbf{K}}) \stackrel{\text{second}}{\underset{\text{order}}{=}} \sum_{i=w}^{s} \sum_{j=1}^{d} \delta\mathbf{V}_{:,j}^\top \frac{\partial \boldsymbol{E}_{i,j}(\mathbf{V}_{:,j}, \mathbf{K})}{\partial \widehat{\mathbf{V}}_{:,j}} + \frac{1}{2}\delta\mathbf{V}_{:,j}^\top \frac{\partial^2 \boldsymbol{E}_{i,j}(\mathbf{V}_{:,j}, \mathbf{K})}{\partial \widehat{\mathbf{V}}_{:,j}^2}\delta\mathbf{V}_{:,j} \tag{13}$$

$$+ \sum_{i=w}^{s} \sum_{j=1}^{d} \delta\mathbf{K}^\top \frac{\partial \boldsymbol{E}_{i,j}(\mathbf{V}_{:,j}, \mathbf{K})}{\partial \widehat{\mathbf{K}}} + \frac{1}{2}\delta\mathbf{K}^\top \frac{\partial^2 \boldsymbol{E}_{i,j}(\mathbf{V}_{:,j}, \mathbf{K})}{\partial \widehat{\mathbf{K}}^2} \tag{14}$$

$$+ \sum_{i=w}^{s} \sum_{j=1}^{d} \delta\mathbf{V}_{:,j}^\top \frac{\partial^2 \boldsymbol{E}_{i,j}(\mathbf{V}_{:,j}, \mathbf{K})}{\partial \widehat{\mathbf{V}}_{:,j}\partial \widehat{\mathbf{K}}}\delta\mathbf{K}. \tag{15}$$

### A.2. Isolated Value-Pruning Score

In isolated value pruning, the key-cache matrix is not perturbed and is considered a constant not affecting $\mathcal{L}$. Therefore, minimizing the pruning-induced eviction error reduces to minimizing Equation 13. To derive saliency scores in closed form, we begin by deriving expressions for the gradient and Hessian with respect to $\widehat{\mathbf{V}}_{:,j}$. The first-order derivative of $\boldsymbol{E}_{i,j}$ with respect to $\widehat{\mathbf{V}}_{:,j}$ is:

$$\frac{\partial \boldsymbol{E}_{i,j}}{\partial \widehat{\mathbf{V}}_{:,j}} = \frac{\partial}{\partial \widehat{\mathbf{V}}_{:,j}} |\widehat{\mathbf{O}}_{i,j} - \mathbf{O}_{i,j}|^2 = 2(\widehat{\mathbf{O}}_{i,j} - \mathbf{O}_{i,j})\frac{\partial}{\partial \widehat{\mathbf{V}}_{:,j}}\hat{\mathbf{a}}_i \widehat{\mathbf{V}}_{:,j} = 2(\widehat{\mathbf{O}}_{i,j} - \mathbf{O}_{i,j})\hat{\mathbf{a}}_i. \tag{16}$$

When evaluated at $(\mathbf{V}_{:,j}, \mathbf{K})$, we again have $\widehat{\mathbf{O}}_{i,j} - \mathbf{O}_{i,j} = 0$, so the gradient term vanishes to zero. The Hessian of $\mathcal{L}$ with respect to $\widehat{\mathbf{V}}_{:,j}$ evaluated at $(\mathbf{V}_{:,j}, \mathbf{K})$ is:

$$\frac{\partial^2 \boldsymbol{E}_{i,j}(\mathbf{V}_{:,j}, \mathbf{K})}{\partial \widehat{\mathbf{V}}_{:,j}^2} = 2\frac{\partial}{\partial \widehat{\mathbf{O}}_{i,j}}\left((\widehat{\mathbf{O}}_{i,j} - \mathbf{O}_{i,j})\hat{\mathbf{a}}_i^\top\right)\left(\frac{\partial \widehat{\mathbf{O}}_{i,j}}{\partial \widehat{\mathbf{V}}_{:,j}}\right)^\top\bigg|_{(\mathbf{V}_{:,j}, \mathbf{K})} = 2\mathbf{a}_i^\top \mathbf{a}_i. \tag{17}$$

Substituting the gradient and Hessian back into Equation 13, we get the pruning-induced eviction error when value states are considered as isolated pruning units:

$$\mathcal{L}^{\text{value}} = \sum_{i=w}^{s}\sum_{j=1}^{d} \delta\mathbf{V}_{:,j}^\top (\mathbf{a}_i^\top \mathbf{a}_i)\delta\mathbf{V}_{:,j} = \sum_{i=w}^{s}\sum_{j=1}^{d} |\mathbf{a}_i \delta\mathbf{V}_{:,j}|^2. \tag{18}$$

Next, the row-wise pruning constraint $\boldsymbol{e}_p^\top \widehat{\mathbf{V}} = \mathbf{0}$ implies that when pruning the $p$-th token position, the value cache perturbation $\delta\mathbf{V}_{:,j}$ is explicitly in the form:

$$\delta\mathbf{V}_{t,j} = \begin{cases} -\mathbf{V}_{p,j}, & \text{when } t = p \\ 0, & \text{when } t \neq p \end{cases}, \quad \forall j = 1, ..., d.$$

Substituting this value perturbation into $\mathcal{L}^{\text{value}}$, we obtain the output perturbation when pruning the $p$-th value vector from the value cache $\mathbf{V}$:

$$\boldsymbol{S}_p^{\text{value}} = \sum_{i=w}^{s}\sum_{j=1}^{d} |\mathbf{A}_{i,p} \cdot \mathbf{V}_{p,j}|^2 = \sum_{i=w}^{s} |\mathbf{A}_{i,p}|^2 \cdot \sum_{j=1}^{d} |\mathbf{V}_{p,j}|^2 = \boxed{\sum_{i=w}^{s} |\mathbf{A}_{i,p}|^2 \cdot \|\mathbf{v}_p\|^2}. \tag{19}$$

This is also the value-pruning saliency score for evicting the $p$-th value vector $\mathbf{v}_p$ from the value cache. It is consistent with the expression in Equation 4 in Section 4.2.

### A.3. Isolated Key-Pruning Score

In isolated key pruning, the value-cache matrix is not perturbed and is assumed a constant. Therefore, minimizing the pruning-induced eviction error reduces to minimizing Equation 14. As with the value-pruning score, we begin by deriving closed-form expressions for the gradient and Hessian with respect to $\widehat{\mathbf{K}}$. The first-order derivative of $\boldsymbol{E}_{i,j}$ with respect to $\widehat{\mathbf{K}}$ is:

$$\frac{\partial \boldsymbol{E}_{i,j}}{\partial \widehat{\mathbf{K}}} = \frac{\partial}{\partial \widehat{\mathbf{K}}} |\widehat{\mathbf{O}}_{i,j} - \mathbf{O}_{i,j}|^2 = 2(\widehat{\mathbf{O}}_{i,j} - \mathbf{O}_{i,j})\frac{\partial \widehat{\mathbf{O}}_{i,j}}{\partial \widehat{\mathbf{K}}}. \tag{20}$$

Before deriving the first-order term explicitly, note that when evaluated at the point $(\mathbf{V}_{:,j}, \mathbf{K})$, we again have $\widehat{\mathbf{O}}_{i,j} - \mathbf{O}_{i,j} = 0$. Consequently, the first-order term is eliminated, a result that is the same as in isolated value pruning. We now explicitly

derive the term $\widehat{\boldsymbol{M}}^{(i,j)} := \frac{\partial \widehat{\mathbf{O}}_{i,j}}{\partial \widehat{\mathbf{K}}}$ above:

$$
\begin{aligned}
\widehat{\boldsymbol{M}}_{p,r}^{(i,j)} &= \left(\frac{\partial \widehat{\mathbf{O}}_{i,j}}{\partial \widehat{\mathbf{K}}}\right)_{p,r} = \frac{\partial}{\partial \widehat{\mathbf{K}}_{p,r}} \sum_{m=1}^{s} \widehat{\mathbf{A}}_{i,m} \cdot \widehat{\mathbf{V}}_{m,j} = \sum_{m=1}^{s} \frac{\partial \widehat{\mathbf{O}}_{i,j}}{\partial \widehat{\mathbf{A}}_{i,m}} \frac{\partial \widehat{\mathbf{A}}_{i,m}}{\partial \widehat{\mathbf{K}}_{p,r}} \\
&= \sum_{m=1}^{s} \frac{\partial \widehat{\mathbf{O}}_{i,j}}{\partial \widehat{\mathbf{A}}_{i,m}} \frac{\partial}{\partial \widehat{\mathbf{K}}_{p,r}} \left(\frac{e^{\widehat{\mathbf{Z}}_{i,m}}}{\sum_{u=1}^{s} e^{\widehat{\mathbf{Z}}_{i,u}}}\right) \\
&= \sum_{m=1}^{s} \frac{\partial \widehat{\mathbf{O}}_{i,j}}{\partial \widehat{\mathbf{A}}_{i,m}} \sum_{u=1}^{s} \frac{\widehat{\mathbf{A}}_{i,m}}{\widehat{\mathbf{Z}}_{i,u}} \frac{\widehat{\mathbf{Z}}_{i,u}}{\partial \widehat{\mathbf{K}}_{p,r}} \\
&= \sum_{m=1}^{s} \widehat{\mathbf{V}}_{m,j} \sum_{u=1}^{s} \widehat{\mathbf{A}}_{i,m}(\delta_{mu} - \widehat{\mathbf{A}}_{i,u}) \frac{1}{\sqrt{d}} \mathbf{Q}_{i,r} \delta_{up} \\
&= \sum_{m=1}^{s} \widehat{\mathbf{V}}_{m,j} \cdot \widehat{\mathbf{A}}_{i,m}(\delta_{mp} - \widehat{\mathbf{A}}_{i,p}) \frac{1}{\sqrt{d}} \mathbf{Q}_{i,r} \quad (\delta_{up} = 0 \text{ when } u \neq p) \\
&= \frac{1}{\sqrt{d}} \mathbf{Q}_{i,r} \cdot \left(\sum_{m=1}^{s} \widehat{\mathbf{V}}_{m,j} \cdot \widehat{\mathbf{A}}_{i,m} \cdot \delta_{mp} - \sum_{m=1}^{s} \widehat{\mathbf{V}}_{m,j} \cdot \widehat{\mathbf{A}}_{i,m} \cdot \widehat{\mathbf{A}}_{i,p}\right) \\
&= \frac{1}{\sqrt{d}} \mathbf{Q}_{i,r} \cdot \left(\widehat{\mathbf{V}}_{p,j} \cdot \widehat{\mathbf{A}}_{i,p} - \widehat{\mathbf{A}}_{i,p} \cdot \sum_{m=1}^{s} \widehat{\mathbf{V}}_{m,j} \cdot \widehat{\mathbf{A}}_{i,m}\right) \quad (\delta_{mp} = 0 \text{ when } m \neq p) \\
&= \frac{1}{\sqrt{d}} \mathbf{Q}_{i,r} \cdot \widehat{\mathbf{A}}_{i,p} \cdot (\widehat{\mathbf{V}}_{p,j} - \widehat{\mathbf{O}}_{i,j}).
\end{aligned}
\tag{21}
$$

Therefore, the first-order derivative of $\boldsymbol{E}_{i,j}$ with respect to the perturbed key cache $\widehat{\mathbf{K}}$ is:

$$
\frac{\partial \boldsymbol{E}_{i,j}}{\partial \widehat{\mathbf{K}}} = 2(\widehat{\mathbf{O}}_{i,j} - \mathbf{O}_{i,j})\widehat{\boldsymbol{M}}^{(i,j)}, \text{ where } \widehat{\boldsymbol{M}}_{p,r}^{(i,j)} = \frac{1}{\sqrt{d}} \mathbf{Q}_{i,r} \cdot \widehat{\mathbf{A}}_{i,p} \cdot (\widehat{\mathbf{V}}_{p,j} - \widehat{\mathbf{O}}_{i,j}).
$$

We now proceed to derive the Hessian, which is a 4-th order tensor of size $s \times d \times s \times d$. The second-order derivative of the element-wise perturbation $\boldsymbol{E}_{i,j}$ with respect to $\widehat{\mathbf{K}}$ is:

$$
\begin{aligned}
\frac{\partial^2 \boldsymbol{E}_{i,j}}{\partial \widehat{\mathbf{K}}^2} &= \widehat{\boldsymbol{M}}^{(i,j)} \otimes \left(\frac{\partial}{\partial \widehat{\mathbf{K}}} 2(\widehat{\mathbf{O}}_{i,j} - \mathbf{O}_{i,j})\right) + 2(\widehat{\mathbf{O}}_{i,j} - \mathbf{O}_{i,j}) \frac{\partial \widehat{\boldsymbol{M}}^{(i,j)}}{\partial \widehat{\mathbf{K}}} \\
&= 2\widehat{\boldsymbol{M}}^{(i,j)} \otimes \widehat{\boldsymbol{M}}^{(i,j)} + 2(\widehat{\mathbf{O}}_{i,j} - \mathbf{O}_{i,j}) \frac{\partial \widehat{\boldsymbol{M}}^{(i,j)}}{\partial \widehat{\mathbf{K}}},
\end{aligned}
\tag{22}
$$

where $\otimes$ denotes the matrix-wise outer product[1]. Since the Hessian will be evaluated at the point $(\mathbf{V}_{:,j}, \mathbf{K})$, where the perturbed attention output again matches the original, i.e., $\widehat{\mathbf{O}}_{i,j} - \mathbf{O}_{i,j} = 0$, the second term in Equation 22 vanishes. Therefore, the Hessian is:

$$
\left. \frac{\partial^2 \boldsymbol{E}_{i,j}}{\partial \widehat{\mathbf{K}}^2} \right|_{(\mathbf{V}_{:,j}, \mathbf{K})} = 2\boldsymbol{M}^{(i,j)} \otimes \boldsymbol{M}^{(i,j)} = 2\text{vec}(\boldsymbol{M}^{(i,j)})^\top \text{vec}(\boldsymbol{M}^{(i,j)}),
\tag{23}
$$

$$
\text{where } \boldsymbol{M}_{p,r}^{(i,j)} = \frac{1}{\sqrt{d}} \mathbf{Q}_{i,r} \cdot \mathbf{A}_{i,p} \cdot (\mathbf{V}_{p,j} - \mathbf{O}_{i,j}).
$$

Here, we use $\text{vec}(\cdot)$ to denote the vectorization of a matrix. Substituting the gradient and Hessian back into Equation 14, we get the pruning-induced eviction error when key states are considered as isolated pruning units:

$$
\begin{aligned}
\mathcal{L}^{\text{key}} &= \sum_{i=w}^{s} \sum_{j=1}^{d} \text{vec}(\delta\mathbf{K})^\top \text{vec}(\boldsymbol{M}^{(i,j)})^\top \text{vec}(\boldsymbol{M}^{(i,j)}) \text{vec}(\delta\mathbf{K}) \\
&= \sum_{i=w}^{s} \sum_{j=1}^{d} |\text{vec}(\boldsymbol{M}^{(i,j)}) \text{vec}(\delta\mathbf{K})|^2.
\end{aligned}
\tag{24}
$$

---

[1]For any two matrices $\boldsymbol{C}, \boldsymbol{D} \in \mathbb{R}^{s \times d}$, the output tensor via outer product is $\boldsymbol{C} \otimes \boldsymbol{D} \in \mathbb{R}^{s \times d \times s \times d}$.

Similarly, the row-wise pruning constraint in $e_p^\top \widehat{\mathbf{K}} = \mathbf{0}$ implies that when pruning the $p$-th token position, the key cache perturbation $\delta \mathbf{K}$ is explicitly in the form:

$$\delta \mathbf{K}_{t,j} = \begin{cases} -\mathbf{K}_{p,j}, & \text{when } t = p \\ 0, & \text{when } t \neq p \end{cases}, \quad \forall\, j = 1, ..., d.$$

Substituting this key perturbation into $\mathcal{L}^{\text{key}}$, we obtain the output perturbation when pruning the $p$-th key vector from the key cache $\mathbf{K}$ (we let $\mathbf{m}_p^{(i,j)}$ denote the $p$-th row vector of $M^{(i,j)}$):

$$
\begin{aligned}
\mathbf{S}_p^{\text{key}} = \sum_{i=w}^{s} \sum_{j=1}^{d} |\mathbf{m}_p^{(i,j)}\, \mathbf{k}_p|^2 &= \sum_{i=w}^{s} \sum_{j=1}^{d} \Big| \sum_{r=1}^{d} \frac{1}{\sqrt{d}} \mathbf{Q}_{i,r} \cdot \mathbf{A}_{i,p}(\mathbf{V}_{p,j} - \mathbf{O}_{i,j}) \cdot \mathbf{K}_{p,r} \Big|^2 \\
&= \frac{1}{d} \sum_{i=w}^{s} \sum_{j=1}^{d} |\mathbf{A}_{i,p}(\mathbf{V}_{p,j} - \mathbf{O}_{i,j}) \cdot \sum_{r=1}^{d} \mathbf{Q}_{i,r} \mathbf{K}_{p,r}|^2 \\
&= \frac{1}{d} \sum_{i=w}^{s} \sum_{j=1}^{d} |\mathbf{A}_{i,p}(\mathbf{V}_{p,j} - \mathbf{O}_{i,j}) \cdot \mathbf{q}_i \mathbf{k}_p^\top|^2 \\
&= \frac{1}{d} \sum_{i=w}^{s} \left( |\mathbf{A}_{i,p}|^2 \cdot |\mathbf{q}_i \mathbf{k}_p^\top|^2 \sum_{j=1}^{d} |\mathbf{V}_{p,j} - \mathbf{O}_{i,j}|^2 \right) \\
&= \boxed{\sum_{i=w}^{s} |\mathbf{A}_{i,p}|^2 \cdot |\mathbf{Z}_{i,p}|^2 \cdot \|\mathbf{v}_p - \mathbf{o}_i\|_2^2}.
\end{aligned}
$$

This is also the key-pruning saliency score for evicting the $p$-th key vector $\mathbf{k}_p$ from the key cache. It is consistent with the expression in Equation 5 in Section 4.2.

### A.4. Joint-Pruning Score

When both $\mathbf{V}_{:,j}$ and $\mathbf{K}$ are treated as variables affecting $\mathcal{L}$, we need to compute the cross-term in the error function, which corresponds to deriving a closed-form expression for Equation 15. Given that the first-order derivative of $\mathbf{E}_{i,j}$ with respect to $\widehat{\mathbf{V}}_{:,j}$ is known, we can compute the cross-term as follows:

$$
\begin{aligned}
\frac{\partial^2 \mathbf{E}_{i,j}}{\partial \widehat{\mathbf{V}}_{:,j} \partial \widehat{\mathbf{K}}} &= \frac{\partial}{\partial \widehat{\mathbf{K}}} 2(\widehat{\mathbf{O}}_{i,j} - \mathbf{O}_{i,j})\hat{\mathbf{a}}_i \\
&= \hat{\mathbf{a}}_i \otimes \left( \frac{\partial}{\partial \widehat{\mathbf{K}}} 2(\widehat{\mathbf{O}}_{i,j} - \mathbf{O}_{i,j}) \right) + 2(\widehat{\mathbf{O}}_{i,j} - \mathbf{O}_{i,j})\frac{\partial \hat{\mathbf{a}}_i}{\partial \widehat{\mathbf{K}}}.
\end{aligned}
\tag{25}
$$

As in the derivation of key-pruning scores, the second term above will be zero when evaluated at the point $(\mathbf{V}_{:,j}, \mathbf{K})$. Therefore, the cross second-order term is:

$$
\left. \frac{\partial^2 \mathbf{E}_{i,j}}{\partial \widehat{\mathbf{V}}_{:,j} \partial \widehat{\mathbf{K}}} \right|_{(\mathbf{V}_{:,j}, \mathbf{K})} = 2\mathbf{a}_i \otimes M^{(i,j)}, \quad \text{where } M_{p,r}^{(i,j)} = \frac{1}{\sqrt{d}} \mathbf{Q}_{i,r} \cdot \mathbf{A}_{i,p} \cdot (\mathbf{V}_{p,j} - \mathbf{O}_{i,j}).
\tag{26}
$$

Substituting this back into Equation 15, we get the cross-term of the pruning-induced eviction error when the value and key states are considered as combined pruning units:

$$
\mathcal{L}^{\text{cross}} = \sum_{i=w}^{s} \sum_{j=1}^{d} \delta \mathbf{V}_{:,j}^\top \big(2\mathbf{a}_i \otimes M^{(i,j)}\big) \delta \mathbf{K} = 2 \sum_{i=w}^{s} \sum_{j=1}^{d} (\delta \mathbf{V}_{:,j}^\top \mathbf{a}_i) \cdot \langle M^{(i,j)}, \delta \mathbf{K} \rangle_F,
\tag{27}
$$

where $\langle M^{(i,j)}, \delta \mathbf{K} \rangle_F = \sum_{p,r} M_{p,r}^{(i,j)} \cdot \delta \mathbf{K}_{p,r} = \mathbf{tr}(M^{(i,j)\top} \delta \mathbf{K})$ is the Frobenius inner product.

When the $p$-th key and value are pruned, the row-wise pruning constraint in $e_p^\top [\widehat{\mathbf{V}} \ \widehat{\mathbf{K}}] = \mathbf{0}$ implies that the value cache perturbation $\delta \mathbf{V}$ and the key cache perturbation $\delta \mathbf{K}$ are explicitly in the form:

$$\delta \mathbf{V}_{t,j} = \begin{cases} -\mathbf{V}_{p,j}, & \text{when } t = p \\ 0, & \text{when } t \neq p \end{cases}, \quad \delta \mathbf{K}_{t,j} = \begin{cases} -\mathbf{K}_{p,j}, & \text{when } t = p \\ 0, & \text{when } t \neq p \end{cases}, \quad \forall\, j = 1, ..., d.$$

Thus, we have each term in $\mathcal{L}^{\text{cross}}$:

$$\delta \mathbf{V}_{:,j}^\top \mathbf{a}_i = -\mathbf{A}_{i,p} \mathbf{V}_{p,j}, \quad \langle \mathbf{M}^{(i,j)}, \delta \mathbf{K} \rangle_F = -\frac{1}{\sqrt{d}} \mathbf{A}_{i,p} \cdot (\mathbf{V}_{p,j} - \mathbf{O}_{i,j}) \cdot (\mathbf{q}_i \mathbf{k}_p^\top).$$

Substituting these two expressions back into Equation 27, we obtain the cross term in closed form:

$$\begin{aligned}
\mathcal{L}^{\text{cross}} &= \frac{2}{\sqrt{d}} \sum_{i=w}^{s} \sum_{j=1}^{d} \mathbf{A}_{i,p}^2 \cdot \mathbf{V}_{p,j} \cdot (\mathbf{V}_{p,j} - \mathbf{O}_{i,j}) \cdot (\mathbf{q}_i \mathbf{k}_p^\top) \\
&= \frac{2}{\sqrt{d}} \sum_{i=w}^{s} \mathbf{A}_{i,p}^2 \cdot (\mathbf{q}_i \mathbf{k}_p^\top) \sum_{j=1}^{d} \mathbf{V}_{p,j} (\mathbf{V}_{p,j} - \mathbf{O}_{i,j}) \\
&= \boxed{2 \sum_{i=w}^{s} \mathbf{A}_{i,p}^2 \cdot \mathbf{Z}_{i,p} \cdot (\|\mathbf{v}_p\|_2^2 - \mathbf{v}_p^\top \mathbf{o}_i)}.
\end{aligned} \tag{28}$$

Finally, the saliency score of the token at position $p$ when both value and key states are treated as pruning units is the summation of $\mathcal{L}^{\text{value}}$, $\mathcal{L}^{\text{key}}$ and $\mathcal{L}^{\text{cross}}$:

$$\boxed{\mathbf{S}_p^{\text{joint}} = 2 \sum_{i=w}^{s} \mathbf{A}_{i,p}^2 \cdot \mathbf{Z}_{i,p} \cdot (\|\mathbf{v}_p\|_2^2 - \mathbf{v}_p^\top \mathbf{o}_i) + \mathbf{S}_p^{\text{value}} + \mathbf{S}_p^{\text{key}}}. \tag{29}$$

This is consistent with the joint score expression in Equation 6 of Section 4.2.

## A.5. Scores for Grouped-Query Attention

In models with Grouped-Query Attention (GQA) (Ainslie et al., 2023), multiple query heads share a single key and value head, reducing KV cache storage at the architectural level. To support cache eviction under this setting, we explicitly incorporate the head-group structure into our analysis and derive the corresponding OBCACHE scores.

We use superscripts $g$ to denote the KV-head index, $h$ to denote query-head index, and $H(g)$ to denote the set of query heads associated with KV head $g$. Our previous derivation omits head indices for simplicity. Here, we specifically include head indexing for multi-head attention (MHA). If the pruning-induced eviction error is defined on the per-head attention output,

$$\mathcal{L} := \|\widehat{\mathbf{O}}^h - \mathbf{O}^h\|_F^2, \text{ where } \mathbf{O}^h, \widehat{\mathbf{O}}^h \in \mathbb{R}^{s \times d},$$

the OBCACHE scores for the $h$-th head naturally extend to:

$$\mathbf{S}_{p,h}^{\text{value}} = \sum_i |\mathbf{A}_{i,p}^h|^2 \cdot \|\mathbf{v}_p^h\|^2 \tag{30}$$

$$\mathbf{S}_{p,h}^{\text{key}} = \sum_i |\mathbf{A}_{i,p}^h|^2 \cdot |\mathbf{Z}_{i,p}^h|^2 \cdot \|\mathbf{v}_p^h - \mathbf{o}_i^h\|_2^2 \tag{31}$$

$$\mathbf{S}_{p,h}^{\text{joint}} = 2 \sum_i |\mathbf{A}_{i,p}^h|^2 \cdot \mathbf{Z}_{i,p}^h \cdot (\|\mathbf{v}_p^h\|_2^2 - \mathbf{v}_p^{h\top} \mathbf{o}_i^h) + \mathbf{S}_{p,h}^{\text{value}} + \mathbf{S}_{p,h}^{\text{key}} \tag{32}$$

In standard MHA, each output head $\mathbf{O}^h$ depends on its own $\mathbf{K}^h$ and $\mathbf{V}^h$. In GQA, however, a group of output heads $\{\mathbf{O}^h\}_{h \in H(g)}$ shares the same key and value $\mathbf{K}^g$ and $\mathbf{V}^g$, i.e.,

$$\mathbf{V}^g \equiv \mathbf{V}^h, \forall\, h \in H(g), \quad \mathbf{K}^g \equiv \mathbf{K}^h, \forall\, h \in H(g)$$

If we now modify the objective to be the sum of output perturbations across heads within a group:

$$\mathcal{L} := \sum_{h \in H(g)} \|\widehat{\mathbf{O}}^h - \mathbf{O}^h\|_F^2,$$

the OBCACHE scores for the $g$-th KV head introduce an additional summation over its associated query heads:

$$\boldsymbol{S}_{p,g}^{\text{value}} = \sum_{h \in H(g)} \boldsymbol{S}_{p,h}^{\text{value}} = \sum_{h \in H(g)} \sum_i |\mathbf{A}_{i,p}^h|^2 \cdot \|\mathbf{v}_p^g\|^2 \tag{33}$$

$$\boldsymbol{S}_{p,g}^{\text{key}} = \sum_{h \in H(g)} \boldsymbol{S}_{p,h}^{\text{key}} = \sum_{h \in H(g)} \sum_i |\mathbf{A}_{i,p}^h|^2 \cdot |\mathbf{Z}_{i,p}^h|^2 \cdot \|\mathbf{v}_p^g - \mathbf{o}_i^h\|_2^2 \tag{34}$$

$$\boldsymbol{S}_{p,g}^{\text{joint}} = \sum_{h \in H(g)} \boldsymbol{S}_{p,h}^{\text{joint}} = 2 \sum_{h \in H(g)} \left( \sum_i |\mathbf{A}_{i,p}^h|^2 \cdot \mathbf{Z}_{i,p}^h \cdot (\|\mathbf{v}_p^g\|_2^2 - \mathbf{v}_p^{g\top} \mathbf{o}_i^h) \right) + \boldsymbol{S}_{p,g}^{\text{value}} + \boldsymbol{S}_{p,g}^{\text{key}} \tag{35}$$

Note that objective formulations other than summation can also yield valid per-KV-head scores. In all experiments, we use the above GQA-aware formulation to gather per-query-head scores without retaining KV cache for all query heads, which creates redundant KV memories as implemented by SnapKV (Li et al., 2024).

## B. Limitations and Future Work

OBCACHE presents several opportunities for further improvement:

**Structural Bias to Earlier Tokens.** One limitation is the structural bias toward earlier tokens, a common drawback in many existing cache eviction methods that also emerges in OBCACHE, particularly as the perturbation window increases. Although heuristics such as retaining a fixed-size recent window help mitigate this bias by leveraging the locality of attention patterns, such strategies remain static and empirically motivated. Future work should investigate the design and adaptability of perturbation windows more systematically to further improve saliency estimation.

**Choice of Perturbation Window in Decoding.** Additionally, OBCACHE's current dynamic cache eviction strategy in the decoding phase directly follows $H_2O$ by accumulating saliency scores over time, effectively minimizing perturbations over the full sequence history. However, under our pruning-based formulation, alternative objectives could also be explored. For example, saliency scores could instead be accumulated only within a recent attention window. Such variants may yield improved adaptability in long-context generation.

**Extended Real-World Evaluation.** The evaluation settings considered in this work can also be extended to broader real-world scenarios. For example, OBCACHE could be further studied under dynamic cache eviction beyond language modeling tasks. In decoding-centric tasks such as mathematical reasoning, dynamic cache eviction may play a more important role than prefill-stage eviction. In addition, OBCACHE has not yet been evaluated on multi-round agentic tasks, which may more directly reflect real-world deployment settings.

**Extension to other KV Tasks.** On the theoretical side, OBCACHE introduces a flexible perturbation-minimization framework that can potentially be extended to a broader class of KV cache compression strategies. For instance, due to the flexibility of structured pruning, the OBCACHE formulation could be adapted to key-channel pruning by redefining the pruning units from tokens to channels. By leveraging suitable approximation techniques, one could derive corresponding closed-form saliency scores from an output-aware perspective, enabling informed channel-level pruning decisions. Moreover, our approach relies on the diagonal Hessian assumption used in Optimal Brain Damage, which naturally supports token removal. In contrast, the classical Optimal Brain Surgeon framework relaxes this assumption by allowing compensation across pruning units, adjusting the remaining parameters to reduce loss. This perspective aligns naturally with recent cache merging methods and may inspire more theoretically grounded cache management techniques for long-context LLMs.

Overall, OBCACHE provides a flexible foundation for future research on KV cache compression. Its structured and theoretically motivated formulation opens promising directions toward more principled KV cache management.

# C. Experimental Setup

All experiments are conducted on NVIDIA A100-80GB GPUs. We adapt the KVPress library[2] (Devoto et al., 2025) with PyTorch (Paszke et al., 2019) to implement the cache eviction algorithms. We use two representative instruction-tuned large language models as backbones: LLaMA-3.1-8B-Instruct (Grattafiori et al., 2024) and Qwen-2.5-7B-Instruct (Bai et al., 2023). Both models employ Grouped Query Attention (GQA) and natively support context windows of up to 128K tokens. All model inference is performed in half precision (bfloat16) without quantization.

For tasks with long prefill prompts, we use FlashAttention-2 (Dao, 2024) for prefill attention computation to reduce memory overhead. Since FlashAttention does not materialize intermediate attention scores, we follow the standard workaround (Li et al., 2024; Cai et al., 2025; Feng et al., 2026) and recompute attention weights for selected query positions when needed to compute saliency scores for token eviction.

## C.1. Datasets

We evaluate OBCACHE and existing cache eviction methods on three benchmarks: RULER (Hsieh et al., 2024), LongBench (Bai et al., 2024), and the PG19 dataset (Rae et al., 2019), targeting prefill-stage static cache eviction and decoding-stage dynamic cache eviction, respectively.

**RULER.** RULER is a synthetic long-context benchmark that expands upon the vanilla Needle-In-A-Haystack (Kamradt, 2023) test and comprises more comprehensive and challenging tasks. It includes 13 synthetic tasks across four categories: retrieval, multi-hop tracing, aggregation, and question answering. We follow the setup from the official repository[3] using the default task configurations. For both LLaMA-3.1 and Qwen-2.5, context lengths of 4K and 32K tokens are evaluated. Each context-length setting contains 1,625 randomly generated samples, with 125 samples for each of the 13 synthetic tasks.

**LongBench.** LongBench includes 16 datasets across six task categories: single-document QA, multi-document QA, summarization, few-shot learning, synthetic reasoning, and code completion. The average input length is 6,711 words, which is approximately 16K tokens. We do not perform input truncation during our experiments. Following SnapKV (Li et al., 2024) and AdaKV (Feng et al., 2026), cache eviction occurs only during the prefill phase. We report task-specific evaluation metrics (e.g., Exact Match/F1 for QA tasks, ROUGE for summarization tasks) as the scores. Following LongBench's official evaluation script[4], we do not apply a chat template for TREC, TriviaQA, SAMSum, Lcc, and RepoBench-P tasks.

**Language Modeling.** For decoding-phase cache eviction, we adopt the PG19 test set following the setup in StreamingLLM (Xiao et al., 2024). PG19 contains 100 full-length books, each averaging around 70K tokens. We evaluate on the standard test split[5] and compute perplexity across varying context lengths (from 1 to 32K), using a fixed KV cache budget of 1024 tokens for all dynamic cache eviction methods.

## C.2. Baseline Setups

### C.2.1. PREFILL EVICTION

To evaluate baselines for prefill-stage cache eviction, we follow the experimental setup of SnapKV (Li et al., 2024) and perform cache eviction only during the prefill phase. In these benchmarks, the prompt length dominates memory usage and serves as the primary bottleneck; therefore, applying decoding-stage eviction has limited impact on task performance. In the query-aware compression setting, the model has access to the task-specific downstream query when evicting tokens. In the query-agnostic setting, the model performs eviction based only on the context KV cache without access to the downstream query (e.g., a question about a document). This setting mimics a more challenging compression scenario under real-world cache reuse settings.

**$H_2O$.** $H_2O$ originally accumulates attention weights across all historical query positions to make eviction decisions. This is incompatible with FlashAttention-based prefill implementations, as it requires recomputing full attention matrices. To enable efficient implementation, we follow SnapKV's implementation of $H_2O$ and accumulate query positions only within a

---

[2] https://github.com/NVIDIA/kvpress
[3] https://github.com/NVIDIA/RULER
[4] https://github.com/THUDM/LongBench/blob/main/LongBench/pred.py
[5] https://huggingface.co/datasets/emozilla/pg19

*Table 2.* Complexity comparison of OBCACHE scores with purely attention-based scores for cache eviction. We define $W := s - w$ as the size of the perturbation window and $d_{\text{head}}$ as the attention head hidden dimension.

| Method | Saliency Score $\boldsymbol{S}_p$ | Complexity |
|---|:---:|:---:|
| Attention-based (Zhang et al., 2023; Oren et al., 2024; Li et al., 2024) | $\sum_i \lvert \mathbf{A}_{i,p} \rvert$ | $O(W)$ |
| OBCACHE-V | $\sum_i \lvert \mathbf{A}_{i,p} \rvert^2 \cdot \lVert \mathbf{v}_p \rVert^2$ | $O(W + d_{\text{head}})$ |
| OBCACHE-K | $\sum_i \lvert \mathbf{A}_{i,p} \rvert^2 \cdot \lvert \mathbf{Z}_{i,p} \rvert^2 \cdot \lVert \mathbf{v}_p - \mathbf{o}_i \rVert_2^2$ | $O(W d_{\text{head}})$ |
| OBCACHE-V&K | $2\sum_i \lvert \mathbf{A}_{i,p} \rvert^2 \cdot \mathbf{Z}_{i,p} \cdot (\lVert \mathbf{v}_p \rVert_2^2 - \mathbf{v}_p^\top \mathbf{o}_i) + \boldsymbol{S}_p^{\text{value}} + \boldsymbol{S}_p^{\text{key}}$ | $O(W d_{\text{head}})$ |

recent perturbation window. For all prefill-stage eviction tasks, the perturbation window size is set to 64. All tokens within the perturbation window are treated as fixed recent tokens, while the remaining cache budget is allocated to heavy hitters.

**TOVA.** TOVA does not accumulate attention weights and instead makes eviction decisions solely based on the most recent attention distribution. In all prefill-stage eviction evaluations, the entire cache budget is allocated to heavy hitters. In its original implementation, TOVA averages scores across all attention heads and evicts the same token positions for all heads. For fair comparison with other baselines, our implementation instead performs per-head eviction, consistent with the other baselines.

**SnapKV.** In SnapKV, the same 64-token perturbation window as in $H_2O$ is used. The key difference is that SnapKV additionally applies a one-dimensional pooling filter to smooth the accumulated attention scores before eviction. For all SnapKV and OBCACHE-enhanced experiments, we follow the default implementation in KVPress (Devoto et al., 2025), using an average-pooling function with a kernel size of 5.

**AdaKV.** AdaKV determines adaptive head-wise budgets by globally ranking the top-$k$ eviction scores across all attention heads. The eviction scores can originate from any prior scoring method. As the baseline configuration, we implement AdaKV using SnapKV scores (accumulated attention within a recent window together with pooling), which serves as our strongest baseline and the current state of the art. To integrate AdaKV with OBCACHE-enhanced counterparts, we replace the SnapKV scores in AdaKV with OBCACHE scores using the same 64-token perturbation window and average-pooling function. The safeguard hyperparameter in AdaKV is kept at its default value of 0.2.

**VATP and CriticalKV.** VATP differs from OBCACHE-V only in the choice of norm. We replace the $\ell_2$-norm in OBCACHE-V with the $\ell_1$-norm to reproduce VATP. CriticalKV adopts a two-stage eviction pipeline. In the first stage, half of the KV budget is selected using a prior attention-only score (SnapKV). In the second stage, the remaining budget is allocated using the value-norm-scaled score. We reproduce CriticalKV using the implementation of KVPress.

C.2.2. DECODING EVICTION

To evaluate baselines for decoding-stage cache eviction, we follow the setup of StreamingLLM (Xiao et al., 2024), where eviction decisions are made at every decoding step. All eviction methods are evaluated under a 1024-token cache budget.

**StreamingLLM.** For StreamingLLM, we retain the first 4 tokens as attention sinks and always preserve the most recent 1,020 tokens. Since attention sinks are essential for maintaining long-context generation quality, we also retain the first 4 sink tokens in all other methods.

**$H_2O$.** At each decoding step, $H_2O$ accumulates attention weights across all historical query positions to make eviction decisions. In all perplexity experiments involving $H_2O$ and its OBCACHE-enhanced variants, a 256-token recent window is always reserved. The remaining 764 tokens are dynamically selected using their respective saliency scores.

## D. Experimental Results

### D.1. Efficiency Evaluation of OBCACHE Scores

To analyze the additional computation overhead introduced by OBCACHE scores, we compare their complexity against purely attention-based scores (Zhang et al., 2023; Oren et al., 2024; Li et al., 2024), as summarized in Table 2. For

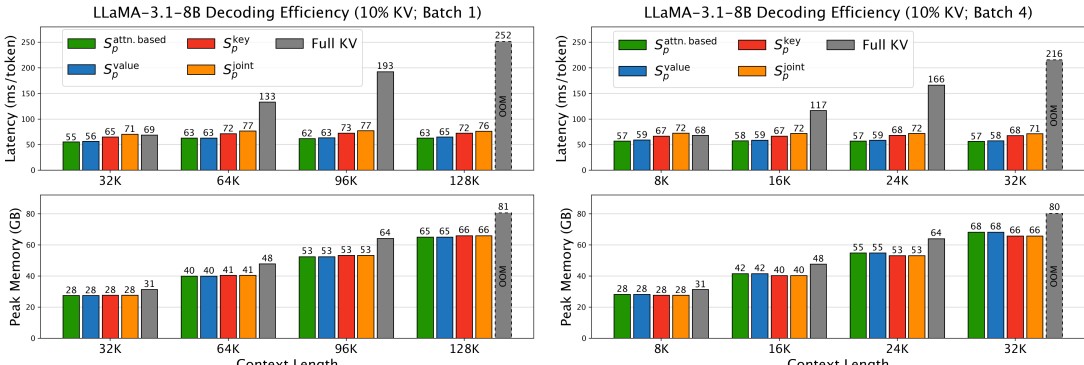

*Figure 6.* Decoding complexity of OBCACHE. We report the per-token decoding latency (averaged over 512 generated tokens) and peak memory consumption across different methods. Results are shown for varying context lengths, with batch size 1 on the left and batch size 4 on the right. Out-of-memory (OOM) results are linearly extrapolated from measured data.

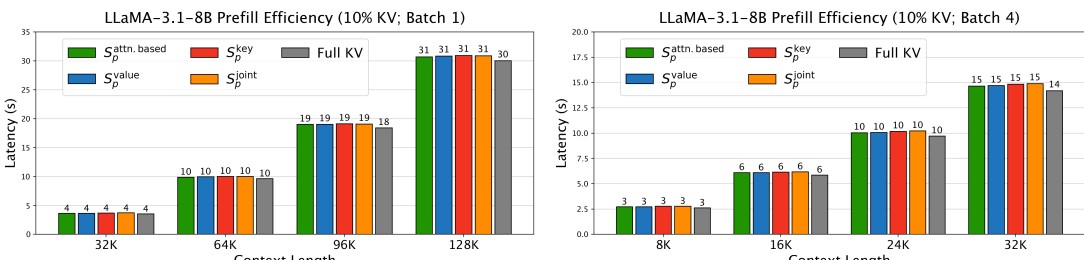

*Figure 7.* Prefill complexity of OBCACHE. We report the time to first token (TTFT) in seconds, including the cache eviction operation. Results are presented for different context lengths, with batch size 1 (left) and batch size 4 (right).

OBCACHE-V, the only additional operation beyond attention-based scores is computing the norm of the value state, which adds a negligible linear term in the per-head hidden dimension. OBCACHE-K and OBCACHE-V&K require computing the norm of the difference between the value vector and each attention output vector within the perturbation window, resulting in a complexity of $O(Wd_{\text{head}})$. This is more expensive than OBCACHE-V. However, in practical settings, the perturbation window $W$ is often very small (e.g., $W = 64$ in the prefill phase and $W = 1$ during decoding), so the additional overhead remains minor relative to the overall model computation.

We empirically benchmark peak memory usage and latency overhead during prefill and decoding with varying context lengths and batch sizes on a single A100-80GB GPU, as shown in Figure 6 and Figure 7. For each configuration, we measure: (1) the average time to first token (TTFT) in seconds, including the cache eviction operation, and (2) the per-token decoding latency where scoring and eviction are dynamically updated.

Across both single- and multi-batch settings, the results show that OBCACHE-V has nearly identical decoding latency (<2ms) to attention-based methods across all context lengths. While OBCACHE-K and OBCACHE-V&K introduce additional latency (<15ms), their cost remains substantially lower than full-cache decoding and does not scale linearly with context length, demonstrating practical efficiency. In the prefill phase, all eviction methods show negligible additional latency compared to the full-KV baseline. These empirical results align with the complexity analysis and demonstrate that OBCache scores can be efficiently computed. We also note that our implementation of score-based cache eviction is built on default PyTorch primitives (e.g., `torch.gather` and `torch.cat`) without customized kernels. As a result, at small context lengths, eviction methods exhibit higher latency than full-cache decoding. In the future, developing specialized kernels for score computation and cache eviction would yield additional speed improvements and further enhance practical efficiency.

### D.2. RULER and LongBench Full Results

Due to space constraints, we present the full tables of results for RULER-4K and RULER-32K (both query-aware and query-agnostic) in Tables 3, 4, 5, 6. The full tables of results for LongBench (both query-aware and query-agnostic) are illustrated in Tables 7 and 8.

# E. Implementation of OBCache

We provide a code implementation of OBCACHE in pseudo PyTorch style, as illustrated in Algorithms 1 and 2. These two algorithms demonstrate the computation of OBCACHE saliency scores and the cache eviction operation in the prefill phase.

---

**Algorithm 1** Implementation of OBCACHE score update in pseudo PyTorch style.

```
1   # key_states/value_states: cache matrix (bsz, num_heads, kv_len, head_dim);
2   # A/Z: attention weight/logit matrix (bsz, num_heads, q_len, kv_len);
3   # O: attention output matrix (bsz, num_heads, q_len, head_dim);
4   # w: perturbation window start index;
5
6   def obcache_score(key_states, value_states, A, Z, O, w):
7       # Target only the recent perturbation window
8       A, Z, O = A[..., -w:, :], Z[..., -w:, :], O[..., -w:, :]
9
10      # Compute accumulated attention-based score
11      S_attn_based = A.pow(2).sum(-2)
12      ### Existing cache eviction scores (e.g., H2O and TOVA) end here ###
13
14      # Compute value-pruning score
15      V_2norm = value_states.pow(2).sum(dim=-1)
16      S_value = S_attn * V_2norm
17
18      # Compute key-pruning score
19      O_2norm = O.pow(2).sum(dim=-1)
20      VO = torch.einsum('bhqd,bhpd->bhqp', O, V)
21      VmO_2norm = O_2norm.unsqueeze(-1) + V_2norm.unsqueeze(-2) - 2 * VO
22      S_key = ((A * Z).pow(2) * VmO_2norm).sum(dim=-2)
23
24      # Compute joint-pruning score
25      VVmO = V_2norm.unsqueeze(-2) - VO
26      S_joint = (2 * A.pow(2) * Z * VVmO).sum(dim=-2) + S_value + S_key
27
28    return S_attn_based, S_value, S_key, S_joint
```

---

**Algorithm 2** Implementation of OBCACHE cache eviction in pseudo PyTorch style.

```
1   # S: cache eviction saliency score matrix (bsz, num_heads, kv_len);
2   # num_hh / num_recent: cache budget allocated for recent and heavy-hitter tokens;
3   # num_coming: number of maximum tokens to come in next step;
4
5   def evict_kv(S, key_states, value_states, num_hh, num_recent, num_coming):
6       bsz, num_heads, kv_len, head_dim = value_states.shape
7       cutoff = kv_len - num_recent + num_coming
8
9       # Select most salient token positions
10      keep_topk_idx = torch.topk(S[..., :cutoff], num_hh, dim=-1)[1].sort().values
11      keep_topk_idx = keep_topk_idx.unsqueeze(-1).expand(-1, -1, -1, head_dim)
12
13      # Evict and keep recent
14      k_hh = torch.gather(key_states[..., :cutoff, :], dim=-2, index=keep_topk_idx)
15      k_compress = torch.cat([k_hh, key_states[..., cutoff:, :]], dim=-2)
16
17      v_hh = torch.gather(value_states[..., :cutoff, :], dim=-2, index=keep_topk_idx)
18      v_compress = torch.cat([v_hh, value_states[..., cutoff:, :]], dim=-2)
19
20    return k_compress, v_compress
```

*Table 3.* RULER-4K full results for LLaMA-3.1-8B-Instruct.

| | Methods | CWE | FWE | MK-1 | MK-2 | MK-3 | MQ | MV | S-1 | S-2 | S-3 | QA-1 | QA-2 | VT | Avg |
|---|---|---|---|---|---|---|---|---|---|---|---|---|---|---|---|
| | ALL KV | 99.6 | 85.07 | 100.0 | 100.0 | 100.0 | 100.0 | 100.0 | 100.0 | 100.0 | 100.0 | 88.0 | 65.6 | 100.0 | 95.25 |
| 10% KV (q-Aware) | H2O | 65.2 | 42.67 | 29.6 | 8.8 | 0.0 | 34.6 | 7.0 | 44.8 | 44.8 | 1.6 | 79.2 | 59.2 | 17.92 | 33.49 |
| | + OBCACHE-K | 69.44 | 41.33 | 75.2 | 29.6 | 0.0 | 51.4 | 9.4 | 68.8 | 81.6 | 1.6 | 83.2 | 60.8 | 48.32 | **47.75** |
| | TOVA | 37.92 | 41.87 | 100.0 | 83.2 | 0.0 | 78.4 | 82.8 | 100.0 | 85.6 | 0.0 | 90.4 | 62.4 | 40.16 | 61.75 |
| | + OBCACHE-K | 42.48 | 41.6 | 99.2 | 89.6 | 0.0 | 85.2 | 84.8 | 100.0 | 90.4 | 0.8 | 90.4 | 61.6 | 62.24 | **65.26** |
| | SnapKV | 40.96 | 51.47 | 96.0 | 44.8 | 0.0 | 92.2 | 24.2 | 98.4 | 91.2 | 1.6 | 84.0 | 60.8 | 47.2 | 56.37 |
| | + OBCACHE-K | 36.24 | 51.2 | 99.2 | 64.0 | 0.0 | 85.4 | 27.0 | 96.8 | 92.0 | 1.6 | 85.6 | 60.8 | 76.16 | **59.69** |
| | AdaKV | 61.04 | 54.13 | 90.4 | 62.4 | 5.6 | 81.0 | 15.0 | 100.0 | 89.6 | 1.6 | 84.8 | 60.0 | 60.64 | 58.94 |
| | + OBCACHE-K | 69.44 | 58.67 | 99.2 | 88.8 | 19.2 | 87.6 | 40.6 | 100.0 | 92.8 | 1.6 | 87.2 | 61.6 | 92.48 | **69.17** |
| 20% KV (q-Aware) | H2O | 86.32 | 48.8 | 85.6 | 28.0 | 16.8 | 74.6 | 17.2 | 75.2 | 79.2 | 1.6 | 88.0 | 63.2 | 40.16 | 54.21 |
| | + OBCACHE-K | 89.04 | 50.67 | 99.2 | 61.6 | 12.8 | 84.0 | 35.4 | 90.4 | 94.4 | 1.6 | 92.0 | 64.0 | 81.6 | **65.9** |
| | TOVA | 64.24 | 61.6 | 99.2 | 96.0 | 0.0 | 95.8 | 97.4 | 100.0 | 88.8 | 7.2 | 91.2 | 64.0 | 92.0 | 73.65 |
| | + OBCACHE-K | 68.48 | 64.27 | 100.0 | 96.0 | 0.0 | 94.6 | 97.2 | 99.2 | 92.0 | 12.0 | 89.6 | 64.0 | 97.28 | **74.97** |
| | SnapKV | 73.44 | 59.2 | 100.0 | 72.0 | 24.0 | 96.8 | 72.4 | 100.0 | 96.8 | 1.6 | 89.6 | 63.2 | 80.48 | 71.5 |
| | + OBCACHE-K | 73.44 | 61.87 | 100.0 | 89.6 | 20.0 | 91.8 | 78.4 | 100.0 | 96.0 | 1.6 | 89.6 | 64.0 | 96.16 | **74.04** |
| | AdaKV | 89.12 | 66.13 | 97.6 | 84.8 | 36.8 | 94.8 | 71.2 | 100.0 | 96.8 | 1.6 | 87.2 | 64.0 | 90.72 | 75.44 |
| | + OBCACHE-K | 92.56 | 65.07 | 100.0 | 99.2 | 73.6 | 97.0 | 86.2 | 100.0 | 100.0 | 1.6 | 92.0 | 67.2 | 99.52 | **82.61** |
| 30% KV (q-Aware) | H2O | 94.96 | 56.8 | 99.2 | 57.6 | 34.4 | 91.0 | 41.0 | 83.2 | 94.4 | 1.6 | 88.8 | 64.0 | 65.76 | 67.13 |
| | + OBCACHE-K | 96.16 | 60.0 | 100.0 | 88.0 | 33.6 | 94.2 | 67.8 | 96.0 | 96.8 | 1.6 | 89.6 | 64.8 | 91.84 | **75.42** |
| | TOVA | 78.72 | 72.0 | 100.0 | 96.8 | 1.6 | 97.0 | 98.8 | 100.0 | 92.0 | 33.6 | 88.8 | 64.0 | 98.56 | 78.61 |
| | + OBCACHE-K | 82.16 | 73.33 | 99.2 | 96.0 | 3.2 | 93.6 | 98.8 | 100.0 | 93.6 | 44.8 | 89.6 | 64.8 | 99.04 | **80.21** |
| | SnapKV | 87.12 | 64.8 | 98.4 | 87.2 | 47.2 | 97.8 | 90.2 | 100.0 | 99.2 | 1.6 | 92.0 | 65.6 | 92.8 | 78.76 |
| | + OBCACHE-K | 88.8 | 66.93 | 100.0 | 96.0 | 39.2 | 93.2 | 89.6 | 100.0 | 97.6 | 1.6 | 91.2 | 66.4 | 99.04 | **79.2** |
| | AdaKV | 95.92 | 68.8 | 99.2 | 96.0 | 68.8 | 99.4 | 89.4 | 100.0 | 99.2 | 1.6 | 89.6 | 66.4 | 97.12 | 82.42 |
| | + OBCACHE-K | 97.12 | 71.47 | 100.0 | 99.2 | 97.6 | 99.2 | 96.8 | 100.0 | 100.0 | 2.4 | 91.2 | 67.2 | 99.36 | **86.27** |
| 40% KV (q-Aware) | H2O | 98.16 | 61.87 | 100.0 | 69.6 | 57.6 | 95.8 | 66.0 | 96.8 | 97.6 | 1.6 | 89.6 | 64.0 | 79.68 | 75.25 |
| | + OBCACHE-K | 98.64 | 68.0 | 100.0 | 96.0 | 65.6 | 96.6 | 84.2 | 96.8 | 97.6 | 2.4 | 88.8 | 66.4 | 95.68 | **81.29** |
| | TOVA | 86.88 | 76.53 | 99.2 | 96.0 | 22.4 | 98.6 | 98.2 | 100.0 | 96.8 | 65.6 | 88.0 | 64.0 | 100.0 | 84.02 |
| | + OBCACHE-K | 92.16 | 77.33 | 99.2 | 97.6 | 27.2 | 98.0 | 98.8 | 100.0 | 96.8 | 73.6 | 87.2 | 66.4 | 100.0 | **85.71** |
| | SnapKV | 95.04 | 66.13 | 99.2 | 94.4 | 73.6 | 98.8 | 96.0 | 100.0 | 99.2 | 1.6 | 91.2 | 67.2 | 96.8 | **83.01** |
| | + OBCACHE-K | 94.72 | 67.47 | 99.2 | 99.2 | 66.4 | 97.2 | 96.2 | 100.0 | 98.4 | 1.6 | 92.0 | 66.4 | 99.36 | 82.93 |
| | AdaKV | 98.32 | 74.4 | 100.0 | 99.2 | 91.2 | 100.0 | 97.6 | 100.0 | 100.0 | 2.4 | 90.4 | 65.6 | 99.36 | 86.04 |
| | + OBCACHE-K | 98.4 | 76.0 | 100.0 | 100.0 | 100.0 | 100.0 | 99.0 | 100.0 | 100.0 | 18.4 | 89.6 | 66.4 | 100.0 | **88.29** |
| 10% KV (q-Agnostic) | H2O | 72.56 | 60.53 | 12.0 | 3.2 | 0.8 | 9.2 | 6.8 | 13.6 | 0.8 | 1.6 | 16.8 | 29.6 | 12.96 | 18.5 |
| | + OBCACHE-K | 73.76 | 64.53 | 15.2 | 8.8 | 0.8 | 10.4 | 7.8 | 36.8 | 4.0 | 1.6 | 16.8 | 33.6 | 28.8 | **23.3** |
| | TOVA | 66.72 | 6.4 | 24.0 | 0.8 | 0.0 | 4.4 | 1.6 | 39.2 | 0.8 | 0.0 | 14.4 | 30.4 | 24.96 | 16.44 |
| | + OBCACHE-K | 70.32 | 10.13 | 30.4 | 0.8 | 0.0 | 5.6 | 2.8 | 56.8 | 0.8 | 0.0 | 16.0 | 28.0 | 41.76 | **20.26** |
| | SnapKV | 10.24 | 69.6 | 19.2 | 6.4 | 1.6 | 15.0 | 10.0 | 44.8 | 6.4 | 1.6 | 14.4 | 31.2 | 21.44 | 19.38 |
| | + OBCACHE-K | 21.2 | 60.27 | 20.0 | 8.0 | 0.8 | 13.4 | 9.2 | 66.4 | 18.4 | 1.6 | 16.8 | 32.0 | 45.44 | **24.12** |
| | AdaKV | 36.8 | 72.53 | 12.8 | 8.0 | 1.6 | 11.0 | 9.0 | 45.6 | 4.0 | 1.6 | 16.0 | 32.8 | 28.0 | 21.52 |
| | + OBCACHE-K | 24.24 | 68.53 | 32.0 | 14.4 | 4.0 | 17.8 | 10.4 | 80.0 | 17.6 | 1.6 | 16.8 | 29.6 | 62.56 | **29.19** |
| 20% KV (q-Agnostic) | H2O | 89.76 | 69.6 | 22.4 | 10.4 | 1.6 | 11.2 | 8.8 | 32.8 | 4.8 | 1.6 | 32.8 | 44.0 | 28.8 | 27.58 |
| | + OBCACHE-K | 90.8 | 72.27 | 35.2 | 25.6 | 0.8 | 15.6 | 9.4 | 54.4 | 15.2 | 1.6 | 34.4 | 44.0 | 55.04 | **34.95** |
| | TOVA | 87.68 | 34.93 | 57.6 | 3.2 | 0.0 | 7.4 | 4.8 | 83.2 | 1.6 | 1.6 | 24.0 | 39.2 | 72.8 | 32.15 |
| | + OBCACHE-K | 90.48 | 44.53 | 75.2 | 5.6 | 0.0 | 13.6 | 6.4 | 93.6 | 1.6 | 1.6 | 25.6 | 44.0 | 77.6 | **36.91** |
| | SnapKV | 68.88 | 72.53 | 42.4 | 16.8 | 3.2 | 23.2 | 14.6 | 51.2 | 32.8 | 1.6 | 26.4 | 43.2 | 48.16 | 34.23 |
| | + OBCACHE-K | 76.4 | 69.33 | 49.6 | 15.2 | 4.0 | 22.8 | 11.4 | 69.6 | 42.4 | 1.6 | 26.4 | 42.4 | 75.2 | **38.95** |
| | AdaKV | 91.2 | 76.0 | 39.2 | 23.2 | 6.4 | 18.4 | 12.8 | 69.6 | 20.8 | 1.6 | 38.4 | 44.8 | 66.08 | 39.11 |
| | + OBCACHE-K | 91.68 | 73.07 | 64.8 | 48.8 | 27.2 | 31.8 | 14.2 | 96.0 | 51.2 | 1.6 | 38.4 | 45.6 | 91.36 | **51.98** |
| 30% KV (q-Agnostic) | H2O | 97.2 | 75.2 | 42.4 | 24.0 | 2.4 | 15.0 | 10.0 | 51.2 | 19.2 | 1.6 | 50.4 | 48.8 | 44.16 | 37.04 |
| | + OBCACHE-K | 97.2 | 76.8 | 50.4 | 47.2 | 2.4 | 21.4 | 11.6 | 64.8 | 38.4 | 1.6 | 47.2 | 56.8 | 65.44 | **44.71** |
| | TOVA | 95.36 | 54.67 | 80.8 | 9.6 | 0.0 | 16.8 | 7.4 | 97.6 | 2.4 | 1.6 | 39.2 | 46.4 | 95.68 | 42.12 |
| | + OBCACHE-K | 95.84 | 65.33 | 88.8 | 12.0 | 0.0 | 25.8 | 9.0 | 100.0 | 6.4 | 1.6 | 44.8 | 52.8 | 96.96 | **46.1** |
| | SnapKV | 88.16 | 77.07 | 60.0 | 28.8 | 4.8 | 31.0 | 16.0 | 67.2 | 49.6 | 1.6 | 40.8 | 48.8 | 64.32 | 44.47 |
| | + OBCACHE-K | 91.76 | 73.33 | 69.6 | 29.6 | 8.8 | 34.2 | 14.2 | 77.6 | 55.2 | 1.6 | 40.0 | 48.0 | 84.48 | **48.34** |
| | AdaKV | 97.76 | 79.47 | 61.6 | 47.2 | 18.4 | 28.4 | 15.8 | 78.4 | 40.0 | 1.6 | 51.2 | 53.6 | 83.68 | 50.55 |
| | + OBCACHE-K | 98.32 | 76.27 | 85.6 | 77.6 | 61.6 | 52.8 | 17.2 | 100.0 | 70.4 | 1.6 | 57.6 | 55.2 | 97.12 | **65.49** |
| 40% KV (q-Agnostic) | H2O | 99.36 | 78.67 | 56.0 | 35.2 | 4.8 | 21.4 | 12.0 | 60.8 | 31.2 | 1.6 | 55.2 | 53.6 | 58.4 | 43.71 |
| | + OBCACHE-K | 98.88 | 77.87 | 70.4 | 57.6 | 8.8 | 25.2 | 14.6 | 82.4 | 53.6 | 1.6 | 58.4 | 57.6 | 77.6 | **52.66** |
| | TOVA | 97.6 | 69.87 | 92.8 | 17.6 | 0.0 | 28.8 | 8.0 | 100.0 | 7.2 | 1.6 | 54.4 | 54.4 | 99.2 | 48.57 |
| | + OBCACHE-K | 97.84 | 80.27 | 94.4 | 24.8 | 0.0 | 42.2 | 13.4 | 100.0 | 12.0 | 1.6 | 59.2 | 56.8 | 99.36 | **52.45** |
| | SnapKV | 97.12 | 77.33 | 72.8 | 41.6 | 17.6 | 38.4 | 19.4 | 81.6 | 61.6 | 1.6 | 52.8 | 54.4 | 78.56 | 53.45 |
| | + OBCACHE-K | 95.84 | 76.8 | 84.8 | 53.6 | 12.8 | 40.8 | 17.8 | 88.0 | 67.2 | 1.6 | 55.2 | 55.2 | 91.36 | **57.0** |
| | AdaKV | 98.72 | 81.87 | 81.6 | 63.2 | 38.4 | 39.8 | 18.8 | 89.6 | 65.6 | 1.6 | 62.4 | 52.8 | 95.2 | 60.74 |
| | + OBCACHE-K | 99.04 | 80.8 | 93.6 | 94.4 | 84.8 | 68.4 | 27.2 | 100.0 | 83.2 | 1.6 | 63.2 | 55.2 | 99.52 | **73.15** |

*Table 4.* RULER-4K full results for Qwen2.5-7B-Instruct.

| Group | Methods | CWE | FWE | MK-1 | MK-2 | MK-3 | MQ | MV | S-1 | S-2 | S-3 | QA-1 | QA-2 | VT | Avg. |
|---|---|---|---|---|---|---|---|---|---|---|---|---|---|---|---|
| | ALL KV | 96.96 | 76.27 | 100.0 | 99.2 | 100.0 | 100.0 | 99.4 | 100.0 | 100.0 | 100.0 | 84.8 | 60.0 | 99.84 | 93.57 |
| 10% KV (q-Aware) | H2O | 60.4 | 41.87 | 11.2 | 3.2 | 0.0 | 9.6 | 8.4 | 24.8 | 0.8 | 1.6 | 71.2 | 51.2 | 10.72 | 22.69 |
| | + OBCACHE-K | 62.72 | 40.27 | 11.2 | 4.0 | 0.0 | 10.4 | 9.2 | 44.0 | 4.8 | 1.6 | 76.8 | 52.0 | 23.52 | **26.19** |
| | TOVA | 27.44 | 30.93 | 19.2 | 0.0 | 0.0 | 0.6 | 4.2 | 71.2 | 40.0 | 0.0 | 78.4 | 52.8 | 2.4 | 25.17 |
| | + OBCACHE-K | 36.08 | 32.0 | 20.0 | 0.0 | 0.0 | 0.2 | 7.4 | 84.0 | 49.6 | 0.0 | 80.0 | 52.8 | 14.08 | **28.94** |
| | SnapKV | 62.56 | 43.47 | 12.0 | 6.4 | 0.0 | 10.4 | 8.8 | 47.2 | 11.2 | 1.6 | 73.6 | 50.4 | 24.48 | 27.09 |
| | + OBCACHE-K | 49.6 | 40.8 | 12.0 | 6.4 | 0.8 | 11.0 | 9.0 | 76.8 | 20.8 | 1.6 | 80.0 | 51.2 | 32.0 | **30.15** |
| | AdaKV | 68.64 | 45.07 | 12.0 | 4.8 | 2.4 | 10.4 | 8.8 | 61.6 | 9.6 | 1.6 | 74.4 | 56.8 | 26.56 | 29.44 |
| | + OBCACHE-K | 48.32 | 41.6 | 12.0 | 12.0 | 0.0 | 11.6 | 9.0 | 77.6 | 26.4 | 1.6 | 75.2 | 49.6 | 20.8 | **29.67** |
| 20% KV (q-Aware) | H2O | 80.08 | 44.0 | 11.2 | 6.4 | 1.6 | 17.0 | 10.2 | 68.8 | 22.4 | 1.6 | 76.0 | 50.4 | 24.0 | 31.82 |
| | + OBCACHE-K | 85.2 | 44.8 | 21.6 | 15.2 | 0.0 | 22.4 | 13.8 | 82.4 | 60.8 | 1.6 | 80.0 | 54.4 | 42.56 | **40.37** |
| | TOVA | 41.2 | 45.07 | 33.6 | 0.0 | 0.0 | 7.0 | 21.8 | 98.4 | 88.0 | 0.0 | 78.4 | 56.8 | 44.8 | 39.62 |
| | + OBCACHE-K | 50.88 | 47.73 | 38.4 | 0.0 | 0.0 | 10.2 | 27.6 | 98.4 | 92.8 | 0.0 | 81.6 | 60.0 | 67.84 | **44.27** |
| | SnapKV | 81.92 | 52.53 | 17.6 | 14.4 | 4.0 | 26.4 | 13.4 | 81.6 | 63.2 | 1.6 | 82.4 | 61.6 | 59.36 | 43.08 |
| | + OBCACHE-K | 72.08 | 50.93 | 35.2 | 21.6 | 4.0 | 29.2 | 17.4 | 95.2 | 72.8 | 1.6 | 82.4 | 58.4 | 77.44 | **47.56** |
| | AdaKV | 90.88 | 54.67 | 17.6 | 22.4 | 6.4 | 25.2 | 13.0 | 90.4 | 56.8 | 1.6 | 84.0 | 57.6 | 60.32 | 44.68 |
| | + OBCACHE-K | 69.84 | 54.4 | 36.0 | 30.4 | 11.2 | 30.6 | 15.8 | 92.0 | 68.8 | 1.6 | 83.2 | 57.6 | 76.32 | **48.29** |
| 30% KV (q-Aware) | H2O | 87.84 | 45.33 | 18.4 | 16.0 | 4.0 | 29.0 | 14.8 | 84.8 | 62.4 | 1.6 | 76.8 | 57.6 | 35.2 | 41.06 |
| | + OBCACHE-K | 93.84 | 50.4 | 40.8 | 35.2 | 4.8 | 45.6 | 24.6 | 96.8 | 88.8 | 1.6 | 81.6 | 57.6 | 56.96 | **52.2** |
| | TOVA | 54.8 | 56.27 | 53.6 | 0.8 | 0.0 | 18.4 | 38.8 | 100.0 | 98.4 | 0.0 | 77.6 | 59.2 | 77.92 | 48.91 |
| | + OBCACHE-K | 63.52 | 62.13 | 64.0 | 2.4 | 0.0 | 23.8 | 49.0 | 100.0 | 98.4 | 0.0 | 77.6 | 59.2 | 89.12 | **53.01** |
| | SnapKV | 91.84 | 55.73 | 28.0 | 28.8 | 13.6 | 44.8 | 22.4 | 90.4 | 77.6 | 1.6 | 82.4 | 60.0 | 77.28 | 51.88 |
| | + OBCACHE-K | 84.32 | 58.13 | 62.4 | 31.2 | 12.0 | 58.8 | 28.8 | 94.4 | 88.0 | 1.6 | 84.8 | 60.0 | 93.76 | **58.32** |
| | AdaKV | 95.36 | 60.8 | 33.6 | 37.6 | 19.2 | 42.6 | 23.8 | 90.4 | 76.8 | 1.6 | 84.0 | 60.8 | 85.12 | 54.74 |
| | + OBCACHE-K | 87.2 | 59.73 | 60.0 | 59.2 | 24.8 | 62.4 | 27.8 | 92.8 | 85.6 | 1.6 | 82.4 | 62.4 | 95.68 | **61.66** |
| 40% KV (q-Aware) | H2O | 92.96 | 48.27 | 36.0 | 31.2 | 20.0 | 48.0 | 20.4 | 95.2 | 76.8 | 1.6 | 79.2 | 58.4 | 48.8 | 50.53 |
| | + OBCACHE-K | 95.28 | 56.8 | 70.4 | 48.0 | 16.0 | 62.6 | 29.8 | 97.6 | 88.8 | 1.6 | 81.6 | 57.6 | 73.6 | **59.98** |
| | TOVA | 64.32 | 64.53 | 73.6 | 1.6 | 0.8 | 34.4 | 58.8 | 100.0 | 98.4 | 0.8 | 79.2 | 59.2 | 90.88 | 56.06 |
| | + OBCACHE-K | 71.36 | 67.2 | 83.2 | 1.6 | 0.8 | 44.8 | 66.0 | 100.0 | 99.2 | 1.6 | 78.4 | 61.6 | 94.4 | **59.24** |
| | SnapKV | 94.24 | 60.27 | 43.2 | 36.8 | 30.4 | 60.2 | 27.0 | 95.2 | 88.0 | 1.6 | 80.8 | 60.0 | 86.08 | 58.75 |
| | + OBCACHE-K | 92.72 | 61.87 | 68.0 | 47.2 | 32.0 | 75.4 | 34.0 | 94.4 | 88.8 | 1.6 | 84.0 | 58.4 | 96.64 | **64.23** |
| | AdaKV | 97.04 | 64.27 | 54.4 | 51.2 | 48.8 | 61.2 | 28.8 | 92.8 | 85.6 | 1.6 | 83.2 | 60.0 | 93.12 | 63.23 |
| | + OBCACHE-K | 94.48 | 66.13 | 80.8 | 73.6 | 56.8 | 76.6 | 39.6 | 92.8 | 87.2 | 1.6 | 82.4 | 59.2 | 96.64 | **69.83** |
| 10% KV (q-Agnostic) | H2O | 61.6 | 60.53 | 10.4 | 1.6 | 0.0 | 8.4 | 8.2 | 11.2 | 0.8 | 1.6 | 22.4 | 23.2 | 5.28 | 16.55 |
| | + OBCACHE-K | 64.08 | 60.8 | 11.2 | 1.6 | 0.0 | 8.4 | 8.6 | 17.6 | 0.8 | 1.6 | 22.4 | 24.8 | 5.6 | **17.5** |
| | TOVA | 63.68 | 4.8 | 1.6 | 0.0 | 0.0 | 0.4 | 0.4 | 22.4 | 0.8 | 0.0 | 20.8 | 20.8 | 0.96 | 10.51 |
| | + OBCACHE-K | 72.0 | 9.33 | 2.4 | 0.0 | 0.0 | 0.6 | 0.6 | 29.6 | 0.8 | 0.0 | 21.6 | 20.8 | 8.64 | **12.8** |
| | SnapKV | 60.32 | 57.07 | 11.2 | 2.4 | 0.0 | 9.6 | 8.8 | 19.2 | 0.8 | 1.6 | 20.8 | 23.2 | 11.2 | 17.4 |
| | + OBCACHE-K | 50.16 | 54.67 | 11.2 | 3.2 | 0.0 | 9.6 | 8.8 | 44.0 | 0.8 | 1.6 | 19.2 | 24.0 | 12.0 | **18.4** |
| | AdaKV | 67.76 | 56.53 | 11.2 | 3.2 | 3.2 | 9.6 | 8.6 | 30.4 | 0.8 | 1.6 | 21.6 | 23.2 | 8.8 | 18.96 |
| | + OBCACHE-K | 45.28 | 59.47 | 11.2 | 4.0 | 0.8 | 9.6 | 8.8 | 58.4 | 1.6 | 1.6 | 21.6 | 24.0 | 6.56 | **19.45** |
| 20% KV (q-Agnostic) | H2O | 75.2 | 65.87 | 11.2 | 2.4 | 0.0 | 9.4 | 8.6 | 28.0 | 0.8 | 1.6 | 40.0 | 35.2 | 8.0 | 22.02 |
| | + OBCACHE-K | 84.16 | 64.53 | 11.2 | 5.6 | 0.0 | 9.4 | 8.8 | 40.0 | 0.8 | 1.6 | 40.8 | 41.6 | 14.08 | **24.81** |
| | TOVA | 76.16 | 17.6 | 4.8 | 0.0 | 0.0 | 1.2 | 1.6 | 57.6 | 0.8 | 0.8 | 28.0 | 25.6 | 25.12 | 18.41 |
| | + OBCACHE-K | 84.64 | 24.0 | 4.0 | 0.0 | 0.0 | 1.8 | 1.6 | 57.6 | 0.8 | 0.8 | 30.4 | 31.2 | 48.32 | **21.94** |
| | SnapKV | 79.52 | 61.33 | 11.2 | 8.0 | 1.6 | 9.6 | 8.8 | 45.6 | 1.6 | 1.6 | 36.0 | 39.2 | 26.4 | 25.42 |
| | + OBCACHE-K | 72.16 | 62.4 | 12.0 | 8.0 | 0.8 | 9.6 | 8.8 | 67.2 | 1.6 | 1.6 | 32.8 | 42.4 | 37.28 | **27.43** |
| | AdaKV | 90.16 | 62.13 | 11.2 | 8.0 | 3.2 | 9.6 | 8.8 | 58.4 | 0.8 | 1.6 | 35.2 | 36.0 | 30.4 | 27.35 |
| | + OBCACHE-K | 65.92 | 63.73 | 11.2 | 11.2 | 4.8 | 9.6 | 8.8 | 81.6 | 2.4 | 1.6 | 38.4 | 35.2 | 28.48 | **27.92** |
| 30% KV (q-Agnostic) | H2O | 86.48 | 65.33 | 11.2 | 4.8 | 0.8 | 9.4 | 8.8 | 48.0 | 2.4 | 1.6 | 47.2 | 40.8 | 12.16 | 26.07 |
| | + OBCACHE-K | 92.08 | 66.4 | 12.0 | 14.4 | 0.8 | 10.0 | 9.0 | 56.8 | 3.2 | 1.6 | 53.6 | 44.0 | 22.72 | **29.74** |
| | TOVA | 84.64 | 25.6 | 8.0 | 0.0 | 0.0 | 3.0 | 3.2 | 83.2 | 0.8 | 0.0 | 39.2 | 37.6 | 71.2 | 27.42 |
| | + OBCACHE-K | 90.24 | 33.07 | 8.8 | 0.0 | 0.0 | 5.2 | 2.8 | 91.2 | 1.6 | 0.8 | 39.2 | 38.4 | 83.68 | **30.38** |
| | SnapKV | 90.48 | 64.27 | 12.0 | 14.4 | 4.0 | 9.6 | 8.8 | 57.6 | 3.2 | 1.6 | 47.2 | 40.8 | 43.36 | 30.56 |
| | + OBCACHE-K | 84.96 | 64.53 | 12.8 | 16.8 | 4.0 | 9.6 | 9.2 | 81.6 | 4.8 | 1.6 | 44.0 | 44.0 | 64.32 | **34.02** |
| | AdaKV | 93.92 | 64.0 | 12.8 | 18.4 | 4.0 | 9.6 | 9.0 | 72.0 | 4.8 | 1.6 | 46.4 | 40.0 | 53.76 | 33.41 |
| | + OBCACHE-K | 87.28 | 66.67 | 16.8 | 20.8 | 16.8 | 10.4 | 9.0 | 87.2 | 11.2 | 1.6 | 48.0 | 45.6 | 56.64 | **36.77** |
| 40% KV (q-Agnostic) | H2O | 90.48 | 69.07 | 11.2 | 9.6 | 2.4 | 9.6 | 8.8 | 61.6 | 1.6 | 1.6 | 60.0 | 52.0 | 21.28 | 30.71 |
| | + OBCACHE-K | 94.72 | 69.87 | 13.6 | 32.8 | 3.2 | 10.8 | 9.0 | 69.6 | 3.2 | 1.6 | 60.0 | 49.6 | 38.72 | **35.13** |
| | TOVA | 91.2 | 39.2 | 9.6 | 0.8 | 0.0 | 5.4 | 4.8 | 96.0 | 0.8 | 0.8 | 44.0 | 45.6 | 85.92 | 32.62 |
| | + OBCACHE-K | 93.6 | 46.93 | 10.4 | 2.4 | 0.0 | 8.8 | 5.8 | 95.2 | 1.6 | 1.6 | 43.2 | 43.2 | 92.0 | **34.21** |
| | SnapKV | 94.08 | 65.07 | 13.6 | 19.2 | 8.0 | 10.0 | 8.8 | 67.2 | 7.2 | 1.6 | 56.0 | 48.0 | 56.16 | 34.99 |
| | + OBCACHE-K | 92.8 | 66.67 | 19.2 | 26.4 | 8.8 | 11.0 | 9.2 | 90.4 | 13.6 | 1.6 | 57.6 | 50.4 | 75.84 | **40.27** |
| | AdaKV | 96.8 | 70.4 | 16.0 | 28.0 | 13.6 | 10.6 | 9.2 | 77.6 | 9.6 | 1.6 | 64.8 | 44.0 | 70.4 | 39.43 |
| | + OBCACHE-K | 93.84 | 69.6 | 29.6 | 40.0 | 35.2 | 13.8 | 9.4 | 98.4 | 28.0 | 1.6 | 66.4 | 49.6 | 84.0 | **47.65** |

*Table 5.* RULER-32K full results for LLaMA-3.1-8B-Instruct.

| | Methods | CWE | FWE | MK-1 | MK-2 | MK-3 | MQ | MV | S-1 | S-2 | S-3 | QA-1 | QA-2 | VT | Avg. |
|---|---|---|---|---|---|---|---|---|---|---|---|---|---|---|---|
| | ALL KV | 31.44 | 88.27 | 100.0 | 98.4 | 99.2 | 99.8 | 99.4 | 100.0 | 100.0 | 100.0 | 84.0 | 52.0 | 98.88 | 88.57 |
| **10% KV (q-Aware)** | H2O | 22.72 | 69.6 | 92.8 | 20.0 | 15.2 | 91.8 | 51.6 | 98.4 | 96.8 | 4.0 | 81.6 | 50.4 | 84.64 | 59.97 |
| | + OBCACHE-K | 32.64 | 69.6 | 99.2 | 40.0 | 12.0 | 93.6 | 69.8 | 98.4 | 96.8 | 5.6 | 84.0 | 48.8 | 95.52 | **65.07** |
| | TOVA | 27.6 | 66.13 | 99.2 | 84.8 | 0.0 | 92.8 | 92.0 | 100.0 | 97.6 | 42.4 | 78.4 | 52.8 | 96.96 | 71.59 |
| | + OBCACHE-K | 35.12 | 67.47 | 99.2 | 86.4 | 0.0 | 92.8 | 93.6 | 99.2 | 98.4 | 46.4 | 82.4 | 49.6 | 98.72 | **73.02** |
| | SnapKV | 5.84 | 61.33 | 99.2 | 72.8 | 36.8 | 97.0 | 92.6 | 100.0 | 97.6 | 6.4 | 83.2 | 51.2 | 95.36 | 69.18 |
| | + OBCACHE-K | 9.76 | 57.6 | 100.0 | 88.0 | 24.8 | 93.0 | 91.2 | 100.0 | 98.4 | 5.6 | 84.0 | 52.0 | 97.6 | **69.38** |
| | AdaKV | 5.76 | 68.53 | 99.2 | 81.6 | 52.0 | 97.4 | 95.2 | 100.0 | 97.6 | 10.4 | 82.4 | 51.2 | 97.28 | 72.2 |
| | + OBCACHE-K | 16.24 | 72.8 | 99.2 | 95.2 | 78.4 | 97.8 | 97.0 | 100.0 | 97.6 | 28.0 | 83.2 | 51.2 | 98.24 | **78.07** |
| **20% KV (q-Aware)** | H2O | 31.36 | 79.73 | 99.2 | 43.2 | 55.2 | 99.0 | 91.6 | 100.0 | 96.8 | 20.0 | 81.6 | 51.2 | 94.24 | 72.55 |
| | + OBCACHE-K | 44.48 | 79.47 | 100.0 | 66.4 | 54.4 | 97.8 | 94.6 | 99.2 | 97.6 | 30.4 | 84.0 | 49.6 | 98.24 | **76.63** |
| | TOVA | 31.44 | 76.53 | 98.4 | 90.4 | 0.0 | 95.8 | 97.8 | 100.0 | 98.4 | 76.8 | 79.2 | 50.4 | 99.36 | 76.5 |
| | + OBCACHE-K | 40.88 | 75.73 | 99.2 | 94.4 | 0.0 | 96.6 | 96.2 | 99.2 | 97.6 | 80.0 | 84.8 | 50.4 | 99.36 | **78.03** |
| | SnapKV | 14.56 | 70.4 | 98.4 | 94.4 | 72.8 | 98.6 | 98.0 | 100.0 | 97.6 | 48.0 | 82.4 | 52.0 | 97.92 | **78.87** |
| | + OBCACHE-K | 20.8 | 68.53 | 99.2 | 95.2 | 64.8 | 98.6 | 98.4 | 100.0 | 96.8 | 47.2 | 84.0 | 52.0 | 99.04 | 78.81 |
| | AdaKV | 19.2 | 75.2 | 100.0 | 93.6 | 88.0 | 99.4 | 99.4 | 100.0 | 99.2 | 56.0 | 84.0 | 51.2 | 99.52 | 81.9 |
| | + OBCACHE-K | 29.28 | 78.67 | 100.0 | 97.6 | 98.4 | 99.6 | 99.4 | 100.0 | 97.6 | 84.0 | 84.0 | 51.2 | 99.04 | **86.06** |
| **30% KV (q-Aware)** | H2O | 31.68 | 83.2 | 99.2 | 58.4 | 76.8 | 100.0 | 97.0 | 99.2 | 98.4 | 47.2 | 82.4 | 51.2 | 96.16 | 78.53 |
| | + OBCACHE-K | 47.68 | 82.93 | 100.0 | 79.2 | 78.4 | 99.4 | 97.6 | 99.2 | 98.4 | 60.0 | 84.0 | 50.4 | 98.4 | **82.74** |
| | TOVA | 28.72 | 79.73 | 100.0 | 92.8 | 8.8 | 97.6 | 97.4 | 100.0 | 97.6 | 94.4 | 80.0 | 52.0 | 99.52 | 79.12 |
| | + OBCACHE-K | 45.28 | 79.73 | 100.0 | 95.2 | 9.6 | 98.0 | 97.2 | 99.2 | 97.6 | 96.8 | 84.8 | 51.2 | 99.52 | **81.09** |
| | SnapKV | 22.4 | 75.47 | 99.2 | 95.2 | 88.8 | 99.8 | 98.2 | 100.0 | 97.6 | 71.2 | 84.0 | 51.2 | 98.24 | 83.18 |
| | + OBCACHE-K | 28.8 | 75.73 | 98.4 | 97.6 | 84.0 | 99.0 | 98.2 | 100.0 | 96.8 | 75.2 | 84.0 | 52.0 | 99.04 | **83.75** |
| | AdaKV | 25.44 | 78.93 | 100.0 | 99.2 | 96.0 | 99.6 | 99.8 | 100.0 | 100.0 | 86.4 | 84.0 | 50.4 | 99.04 | 86.06 |
| | + OBCACHE-K | 33.76 | 84.27 | 100.0 | 98.4 | 98.4 | 99.8 | 99.4 | 100.0 | 100.0 | 96.0 | 84.0 | 50.4 | 99.84 | **88.02** |
| **40% KV (q-Aware)** | H2O | 26.72 | 83.73 | 100.0 | 74.4 | 96.0 | 99.2 | 97.8 | 99.2 | 98.4 | 68.0 | 81.6 | 50.4 | 97.28 | 82.52 |
| | + OBCACHE-K | 47.28 | 85.07 | 100.0 | 88.8 | 92.8 | 99.6 | 98.8 | 98.4 | 98.4 | 80.0 | 84.0 | 51.2 | 99.52 | **86.45** |
| | TOVA | 27.92 | 83.47 | 100.0 | 95.2 | 31.2 | 98.6 | 98.2 | 100.0 | 98.4 | 100.0 | 80.0 | 51.2 | 99.52 | 81.85 |
| | + OBCACHE-K | 42.8 | 82.13 | 100.0 | 96.0 | 31.2 | 97.8 | 98.2 | 99.2 | 99.2 | 84.8 | 84.8 | 50.4 | 99.52 | **83.05** |
| | SnapKV | 24.56 | 79.47 | 100.0 | 99.2 | 94.4 | 99.2 | 99.2 | 100.0 | 98.4 | 89.6 | 84.0 | 51.2 | 98.72 | 86.0 |
| | + OBCACHE-K | 30.4 | 78.67 | 99.2 | 98.4 | 96.8 | 99.6 | 99.0 | 100.0 | 97.6 | 92.0 | 84.0 | 51.2 | 98.4 | **86.56** |
| | AdaKV | 29.12 | 82.4 | 100.0 | 98.4 | 99.2 | 99.6 | 99.6 | 100.0 | 100.0 | 94.4 | 84.0 | 51.2 | 99.04 | 87.46 |
| | + OBCACHE-K | 33.92 | 86.4 | 100.0 | 98.4 | 99.2 | 99.6 | 99.6 | 100.0 | 100.0 | 98.4 | 84.0 | 51.2 | 99.84 | **88.5** |
| **10% KV (q-Agnostic)** | H2O | 0.0 | 72.53 | 10.4 | 3.2 | 0.0 | 11.2 | 7.0 | 76.0 | 6.4 | 2.4 | 21.6 | 32.0 | 31.52 | 21.1 |
| | + OBCACHE-K | 0.4 | 69.33 | 12.0 | 4.8 | 0.0 | 12.4 | 8.6 | 85.6 | 8.0 | 2.4 | 26.4 | 33.6 | 54.24 | **24.44** |
| | TOVA | 0.0 | 29.33 | 12.0 | 0.8 | 0.0 | 9.8 | 6.0 | 95.2 | 0.8 | 2.4 | 19.2 | 29.6 | 76.32 | 21.65 |
| | + OBCACHE-K | 0.0 | 39.73 | 13.6 | 0.0 | 0.0 | 10.4 | 6.2 | 96.8 | 1.6 | 2.4 | 20.0 | 28.0 | 89.12 | 23.68 |
| | SnapKV | 0.0 | 65.07 | 12.0 | 4.8 | 0.0 | 12.8 | 8.2 | 89.6 | 8.8 | 2.4 | 25.6 | 31.2 | 57.44 | 24.45 |
| | + OBCACHE-K | 0.0 | 52.8 | 11.2 | 4.8 | 0.0 | 11.8 | 7.8 | 100.0 | 8.0 | 2.4 | 24.0 | 36.8 | 78.72 | **26.02** |
| | AdaKV | 0.0 | 72.0 | 12.0 | 8.0 | 0.8 | 13.2 | 8.6 | 92.0 | 13.6 | 2.4 | 24.8 | 30.4 | 72.16 | 26.92 |
| | + OBCACHE-K | 0.32 | 71.47 | 15.2 | 16.8 | 4.0 | 14.2 | 10.0 | 99.2 | 20.8 | 2.4 | 29.6 | 36.8 | 91.04 | **31.68** |
| **20% KV (q-Agnostic)** | H2O | 0.56 | 80.27 | 20.0 | 4.8 | 0.0 | 15.2 | 11.6 | 86.4 | 18.4 | 4.0 | 26.4 | 37.6 | 51.68 | 27.45 |
| | + OBCACHE-K | 2.64 | 78.4 | 21.6 | 12.0 | 0.0 | 19.2 | 15.4 | 93.6 | 15.2 | 4.0 | 35.2 | 38.4 | 71.04 | **31.28** |
| | TOVA | 0.16 | 59.47 | 34.4 | 0.8 | 0.0 | 12.8 | 10.6 | 98.4 | 4.0 | 2.4 | 26.4 | 37.6 | 96.96 | 29.54 |
| | + OBCACHE-K | 0.16 | 62.13 | 39.2 | 1.6 | 0.0 | 14.0 | 11.6 | 98.4 | 5.6 | 2.4 | 27.2 | 35.2 | 98.72 | 30.48 |
| | SnapKV | 0.32 | 70.4 | 23.2 | 9.6 | 1.6 | 16.6 | 12.8 | 97.6 | 28.8 | 4.8 | 27.2 | 36.0 | 75.04 | 31.07 |
| | + OBCACHE-K | 0.48 | 64.8 | 20.0 | 15.2 | 1.6 | 15.6 | 11.8 | 100.0 | 23.2 | 2.4 | 32.0 | 40.8 | 89.12 | **32.08** |
| | AdaKV | 1.12 | 75.47 | 25.6 | 21.6 | 12.0 | 20.0 | 18.2 | 99.2 | 52.8 | 8.0 | 32.0 | 41.6 | 89.44 | 38.23 |
| | + OBCACHE-K | 2.32 | 77.33 | 34.4 | 49.6 | 26.4 | 31.2 | 33.4 | 100.0 | 77.6 | 8.0 | 37.6 | 42.4 | 97.44 | **47.51** |
| **30% KV (q-Agnostic)** | H2O | 1.28 | 83.47 | 31.2 | 12.8 | 0.8 | 25.6 | 20.2 | 90.4 | 35.2 | 8.0 | 32.8 | 40.0 | 65.44 | 34.45 |
| | + OBCACHE-K | 4.64 | 83.2 | 37.6 | 19.2 | 0.8 | 33.0 | 32.4 | 96.0 | 42.4 | 8.8 | 38.4 | 43.2 | 81.28 | **40.07** |
| | TOVA | 0.88 | 72.27 | 55.2 | 3.2 | 0.0 | 18.6 | 24.0 | 100.0 | 8.8 | 2.4 | 29.6 | 42.4 | 98.72 | 35.08 |
| | + OBCACHE-K | 3.2 | 74.13 | 60.8 | 5.6 | 0.0 | 25.0 | 31.0 | 99.2 | 13.6 | 2.4 | 32.8 | 40.8 | 99.36 | **37.53** |
| | SnapKV | 0.88 | 73.33 | 32.8 | 15.2 | 4.8 | 23.8 | 20.2 | 100.0 | 62.4 | 11.2 | 32.8 | 43.2 | 86.4 | **39.0** |
| | + OBCACHE-K | 0.96 | 71.73 | 34.4 | 20.8 | 4.8 | 22.2 | 17.8 | 100.0 | 53.6 | 7.2 | 34.4 | 43.2 | 93.76 | 38.83 |
| | AdaKV | 2.24 | 80.0 | 52.8 | 45.6 | 31.2 | 38.0 | 43.6 | 100.0 | 81.6 | 24.0 | 39.2 | 47.2 | 97.92 | 52.57 |
| | + OBCACHE-K | 5.28 | 83.47 | 65.6 | 72.0 | 63.2 | 50.0 | 70.2 | 100.0 | 90.4 | 30.4 | 50.4 | 46.4 | 99.2 | **63.58** |
| **40% KV (q-Agnostic)** | H2O | 3.76 | 84.0 | 44.8 | 20.0 | 5.6 | 37.6 | 40.6 | 92.8 | 53.6 | 17.6 | 38.4 | 43.2 | 75.84 | 42.91 |
| | + OBCACHE-K | 11.76 | 82.4 | 49.6 | 31.2 | 7.2 | 42.4 | 53.6 | 96.8 | 63.2 | 17.6 | 44.8 | 44.8 | 89.28 | **48.82** |
| | TOVA | 3.76 | 83.73 | 72.0 | 10.4 | 0.0 | 30.0 | 47.2 | 100.0 | 17.6 | 4.8 | 36.0 | 43.2 | 99.52 | 42.17 |
| | + OBCACHE-K | 7.04 | 85.33 | 76.0 | 11.2 | 0.0 | 38.0 | 52.8 | 100.0 | 24.8 | 4.8 | 39.2 | 44.8 | 99.52 | **44.88** |
| | SnapKV | 1.6 | 76.53 | 50.4 | 25.6 | 15.2 | 37.4 | 39.0 | 99.2 | 79.2 | 19.2 | 42.4 | 45.6 | 91.36 | 47.9 |
| | + OBCACHE-K | 2.64 | 75.2 | 43.2 | 36.8 | 15.2 | 34.2 | 37.4 | 100.0 | 76.8 | 13.6 | 45.6 | 48.0 | 95.68 | **48.02** |
| | AdaKV | 3.68 | 83.47 | 73.6 | 61.6 | 52.0 | 57.2 | 72.0 | 100.0 | 91.2 | 42.4 | 52.8 | 46.4 | 99.2 | 64.27 |
| | + OBCACHE-K | 11.28 | 85.07 | 88.8 | 86.4 | 80.8 | 70.2 | 86.8 | 100.0 | 96.8 | 55.2 | 62.4 | 50.4 | 98.72 | **74.84** |

*Table 6.* RULER-32K full results for Qwen2.5-7B-Instruct.

| | Methods | CWE | FWE | MK-1 | MK-2 | MK-3 | MQ | MV | S-1 | S-2 | S-3 | QA-1 | QA-2 | VT | Avg. |
|---|---|---|---|---|---|---|---|---|---|---|---|---|---|---|---|
| | ALL KV | 62.8 | 80.53 | 98.4 | 87.2 | 62.4 | 99.8 | 91.2 | 100.0 | 100.0 | 100.0 | 70.4 | 56.0 | 95.04 | 84.91 |
| **10% KV (q-Aware)** | H2O | 31.04 | 57.87 | 21.6 | 4.0 | 0.0 | 29.6 | 7.0 | 91.2 | 35.2 | 2.4 | 60.0 | 52.0 | 72.0 | 35.69 |
| | + OBCACHE-K | 38.96 | 62.93 | 52.8 | 4.8 | 0.0 | 50.2 | 10.2 | 91.2 | 68.0 | 2.4 | 58.4 | 50.4 | 88.0 | **44.48** |
| | TOVA | 22.4 | 49.07 | 26.4 | 0.0 | 0.0 | 5.2 | 8.0 | 99.2 | 46.4 | 0.0 | 57.6 | 49.6 | 53.28 | 32.09 |
| | + OBCACHE-K | 26.48 | 48.8 | 32.0 | 0.0 | 0.0 | 11.2 | 14.2 | 99.2 | 66.4 | 0.0 | 57.6 | 46.4 | 63.84 | **35.86** |
| | SnapKV | 46.56 | 62.4 | 43.2 | 7.2 | 0.8 | 55.4 | 13.2 | 92.0 | 76.0 | 2.4 | 60.8 | 54.4 | 92.96 | 46.72 |
| | + OBCACHE-K | 40.48 | 59.73 | 71.2 | 8.0 | 0.8 | 65.0 | 18.0 | 100.0 | 84.0 | 2.4 | 66.4 | 53.6 | 95.68 | **51.18** |
| | AdaKV | 45.2 | 65.6 | 60.8 | 7.2 | 1.6 | 64.4 | 16.6 | 95.2 | 87.2 | 2.4 | 61.6 | 55.2 | 92.64 | 50.43 |
| | + OBCACHE-K | 40.64 | 59.73 | 75.2 | 14.4 | 2.4 | 70.0 | 19.6 | 99.2 | 86.4 | 3.2 | 66.4 | 56.0 | 96.96 | **53.09** |
| **20% KV (q-Aware)** | H2O | 38.08 | 68.27 | 55.2 | 7.2 | 2.4 | 66.0 | 16.2 | 92.0 | 80.8 | 2.4 | 61.6 | 53.6 | 82.24 | 48.15 |
| | + OBCACHE-K | 47.04 | 70.4 | 72.0 | 14.4 | 0.8 | 71.0 | 28.0 | 98.4 | 87.2 | 2.4 | 63.2 | 58.4 | 92.0 | **54.25** |
| | TOVA | 29.68 | 59.73 | 49.6 | 0.0 | 0.0 | 22.0 | 22.8 | 100.0 | 84.8 | 0.0 | 59.2 | 53.6 | 82.08 | 43.35 |
| | + OBCACHE-K | 38.08 | 60.8 | 55.2 | 0.8 | 0.0 | 24.8 | 30.6 | 100.0 | 87.2 | 0.0 | 60.8 | 50.4 | 87.04 | **45.82** |
| | SnapKV | 55.68 | 69.33 | 76.8 | 15.2 | 7.2 | 86.6 | 28.0 | 97.6 | 95.2 | 3.2 | 66.4 | 54.4 | 93.44 | 57.62 |
| | + OBCACHE-K | 52.96 | 68.27 | 86.4 | 23.2 | 6.4 | 88.2 | 39.2 | 100.0 | 98.4 | 6.4 | 66.4 | 53.6 | 98.08 | **60.58** |
| | AdaKV | 57.68 | 73.07 | 86.4 | 17.6 | 9.6 | 95.2 | 45.4 | 96.8 | 98.4 | 5.6 | 64.0 | 52.8 | 93.6 | 61.24 |
| | + OBCACHE-K | 60.16 | 68.27 | 90.4 | 24.8 | 14.4 | 93.2 | 57.0 | 100.0 | 99.2 | 10.4 | 65.6 | 56.8 | 96.8 | **64.39** |
| **30% KV (q-Aware)** | H2O | 44.16 | 72.53 | 70.4 | 12.0 | 10.4 | 83.8 | 27.8 | 92.0 | 88.8 | 3.2 | 64.0 | 55.2 | 86.24 | 54.66 |
| | + OBCACHE-K | 54.56 | 74.13 | 87.2 | 23.2 | 5.6 | 85.8 | 41.4 | 99.2 | 91.2 | 6.4 | 68.8 | 58.4 | 94.08 | **60.77** |
| | TOVA | 35.68 | 66.67 | 65.6 | 0.8 | 0.0 | 35.0 | 39.4 | 100.0 | 91.2 | 1.6 | 63.2 | 51.2 | 93.12 | 49.5 |
| | + OBCACHE-K | 43.6 | 68.0 | 73.6 | 0.0 | 0.0 | 43.2 | 50.8 | 100.0 | 97.6 | 3.2 | 66.4 | 56.0 | 93.76 | **53.55** |
| | SnapKV | 60.64 | 71.47 | 88.8 | 20.0 | 19.2 | 96.2 | 47.6 | 97.6 | 99.2 | 8.0 | 64.0 | 56.0 | 92.8 | 63.19 |
| | + OBCACHE-K | 58.64 | 70.4 | 92.0 | 25.6 | 9.6 | 94.2 | 57.8 | 100.0 | 99.2 | 13.6 | 61.6 | 57.6 | 96.8 | **64.39** |
| | AdaKV | 60.0 | 72.8 | 94.4 | 26.4 | 21.6 | 98.2 | 68.2 | 99.2 | 100.0 | 12.0 | 64.0 | 57.6 | 94.4 | 66.83 |
| | + OBCACHE-K | 66.64 | 71.73 | 95.2 | 37.6 | 29.6 | 98.6 | 75.2 | 100.0 | 100.0 | 24.0 | 66.4 | 57.6 | 96.96 | **70.73** |
| **40% KV (q-Aware)** | H2O | 50.32 | 74.13 | 84.0 | 19.2 | 18.4 | 91.8 | 38.6 | 89.6 | 92.8 | 6.4 | 65.6 | 56.8 | 87.84 | 59.65 |
| | + OBCACHE-K | 57.52 | 76.53 | 88.0 | 38.4 | 15.2 | 94.0 | 56.4 | 99.2 | 95.2 | 10.4 | 64.0 | 58.4 | 93.92 | **65.17** |
| | TOVA | 47.44 | 71.2 | 84.0 | 2.4 | 0.0 | 54.2 | 55.2 | 100.0 | 99.2 | 8.0 | 64.8 | 56.0 | 92.8 | 56.56 |
| | + OBCACHE-K | 48.4 | 72.53 | 90.4 | 2.4 | 0.0 | 60.2 | 64.4 | 100.0 | 100.0 | 8.8 | 63.2 | 54.4 | 93.12 | **58.3** |
| | SnapKV | 61.2 | 73.33 | 93.6 | 25.6 | 30.4 | 98.4 | 61.4 | 100.0 | 100.0 | 13.6 | 64.8 | 57.6 | 93.76 | 67.21 |
| | + OBCACHE-K | 60.88 | 72.8 | 93.6 | 40.0 | 17.6 | 98.2 | 72.8 | 100.0 | 100.0 | 21.6 | 64.0 | 56.8 | 96.8 | **68.85** |
| | AdaKV | 61.84 | 76.0 | 99.2 | 35.2 | 30.4 | 99.2 | 79.2 | 100.0 | 100.0 | 30.4 | 64.8 | 59.2 | 96.48 | 71.69 |
| | + OBCACHE-K | 67.04 | 76.27 | 98.4 | 56.0 | 40.0 | 99.2 | 83.8 | 100.0 | 100.0 | 59.2 | 66.4 | 58.4 | 95.84 | **76.97** |
| **10% KV (q-Agnostic)** | H2O | 31.68 | 61.6 | 8.8 | 2.4 | 0.0 | 9.6 | 6.4 | 68.0 | 0.8 | 2.4 | 16.8 | 28.8 | 24.16 | 20.11 |
| | + OBCACHE-K | 40.56 | 62.4 | 8.8 | 3.2 | 0.0 | 10.0 | 6.4 | 80.8 | 0.8 | 2.4 | 16.8 | 28.8 | 53.6 | **24.2** |
| | TOVA | 35.76 | 10.4 | 4.0 | 0.0 | 0.0 | 3.0 | 0.8 | 80.0 | 0.8 | 0.0 | 16.8 | 24.0 | 19.04 | 14.97 |
| | + OBCACHE-K | 43.6 | 14.13 | 3.2 | 0.0 | 0.0 | 3.2 | 1.0 | 85.6 | 0.8 | 0.0 | 17.6 | 26.4 | 38.72 | **18.02** |
| | SnapKV | 43.44 | 63.73 | 9.6 | 4.8 | 0.0 | 10.4 | 6.4 | 73.6 | 0.8 | 2.4 | 16.8 | 29.6 | 71.36 | 25.61 |
| | + OBCACHE-K | 38.4 | 57.6 | 9.6 | 4.8 | 0.0 | 10.4 | 6.4 | 93.6 | 0.8 | 2.4 | 16.8 | 28.8 | 85.92 | **27.35** |
| | AdaKV | 45.76 | 67.2 | 9.6 | 4.8 | 0.0 | 10.4 | 6.4 | 79.2 | 1.6 | 2.4 | 17.6 | 28.8 | 72.8 | 26.66 |
| | + OBCACHE-K | 42.0 | 60.8 | 9.6 | 5.6 | 0.0 | 10.4 | 6.4 | 92.0 | 1.6 | 2.4 | 18.4 | 31.2 | 86.88 | **28.25** |
| **20% KV (q-Agnostic)** | H2O | 40.24 | 66.4 | 8.8 | 4.8 | 0.0 | 10.2 | 6.2 | 83.2 | 0.8 | 2.4 | 20.0 | 32.8 | 52.0 | 25.22 |
| | + OBCACHE-K | 49.28 | 69.6 | 8.8 | 6.4 | 0.0 | 10.0 | 6.4 | 89.6 | 0.8 | 2.4 | 22.4 | 32.8 | 71.2 | **28.44** |
| | TOVA | 46.4 | 22.93 | 4.0 | 0.0 | 0.0 | 4.8 | 2.0 | 95.2 | 0.8 | 0.0 | 23.2 | 26.4 | 64.96 | 22.36 |
| | + OBCACHE-K | 50.96 | 32.0 | 4.8 | 0.8 | 0.0 | 5.8 | 2.2 | 95.2 | 0.8 | 0.0 | 20.8 | 27.2 | 70.88 | **23.96** |
| | SnapKV | 57.2 | 69.6 | 9.6 | 7.2 | 0.8 | 10.4 | 6.4 | 86.4 | 1.6 | 2.4 | 18.4 | 33.6 | 78.4 | 29.38 |
| | + OBCACHE-K | 52.4 | 63.47 | 9.6 | 8.8 | 0.0 | 10.4 | 6.4 | 98.4 | 2.4 | 2.4 | 23.2 | 35.2 | 89.6 | **30.94** |
| | AdaKV | 60.64 | 72.8 | 9.6 | 7.2 | 0.8 | 10.6 | 6.4 | 88.0 | 3.2 | 2.4 | 20.8 | 36.0 | 85.44 | 31.07 |
| | + OBCACHE-K | 60.16 | 65.87 | 9.6 | 8.8 | 4.8 | 10.6 | 6.4 | 94.4 | 1.6 | 2.4 | 28.8 | 39.2 | 89.76 | **32.49** |
| **30% KV (q-Agnostic)** | H2O | 44.08 | 71.73 | 8.8 | 5.6 | 0.8 | 10.2 | 6.4 | 91.2 | 0.8 | 2.4 | 24.0 | 39.2 | 63.68 | 28.38 |
| | + OBCACHE-K | 55.28 | 72.0 | 6.8 | 6.8 | 0.8 | 10.6 | 6.2 | 92.8 | 3.2 | 2.4 | 28.8 | 40.0 | 79.2 | **31.51** |
| | TOVA | 54.24 | 44.53 | 7.2 | 0.8 | 0.0 | 6.8 | 2.6 | 98.4 | 0.8 | 0.0 | 24.8 | 34.4 | 80.48 | 27.31 |
| | + OBCACHE-K | 59.28 | 54.13 | 8.8 | 0.8 | 0.0 | 7.0 | 3.2 | 97.6 | 0.8 | 0.0 | 24.8 | 34.4 | 81.6 | **28.65** |
| | SnapKV | 62.32 | 71.47 | 9.6 | 8.8 | 1.6 | 10.8 | 6.4 | 87.2 | 4.8 | 2.4 | 27.2 | 37.6 | 85.12 | 31.95 |
| | + OBCACHE-K | 57.76 | 70.13 | 10.4 | 12.8 | 1.6 | 10.4 | 6.6 | 98.4 | 7.2 | 2.4 | 28.8 | 42.4 | 89.92 | **33.75** |
| | AdaKV | 65.04 | 74.93 | 10.4 | 13.6 | 3.2 | 10.8 | 6.4 | 91.2 | 9.6 | 2.4 | 25.6 | 40.0 | 87.36 | 33.89 |
| | + OBCACHE-K | 66.08 | 71.73 | 11.2 | 18.4 | 13.6 | 11.6 | 7.4 | 95.2 | 12.0 | 2.4 | 32.0 | 41.6 | 91.2 | **36.49** |
| **40% KV (q-Agnostic)** | H2O | 50.32 | 70.4 | 9.6 | 6.4 | 1.6 | 10.8 | 6.4 | 90.4 | 4.0 | 2.4 | 28.8 | 42.4 | 69.6 | 30.24 |
| | + OBCACHE-K | 56.24 | 73.33 | 9.6 | 19.2 | 3.2 | 10.8 | 6.6 | 94.4 | 4.8 | 2.4 | 32.8 | 44.8 | 79.36 | **33.66** |
| | TOVA | 59.84 | 60.0 | 8.8 | 1.6 | 0.0 | 8.4 | 4.2 | 99.2 | 0.8 | 0.8 | 25.6 | 42.4 | 85.76 | 30.57 |
| | + OBCACHE-K | 62.72 | 66.13 | 8.8 | 1.6 | 0.0 | 8.8 | 4.4 | 100.0 | 0.8 | 1.6 | 28.0 | 43.2 | 87.68 | **31.83** |
| | SnapKV | 62.32 | 74.93 | 9.6 | 12.8 | 4.0 | 10.8 | 6.6 | 89.6 | 8.0 | 2.4 | 26.4 | 44.0 | 87.52 | 33.77 |
| | + OBCACHE-K | 61.2 | 71.2 | 11.2 | 17.6 | 4.8 | 12.0 | 7.0 | 100.0 | 7.2 | 2.4 | 31.2 | 42.4 | 88.64 | **35.14** |
| | AdaKV | 63.84 | 78.4 | 11.2 | 20.0 | 8.0 | 12.6 | 7.8 | 91.2 | 13.6 | 3.2 | 36.8 | 44.0 | 87.84 | 36.81 |
| | + OBCACHE-K | 66.88 | 76.8 | 13.6 | 31.2 | 27.2 | 15.4 | 12.0 | 96.8 | 24.0 | 4.0 | 38.4 | 49.6 | 92.32 | **42.17** |

Table 7. LongBench full results for LLaMA-3.1-8B-Instruct.

| | Methods | Single-Document QA | | | Multi-Document QA | | | Summarization | | | Few-shot Learning | | | Synthetic | | Code | | |
| | | NrtvQA | Qasper | MF-en | HotpotQA | 2WikiMQA | Musique | GovReport | QMSum | MultiNews | TREC | TriviaQA | SAMSum | PCount | PRe | Lcc | RB-P | Avg. |
|---|---|---|---|---|---|---|---|---|---|---|---|---|---|---|---|---|---|---|
| | ALL KV | 30.68 | 47.72 | 54.64 | 59.49 | 51.04 | 32.86 | 35.33 | 25.2 | 26.99 | 72.5 | 89.55 | 44.83 | 10.15 | 100.0 | 52.65 | 49.24 | 48.93 |
| 10% KV (q-Aware) | H2O | 31.23 | 30.27 | 50.72 | 55.79 | 47.69 | 27.76 | 26.4 | 23.87 | 20.68 | 43.5 | 89.52 | 43.26 | 13.12 | 99.5 | 52.27 | 52.22 | 44.24 |
| | + OBCACHE-K | 29.6 | 34.32 | 52.46 | 57.58 | 49.51 | 30.15 | 26.81 | 23.96 | 20.87 | 46.5 | 90.24 | 43.6 | 11.62 | 99.5 | 50.9 | 52.61 | **45.01** |
| | TOVA | 30.27 | 31.82 | 53.47 | 55.04 | 50.5 | 28.63 | 26.2 | 23.5 | 20.75 | 52.5 | 91.25 | 44.9 | 11.58 | 100.0 | 51.55 | 48.27 | 45.01 |
| | + OBCACHE-K | 30.92 | 36.81 | 52.73 | 56.31 | 49.77 | 30.82 | 26.51 | 24.09 | 21.06 | 57.5 | 91.75 | 44.57 | 11.05 | 100.0 | 51.59 | 49.64 | **45.95** |
| | SnapKV | 30.91 | 37.07 | 53.87 | 57.11 | 44.36 | 27.94 | 25.71 | 23.85 | 19.6 | 48.0 | 87.8 | 42.54 | 10.05 | 99.5 | 52.97 | 52.16 | 44.59 |
| | + OBCACHE-K | 31.18 | 37.54 | 55.41 | 56.02 | 46.02 | 29.03 | 26.65 | 23.89 | 20.25 | 48.0 | 89.09 | 43.46 | 11.55 | 99.5 | 53.11 | 52.76 | **45.22** |
| | AdaKV | 30.93 | 38.91 | 53.25 | 57.78 | 47.13 | 30.71 | 26.13 | 24.29 | 19.98 | 52.0 | 89.59 | 43.17 | 10.1 | 99.5 | 53.98 | 52.72 | 45.64 |
| | + OBCACHE-K | 30.23 | 41.2 | 54.78 | 58.2 | 48.9 | 32.56 | 26.7 | 23.92 | 20.49 | 59.5 | 89.54 | 43.75 | 10.55 | 99.5 | 55.82 | 53.46 | **46.82** |
| 20% KV (q-Aware) | H2O | 31.84 | 37.48 | 52.84 | 57.31 | 50.9 | 29.83 | 28.47 | 24.28 | 22.79 | 53.0 | 90.17 | 43.13 | 13.65 | 99.5 | 52.38 | 52.19 | 46.24 |
| | + OBCACHE-K | 30.9 | 39.72 | 53.65 | 57.74 | 48.47 | 32.22 | 28.96 | 23.88 | 23.41 | 54.5 | 90.32 | 43.43 | 11.65 | 99.5 | 51.97 | 51.35 | **46.35** |
| | TOVA | 31.04 | 40.65 | 53.53 | 56.09 | 50.37 | 29.91 | 29.19 | 24.36 | 22.12 | 61.5 | 90.32 | 44.63 | 14.55 | 100.0 | 53.31 | 49.7 | 46.95 |
| | + OBCACHE-K | 29.56 | 42.47 | 55.35 | 57.06 | 50.17 | 31.04 | 29.22 | 24.62 | 22.61 | 64.0 | 90.74 | 44.69 | 10.55 | 100.0 | 52.95 | 50.61 | **47.23** |
| | SnapKV | 29.8 | 44.38 | 54.97 | 58.83 | 50.17 | 32.33 | 28.56 | 24.63 | 22.76 | 59.0 | 89.5 | 44.2 | 11.05 | 99.5 | 53.66 | 51.48 | 47.18 |
| | + OBCACHE-K | 29.53 | 45.05 | 55.59 | 59.17 | 49.54 | 31.9 | 29.1 | 24.09 | 23.21 | 59.0 | 89.36 | 43.94 | 11.63 | 99.5 | 53.55 | 52.25 | **47.28** |
| | AdaKV | 29.55 | 43.13 | 55.13 | 58.49 | 49.29 | 32.33 | 28.63 | 24.5 | 23.47 | 62.5 | 89.62 | 43.51 | 11.05 | 100.0 | 53.1 | 51.63 | 47.25 |
| | + OBCACHE-K | 30.0 | 45.32 | 56.47 | 58.58 | 51.11 | 33.27 | 29.33 | 24.55 | 23.33 | 68.0 | 89.86 | 43.74 | 11.05 | 99.5 | 52.7 | 52.19 | **48.06** |
| 30% KV (q-Aware) | H2O | 30.68 | 41.1 | 54.1 | 55.89 | 49.65 | 29.77 | 30.03 | 24.94 | 24.07 | 56.0 | 90.02 | 43.58 | 11.65 | 99.5 | 52.54 | 51.44 | 46.56 |
| | + OBCACHE-K | 30.09 | 43.48 | 54.87 | 57.75 | 50.17 | 31.81 | 30.37 | 24.54 | 24.5 | 59.5 | 89.92 | 44.31 | 12.15 | 99.5 | 52.37 | 50.71 | **47.25** |
| | TOVA | 30.77 | 43.96 | 54.65 | 55.82 | 50.77 | 29.51 | 30.79 | 24.64 | 24.04 | 65.5 | 90.31 | 44.45 | 12.65 | 99.5 | 54.01 | 50.64 | 47.63 |
| | + OBCACHE-K | 29.45 | 44.06 | 55.29 | 58.29 | 50.38 | 32.32 | 30.57 | 25.02 | 24.14 | 66.0 | 90.35 | 44.46 | 11.15 | 99.5 | 54.22 | 51.95 | **47.95** |
| | SnapKV | 29.75 | 46.94 | 56.06 | 58.86 | 51.5 | 32.6 | 30.47 | 24.42 | 23.97 | 64.0 | 89.36 | 43.14 | 10.13 | 99.5 | 53.31 | 50.98 | 47.81 |
| | + OBCACHE-K | 29.72 | 45.84 | 55.48 | 59.2 | 50.45 | 31.84 | 30.47 | 24.73 | 24.39 | 65.0 | 89.49 | 44.19 | 12.63 | 99.5 | 52.97 | 50.84 | **47.92** |
| | AdaKV | 30.06 | 46.29 | 56.32 | 58.74 | 51.71 | 32.25 | 30.44 | 24.75 | 24.14 | 66.0 | 89.86 | 44.22 | 11.05 | 100.0 | 52.11 | 51.02 | 48.06 |
| | + OBCACHE-K | 29.67 | 46.51 | 56.27 | 58.6 | 52.86 | 32.39 | 31.36 | 25.23 | 24.77 | 70.5 | 89.92 | 44.18 | 11.13 | 100.0 | 52.07 | 51.13 | **48.54** |
| 40% KV (q-Aware) | H2O | 30.7 | 43.68 | 53.4 | 56.76 | 50.7 | 29.31 | 31.26 | 24.64 | 25.02 | 59.5 | 89.92 | 44.1 | 11.15 | 99.5 | 53.14 | 49.46 | 47.02 |
| | + OBCACHE-K | 30.94 | 45.21 | 55.08 | 58.21 | 50.94 | 32.47 | 31.47 | 24.68 | 25.45 | 62.0 | 89.92 | 43.94 | 11.65 | 99.5 | 53.43 | 50.41 | **47.83** |
| | TOVA | 29.9 | 45.38 | 55.23 | 57.8 | 51.18 | 30.99 | 31.32 | 25.12 | 24.94 | 68.0 | 90.21 | 44.24 | 11.2 | 99.5 | 53.86 | 50.97 | 48.13 |
| | + OBCACHE-K | 29.67 | 46.39 | 56.46 | 57.91 | 51.27 | 31.68 | 31.93 | 25.27 | 24.95 | 68.0 | 90.35 | 44.38 | 11.2 | 99.5 | 53.51 | 50.76 | **48.33** |
| | SnapKV | 30.24 | 46.11 | 55.96 | 58.83 | 50.11 | 32.86 | 31.64 | 24.77 | 25.11 | 68.0 | 89.61 | 43.62 | 10.15 | 100.0 | 53.2 | 51.41 | 48.23 |
| | + OBCACHE-K | 30.29 | 45.91 | 55.53 | 59.24 | 51.09 | 32.65 | 31.62 | 24.57 | 25.13 | 67.0 | 90.22 | 44.14 | 9.65 | 100.0 | 52.6 | 50.56 | 48.14 |
| | AdaKV | 30.35 | 45.36 | 55.35 | 58.89 | 51.8 | 32.38 | 31.53 | 24.51 | 25.27 | 69.5 | 90.07 | 44.17 | 10.15 | 100.0 | 52.03 | 50.67 | 48.25 |
| | + OBCACHE-K | 29.93 | 46.79 | 54.44 | 58.89 | 51.79 | 31.48 | 32.55 | 24.99 | 25.62 | 70.0 | 89.7 | 44.4 | 11.15 | 100.0 | 51.26 | 50.08 | **48.32** |
| 10% KV (q-Agnostic) | H2O | 25.82 | 15.29 | 26.14 | 41.79 | 24.05 | 18.02 | 25.38 | 19.85 | 20.9 | 44.0 | 90.93 | 43.38 | 9.5 | 50.0 | 52.27 | 47.91 | 34.7 |
| | + OBCACHE-K | 26.41 | 16.47 | 27.74 | 45.34 | 27.58 | 19.92 | 26.07 | 20.01 | 21.33 | 43.5 | 89.92 | 43.38 | 9.6 | 67.0 | 50.9 | 48.25 | **36.46** |
| | TOVA | 19.27 | 12.95 | 23.58 | 42.47 | 23.84 | 16.52 | 25.36 | 19.35 | 20.31 | 28.5 | 90.15 | 43.29 | 10.6 | 79.0 | 51.55 | 47.19 | 34.63 |
| | + OBCACHE-K | 21.5 | 14.39 | 23.33 | 43.08 | 21.65 | 17.22 | 25.87 | 19.65 | 21.37 | 35.5 | 90.21 | 43.1 | 10.1 | 81.0 | 51.59 | 48.16 | **35.48** |
| | SnapKV | 24.24 | 20.49 | 23.43 | 43.19 | 23.36 | 19.85 | 25.3 | 19.8 | 19.96 | 41.5 | 89.57 | 42.79 | 7.0 | 54.5 | 52.97 | 48.64 | 34.79 |
| | + OBCACHE-K | 26.32 | 21.04 | 24.77 | 46.44 | 26.35 | 20.28 | 25.5 | 20.65 | 20.52 | 45.0 | 89.25 | 43.31 | 7.0 | 55.0 | 53.11 | 47.78 | **35.77** |
| | AdaKV | 25.18 | 21.6 | 26.73 | 47.05 | 26.01 | 19.35 | 25.48 | 20.6 | 20.54 | 49.0 | 89.73 | 43.0 | 6.0 | 55.5 | 53.98 | 50.43 | 36.26 |
| | + OBCACHE-K | 27.76 | 24.39 | 29.29 | 51.45 | 25.7 | 19.84 | 25.89 | 21.05 | 20.65 | 49.5 | 89.43 | 43.83 | 9.0 | 76.5 | 55.82 | 50.89 | **38.81** |
| 20% KV (q-Agnostic) | H2O | 28.66 | 23.39 | 35.57 | 49.63 | 36.15 | 21.25 | 28.05 | 21.2 | 22.48 | 48.5 | 90.52 | 44.21 | 10.0 | 84.0 | 52.38 | 48.83 | 40.3 |
| | + OBCACHE-K | 30.19 | 27.17 | 37.11 | 52.87 | 38.88 | 24.74 | 28.21 | 21.63 | 23.68 | 49.5 | 90.11 | 45.01 | 10.6 | 92.0 | 51.97 | 48.88 | **42.03** |
| | TOVA | 23.04 | 20.9 | 31.3 | 52.87 | 34.91 | 21.56 | 28.51 | 20.93 | 22.27 | 41.0 | 90.17 | 43.88 | 11.05 | 97.5 | 53.31 | 49.21 | 40.15 |
| | + OBCACHE-K | 22.89 | 20.58 | 31.15 | 50.96 | 34.74 | 23.89 | 28.89 | 21.33 | 22.93 | 41.5 | 89.33 | 44.06 | 12.05 | 95.5 | 52.95 | 49.09 | 40.12 |
| | SnapKV | 27.89 | 25.31 | 30.21 | 53.37 | 36.29 | 25.96 | 27.7 | 21.54 | 22.62 | 50.0 | 89.31 | 44.09 | 7.1 | 86.0 | 53.66 | 48.97 | 40.63 |
| | + OBCACHE-K | 27.51 | 28.12 | 34.03 | 52.55 | 36.38 | 26.84 | 28.25 | 21.58 | 23.04 | 50.0 | 89.99 | 44.32 | 7.6 | 89.5 | 53.55 | 49.6 | **41.43** |
| | AdaKV | 27.71 | 29.17 | 33.42 | 55.56 | 39.76 | 26.13 | 28.07 | 22.2 | 23.14 | 56.0 | 89.8 | 44.3 | 9.6 | 86.5 | 53.1 | 49.46 | 42.1 |
| | + OBCACHE-K | 30.02 | 31.72 | 34.78 | 55.68 | 42.05 | 28.24 | 28.59 | 22.5 | 23.21 | 59.0 | 90.06 | 44.53 | 9.6 | 96.0 | 52.7 | 49.99 | **43.67** |
| 30% KV (q-Agnostic) | H2O | 29.0 | 30.62 | 42.45 | 54.4 | 43.54 | 24.47 | 29.5 | 22.33 | 24.48 | 51.0 | 89.79 | 44.87 | 12.55 | 94.5 | 52.54 | 49.24 | 43.46 |
| | + OBCACHE-K | 28.69 | 33.04 | 44.03 | 56.21 | 45.22 | 29.8 | 29.73 | 22.52 | 24.78 | 53.5 | 90.24 | 44.79 | 11.05 | 99.0 | 52.37 | 50.1 | **44.69** |
| | TOVA | 24.25 | 30.95 | 39.23 | 57.13 | 42.52 | 25.78 | 30.38 | 22.14 | 23.67 | 45.5 | 89.65 | 44.23 | 13.04 | 98.0 | 54.01 | 49.98 | 43.15 |
| | + OBCACHE-K | 25.33 | 33.03 | 42.04 | 54.82 | 46.37 | 28.14 | 30.19 | 22.37 | 23.96 | 48.0 | 89.79 | 44.66 | 10.54 | 98.0 | 54.22 | 50.3 | **43.86** |
| | SnapKV | 26.99 | 31.44 | 39.45 | 55.48 | 46.52 | 27.25 | 29.75 | 23.04 | 23.79 | 56.5 | 90.49 | 44.0 | 9.05 | 97.5 | 53.31 | 49.23 | 43.67 |
| | + OBCACHE-K | 29.62 | 33.73 | 41.9 | 55.23 | 45.95 | 30.92 | 30.06 | 23.0 | 24.24 | 55.5 | 89.97 | 44.5 | 8.04 | 97.5 | 52.97 | 49.01 | **44.51** |
| | AdaKV | 28.81 | 35.76 | 41.96 | 55.14 | 46.06 | 29.53 | 29.72 | 23.41 | 24.42 | 60.5 | 89.71 | 43.62 | 10.05 | 96.0 | 52.11 | 49.69 | 44.78 |
| | + OBCACHE-K | 28.72 | 39.04 | 44.57 | 56.37 | 47.13 | 29.09 | 30.62 | 23.2 | 24.59 | 66.0 | 89.79 | 44.59 | 10.05 | 100.0 | 52.07 | 50.07 | **45.99** |
| 40% KV (q-Agnostic) | H2O | 30.22 | 36.72 | 45.48 | 55.51 | 46.81 | 25.34 | 31.0 | 23.4 | 25.19 | 54.0 | 89.96 | 44.2 | 12.63 | 98.5 | 53.14 | 49.31 | 45.09 |
| | + OBCACHE-K | 28.69 | 38.83 | 46.01 | 56.01 | 48.16 | 28.7 | 31.13 | 23.48 | 25.58 | 56.0 | 89.89 | 44.8 | 12.05 | 99.5 | 53.43 | 49.82 | **45.76** |
| | TOVA | 25.95 | 38.56 | 44.73 | 56.21 | 46.15 | 30.63 | 31.24 | 22.93 | 24.85 | 50.0 | 89.95 | 44.69 | 11.64 | 99.0 | 53.86 | 50.04 | 45.03 |
| | + OBCACHE-K | 27.37 | 38.42 | 46.24 | 56.92 | 47.39 | 30.99 | 31.23 | 23.07 | 25.33 | 50.0 | 89.79 | 44.44 | 11.64 | 99.0 | 53.51 | 50.31 | **45.4** |
| | SnapKV | 28.93 | 35.77 | 42.78 | 56.92 | 47.87 | 30.6 | 30.93 | 23.29 | 24.87 | 58.5 | 89.96 | 43.94 | 10.05 | 98.5 | 53.2 | 49.87 | **45.37** |
| | + OBCACHE-K | 27.99 | 37.44 | 45.71 | 55.58 | 46.75 | 28.07 | 31.28 | 23.67 | 25.2 | 58.5 | 90.01 | 44.22 | 9.05 | 100.0 | 52.6 | 49.73 | 45.36 |
| | AdaKV | 28.37 | 39.51 | 44.22 | 56.4 | 48.88 | 31.18 | 31.16 | 23.08 | 25.4 | 62.0 | 89.79 | 44.16 | 11.55 | 97.5 | 52.03 | 49.93 | 45.95 |
| | + OBCACHE-K | 28.47 | 42.7 | 49.08 | 57.31 | 49.69 | 28.87 | 32.32 | 23.62 | 25.52 | 70.0 | 89.5 | 44.72 | 11.55 | 100.0 | 51.26 | 49.42 | **47.13** |

*Table 8.* LongBench full results for Qwen2.5-7B-Instruct.

| Setting | Methods | Single-Document QA | | | Multi-Document QA | | | Summarization | | | Few-shot Learning | | | Synthetic | | Code | | |
|---|---|---|---|---|---|---|---|---|---|---|---|---|---|---|---|---|---|---|
| | | NrtvQA | Qasper | MF-en | HotpotQA | 2WikiMQA | Musique | GovReport | QMSum | MultiNews | TREC | TriviaQA | SAMSum | PCount | PRe | Lcc | RB-P | Avg. |
| | ALL KV | 29.33 | 46.36 | 50.27 | 55.96 | 42.49 | 27.55 | 33.73 | 24.27 | 25.41 | 73.5 | 86.51 | 41.2 | 12.0 | 100.0 | 58.19 | 63.39 | 48.14 |
| 10% KV (q-Aware) | H2O | 28.52 | 33.75 | 38.55 | 48.07 | 33.37 | 19.5 | 26.63 | 23.18 | 18.66 | 39.0 | 86.92 | 40.78 | 10.5 | 99.83 | 49.32 | 56.31 | 40.81 |
| | + OBCACHE-K | 28.31 | 36.19 | 42.15 | 50.47 | 35.87 | 21.03 | 27.43 | 23.73 | 19.52 | 42.0 | 86.56 | 41.02 | 11.0 | 100.0 | 50.83 | 56.61 | **42.05** |
| | TOVA | 27.77 | 31.2 | 42.03 | 48.08 | 32.46 | 19.91 | 27.05 | 21.96 | 19.67 | 35.0 | 85.01 | 41.3 | 9.5 | 99.0 | 48.18 | 51.94 | 40.0 |
| | + OBCACHE-K | 27.16 | 32.87 | 45.62 | 49.52 | 31.77 | 21.48 | 27.8 | 22.41 | 20.4 | 41.75 | 85.8 | 41.46 | 12.5 | 99.5 | 50.91 | 53.32 | **41.52** |
| | SnapKV | 28.76 | 37.69 | 42.88 | 52.81 | 34.64 | 25.04 | 26.45 | 22.78 | 18.14 | 42.0 | 86.87 | 40.39 | 10.5 | 99.5 | 51.9 | 54.15 | 42.16 |
| | + OBCACHE-K | 30.53 | 39.19 | 45.16 | 54.49 | 35.51 | 24.69 | 27.11 | 23.24 | 18.28 | 43.5 | 86.24 | 40.38 | 11.0 | 99.5 | 50.91 | 55.5 | **42.83** |
| | AdaKV | 29.67 | 35.91 | 44.92 | 52.7 | 35.02 | 25.96 | 26.83 | 22.66 | 18.33 | 46.0 | 87.45 | 40.49 | 10.5 | 99.83 | 53.73 | 52.39 | **42.57** |
| | + OBCACHE-K | 29.62 | 36.01 | 44.97 | 51.09 | 35.42 | 23.91 | 27.17 | 23.23 | 18.49 | 45.5 | 87.45 | 40.12 | 11.0 | 99.5 | 50.03 | 54.5 | 42.38 |
| 20% KV (q-Aware) | H2O | 28.45 | 38.23 | 46.61 | 52.73 | 36.58 | 25.24 | 29.32 | 23.79 | 21.44 | 44.0 | 86.34 | 41.02 | 10.5 | 99.75 | 55.39 | 60.12 | 43.72 |
| | + OBCACHE-K | 29.65 | 41.95 | 46.54 | 53.1 | 37.03 | 24.1 | 30.05 | 24.01 | 21.87 | 47.5 | 87.02 | 40.98 | 11.0 | 99.75 | 56.83 | 60.18 | **44.47** |
| | TOVA | 28.09 | 37.15 | 47.12 | 52.15 | 34.53 | 22.65 | 30.04 | 22.62 | 21.76 | 47.5 | 85.26 | 41.17 | 11.5 | 99.12 | 53.95 | 57.76 | 43.27 |
| | + OBCACHE-K | 27.72 | 39.62 | 47.17 | 52.34 | 35.98 | 24.36 | 29.94 | 22.63 | 21.9 | 55.5 | 85.64 | 41.88 | 12.0 | 100.0 | 54.58 | 58.25 | **44.34** |
| | SnapKV | 29.69 | 41.63 | 48.15 | 55.27 | 36.08 | 24.47 | 29.54 | 24.04 | 20.59 | 53.0 | 86.84 | 40.46 | 10.0 | 100.0 | 56.89 | 58.53 | 44.7 |
| | + OBCACHE-K | 30.31 | 41.22 | 47.73 | 55.96 | 39.79 | 25.58 | 29.87 | 23.63 | 20.85 | 53.5 | 86.9 | 40.67 | 12.5 | 100.0 | 57.85 | 59.74 | **45.38** |
| | AdaKV | 29.09 | 41.82 | 46.89 | 54.88 | 38.0 | 24.86 | 29.24 | 23.27 | 20.85 | 55.5 | 86.66 | 40.64 | 11.0 | 100.0 | 58.0 | 57.42 | 44.88 |
| | + OBCACHE-K | 30.67 | 40.79 | 47.67 | 54.43 | 36.52 | 24.8 | 29.63 | 23.78 | 20.88 | 52.5 | 86.84 | 41.36 | 12.0 | 100.0 | 59.51 | 58.01 | **44.96** |
| 30% KV (q-Aware) | H2O | 28.66 | 42.07 | 47.58 | 54.94 | 38.41 | 24.87 | 30.03 | 23.67 | 22.56 | 47.0 | 86.86 | 41.22 | 13.5 | 99.08 | 58.6 | 61.81 | 45.05 |
| | + OBCACHE-K | 29.42 | 41.68 | 47.57 | 55.43 | 39.95 | 26.25 | 30.88 | 23.88 | 23.05 | 51.0 | 87.43 | 41.19 | 12.5 | 99.75 | 59.01 | 61.92 | **45.68** |
| | TOVA | 27.83 | 39.09 | 50.07 | 54.38 | 38.42 | 26.13 | 31.02 | 23.48 | 22.67 | 56.0 | 87.03 | 41.97 | 11.0 | 100.0 | 56.83 | 59.29 | 45.33 |
| | + OBCACHE-K | 27.97 | 40.89 | 47.97 | 54.25 | 38.03 | 25.47 | 31.35 | 23.35 | 22.64 | 60.5 | 87.34 | 41.67 | 12.5 | 99.75 | 56.61 | 59.8 | **45.64** |
| | SnapKV | 29.46 | 43.66 | 48.46 | 55.95 | 41.09 | 26.53 | 30.75 | 23.96 | 22.27 | 56.0 | 86.17 | 40.65 | 10.5 | 100.0 | 59.72 | 61.58 | 46.05 |
| | + OBCACHE-K | 29.32 | 43.97 | 48.06 | 56.01 | 41.62 | 26.1 | 31.0 | 23.96 | 22.14 | 58.5 | 86.58 | 40.61 | 12.5 | 100.0 | 59.04 | 62.42 | **46.36** |
| | AdaKV | 29.2 | 43.44 | 48.95 | 55.5 | 40.7 | 25.42 | 30.29 | 24.09 | 22.05 | 59.0 | 86.59 | 40.69 | 12.0 | 100.0 | 59.06 | 60.65 | 46.1 |
| | + OBCACHE-K | 30.07 | 43.49 | 50.06 | 55.25 | 40.12 | 26.02 | 31.18 | 23.84 | 22.73 | 58.0 | 86.51 | 40.93 | 11.5 | 100.0 | 59.71 | 62.07 | **46.34** |
| 40% KV (q-Aware) | H2O | 29.05 | 43.08 | 49.27 | 54.31 | 40.65 | 26.55 | 31.41 | 23.77 | 23.22 | 53.0 | 86.9 | 41.25 | 12.5 | 99.75 | 59.7 | 63.33 | 46.11 |
| | + OBCACHE-K | 29.22 | 43.41 | 48.88 | 54.97 | 40.39 | 26.46 | 31.68 | 24.29 | 23.36 | 54.5 | 87.2 | 41.51 | 12.5 | 100.0 | 59.86 | 63.19 | **46.34** |
| | TOVA | 29.44 | 41.95 | 49.41 | 55.45 | 39.36 | 27.11 | 31.8 | 23.63 | 23.21 | 64.0 | 86.45 | 41.5 | 11.5 | 99.75 | 56.37 | 61.61 | 46.41 |
| | + OBCACHE-K | 28.61 | 42.27 | 49.74 | 55.3 | 41.99 | 27.23 | 32.15 | 23.69 | 23.34 | 68.5 | 87.09 | 41.47 | 12.5 | 100.0 | 57.91 | 61.67 | **47.09** |
| | SnapKV | 29.95 | 44.58 | 49.09 | 54.98 | 41.73 | 26.29 | 31.69 | 23.79 | 23.1 | 58.5 | 86.43 | 41.24 | 11.5 | 100.0 | 58.57 | 62.49 | 46.5 |
| | + OBCACHE-K | 29.36 | 44.62 | 49.13 | 55.97 | 41.44 | 26.1 | 31.78 | 24.02 | 23.4 | 62.5 | 86.67 | 40.78 | 12.5 | 100.0 | 60.31 | 61.89 | **46.9** |
| | AdaKV | 29.21 | 44.73 | 49.02 | 55.81 | 42.11 | 27.31 | 31.29 | 23.96 | 23.21 | 61.5 | 86.51 | 41.26 | 11.5 | 100.0 | 58.4 | 61.36 | 46.7 |
| | + OBCACHE-K | 29.41 | 44.41 | 49.94 | 55.25 | 42.75 | 25.84 | 32.19 | 24.02 | 23.48 | 63.0 | 86.45 | 41.81 | 10.5 | 100.0 | 58.95 | 62.21 | **46.89** |
| 10% KV (q-Agnostic) | H2O | 20.28 | 13.38 | 24.14 | 31.35 | 23.28 | 14.87 | 26.3 | 18.17 | 19.36 | 42.5 | 86.89 | 39.26 | 8.0 | 54.92 | 49.54 | 57.79 | 33.13 |
| | + OBCACHE-K | 20.23 | 15.36 | 25.13 | 33.49 | 23.17 | 15.01 | 27.14 | 19.29 | 19.76 | 43.0 | 86.0 | 39.06 | 8.0 | 73.87 | 50.76 | 58.19 | **34.84** |
| | TOVA | 17.45 | 13.51 | 22.5 | 32.52 | 18.87 | 14.42 | 27.6 | 18.3 | 18.74 | 29.0 | 86.21 | 40.03 | 10.0 | 72.92 | 48.41 | 57.72 | 33.01 |
| | + OBCACHE-K | 19.59 | 13.78 | 23.69 | 34.08 | 19.3 | 17.35 | 28.12 | 19.17 | 19.8 | 30.5 | 86.27 | 40.31 | 12.5 | 81.78 | 50.91 | 57.51 | **34.67** |
| | SnapKV | 19.43 | 14.06 | 24.55 | 36.48 | 23.98 | 17.13 | 26.35 | 18.34 | 18.69 | 44.5 | 87.71 | 39.17 | 7.5 | 44.58 | 52.23 | 56.82 | 33.22 |
| | + OBCACHE-K | 20.59 | 14.73 | 24.81 | 39.07 | 23.58 | 16.6 | 26.9 | 18.44 | 18.78 | 44.0 | 87.61 | 39.35 | 7.5 | 68.42 | 50.21 | 56.61 | **34.83** |
| | AdaKV | 20.78 | 15.22 | 25.67 | 37.27 | 22.13 | 17.0 | 26.27 | 18.09 | 19.04 | 43.5 | 87.14 | 39.67 | 7.0 | 43.33 | 54.05 | 56.34 | 33.28 |
| | + OBCACHE-K | 21.05 | 16.69 | 26.99 | 36.13 | 24.54 | 15.95 | 27.19 | 19.2 | 18.84 | 44.5 | 88.21 | 39.3 | 9.0 | 78.83 | 50.4 | 56.37 | **35.82** |
| 20% KV (q-Agnostic) | H2O | 23.83 | 20.25 | 29.15 | 40.58 | 27.65 | 17.4 | 28.96 | 19.52 | 21.64 | 46.0 | 85.59 | 39.98 | 6.0 | 91.08 | 55.41 | 59.74 | 38.3 |
| | + OBCACHE-K | 25.51 | 23.22 | 29.65 | 44.35 | 28.96 | 19.06 | 29.7 | 20.39 | 22.22 | 47.0 | 85.7 | 39.82 | 9.5 | 95.83 | 56.74 | 59.66 | **39.83** |
| | TOVA | 21.85 | 21.1 | 30.09 | 42.25 | 30.21 | 18.21 | 30.17 | 19.5 | 21.09 | 42.0 | 86.79 | 40.26 | 10.0 | 95.33 | 53.8 | 59.13 | 38.86 |
| | + OBCACHE-K | 22.32 | 22.02 | 30.46 | 44.45 | 32.2 | 19.5 | 30.29 | 19.89 | 21.24 | 42.5 | 86.53 | 40.59 | 10.0 | 96.33 | 54.9 | 59.9 | **39.57** |
| | SnapKV | 22.21 | 22.43 | 30.67 | 45.89 | 28.77 | 22.0 | 28.98 | 19.94 | 21.28 | 49.0 | 87.61 | 40.2 | 7.0 | 81.33 | 57.05 | 59.98 | 39.02 |
| | + OBCACHE-K | 23.48 | 24.08 | 31.0 | 47.07 | 29.61 | 21.08 | 29.46 | 20.11 | 21.24 | 48.0 | 87.54 | 40.42 | 8.0 | 95.83 | 57.72 | 61.07 | **40.36** |
| | AdaKV | 23.35 | 22.53 | 29.57 | 44.57 | 28.65 | 21.37 | 28.61 | 20.09 | 21.45 | 53.5 | 87.89 | 40.0 | 5.0 | 79.0 | 56.82 | 59.35 | 38.86 |
| | + OBCACHE-K | 24.71 | 26.56 | 33.11 | 47.53 | 29.6 | 19.6 | 29.38 | 20.45 | 21.23 | 50.5 | 87.09 | 40.46 | 9.0 | 99.17 | 59.49 | 60.41 | **41.14** |
| 30% KV (q-Agnostic) | H2O | 25.01 | 27.38 | 33.76 | 47.21 | 32.16 | 21.49 | 30.1 | 20.53 | 22.64 | 48.0 | 87.08 | 40.62 | 9.0 | 97.5 | 58.82 | 61.68 | 41.44 |
| | + OBCACHE-K | 25.35 | 30.05 | 36.01 | 48.88 | 34.06 | 23.8 | 31.03 | 21.19 | 23.04 | 49.0 | 86.55 | 40.78 | 9.5 | 98.92 | 58.7 | 62.08 | **42.43** |
| | TOVA | 24.1 | 29.01 | 35.55 | 47.08 | 32.85 | 19.81 | 31.3 | 20.75 | 22.08 | 47.0 | 85.92 | 41.06 | 10.5 | 99.25 | 56.98 | 60.89 | 41.51 |
| | + OBCACHE-K | 25.54 | 31.25 | 36.86 | 48.96 | 33.69 | 23.0 | 31.45 | 21.12 | 22.17 | 51.0 | 86.75 | 40.92 | 12.0 | 98.92 | 57.27 | 61.14 | **42.63** |
| | SnapKV | 24.17 | 27.85 | 35.01 | 50.63 | 34.44 | 22.02 | 30.71 | 20.98 | 22.6 | 54.0 | 87.28 | 40.67 | 7.5 | 94.5 | 59.5 | 61.69 | 42.08 |
| | + OBCACHE-K | 25.18 | 29.21 | 35.43 | 52.18 | 33.2 | 22.76 | 30.71 | 21.56 | 22.57 | 56.0 | 87.25 | 40.96 | 9.5 | 99.0 | 58.49 | 62.32 | **42.9** |
| | AdaKV | 24.82 | 27.41 | 35.5 | 50.57 | 32.93 | 21.83 | 30.22 | 21.28 | 22.64 | 57.5 | 87.5 | 40.27 | 6.5 | 93.17 | 58.69 | 60.71 | 41.97 |
| | + OBCACHE-K | 26.1 | 31.01 | 36.72 | 51.59 | 31.62 | 22.01 | 31.12 | 22.02 | 22.66 | 54.5 | 86.73 | 41.46 | 8.5 | 99.75 | 59.26 | 61.35 | **42.9** |
| 40% KV (q-Agnostic) | H2O | 26.03 | 34.43 | 37.99 | 52.64 | 34.06 | 24.9 | 31.11 | 21.45 | 23.38 | 49.5 | 87.38 | 41.31 | 10.5 | 99.67 | 59.21 | 62.32 | 43.49 |
| | + OBCACHE-K | 25.97 | 35.75 | 39.17 | 50.67 | 34.59 | 26.89 | 31.82 | 22.2 | 23.58 | 54.5 | 86.92 | 40.97 | 11.5 | 99.83 | 60.04 | 62.98 | **44.21** |
| | TOVA | 24.17 | 34.92 | 39.63 | 51.48 | 34.64 | 24.9 | 32.14 | 21.38 | 22.76 | 52.5 | 87.16 | 41.74 | 12.5 | 98.92 | 56.71 | 61.41 | 43.56 |
| | + OBCACHE-K | 23.6 | 35.32 | 39.11 | 51.18 | 36.49 | 24.17 | 32.21 | 21.36 | 23.19 | 55.0 | 87.25 | 41.35 | 11.5 | 99.08 | 57.64 | 62.39 | **43.8** |
| | SnapKV | 23.88 | 34.33 | 38.09 | 55.21 | 36.26 | 24.88 | 31.55 | 22.17 | 23.2 | 59.0 | 87.54 | 40.65 | 8.0 | 97.67 | 58.45 | 62.84 | 43.98 |
| | + OBCACHE-K | 24.95 | 36.31 | 39.28 | 53.27 | 36.49 | 26.65 | 31.53 | 22.17 | 23.34 | 56.5 | 87.7 | 40.58 | 8.5 | 99.58 | 59.31 | 61.92 | **44.26** |
| | AdaKV | 25.24 | 32.87 | 37.13 | 54.02 | 34.47 | 24.07 | 31.17 | 21.91 | 23.36 | 62.0 | 86.66 | 40.36 | 10.0 | 98.17 | 58.33 | 61.62 | 43.84 |
| | + OBCACHE-K | 26.78 | 35.89 | 40.77 | 53.19 | 37.81 | 25.75 | 32.14 | 22.48 | 23.62 | 59.5 | 86.99 | 41.0 | 9.0 | 99.75 | 58.96 | 62.66 | **44.77** |

