# OpenReview forum: "OBCache: Optimal Brain KV Cache Pruning for Efficient Long-Context LLM Inference"
_ICML.cc/2026/Conference — ICML 2026 regular_

### Official Review · Reviewer_Wf94 · 2026-03-01

**Soundness:** 3
**Presentation:** 3
**Significance:** 2
**Originality:** 3
**Overall Recommendation:** 4
**Confidence:** 4

**Summary:**

The paper addresses the memory bottleneck in LLM inference caused by the linear growth of the KV cache in long-context scenarios. The authors propose OBCache, a theoretical framework that formulates KV cache eviction as a layer-wise structured pruning problem. Drawing upon the Optimal Brain Damage (OBD) theory, the authors derive closed-form saliency scores that estimate the impact of pruning specific KV pairs on the model's attention outputs. Unlike existing heuristic methods (like H2O, TOVA, or SnapKV) that rely primarily on accumulated attention scores, OBCache incorporates "output-aware" signals, accounting for the influence of Value states, Key states, and their interactions. The authors demonstrate that existing attention-based methods are special cases of their framework under simplified assumptions. Extensive experiments on LLaMA-3.1 and Qwen-2.5 show that integrating OBCache scores into existing eviction policies consistently improves performance compared to heuristic baselines.

**Compliance With Llm Reviewing Policy:**

Affirmed.

**Final Justification:**

Having read the author's rebuttal carefully, I will keep my positive score.

**Key Questions For Authors:**

1-3 see weaknesses.
4. Perturbation Window Sensitivity: The definition of the perturbation window is crucial for the "proxy" error. How sensitive is the performance to the size of this window?
5. Grouped Query Attention (GQA): You mention using GQA models (LLaMA-3.1/Qwen-2.5). When computing saliency for a KV pair shared by multiple query heads, do you sum the saliency scores across all associated query heads? Equation 35 in the Appendix suggests this, but does this lead to a "smoothing" effect that might dilute the importance of a key critical to only one specific head?

**Limitations:**

The authors have adequately discussed the limitations. They explicitly mention the "structural bias towards earlier tokens" inherent in accumulation-based methods and the limitation of the diagonal Hessian assumption (Appendix A). They also acknowledge that the method relies on a fixed-size recent window to mitigate bias, which is a heuristic. No negative societal impacts specific to this compression technique were identified, which is reasonable.

**Strengths And Weaknesses:**

# Strengths:

The paper is technically sound. The derivation of the saliency scores via second-order Taylor expansion is rigorous and logically follows from the established OBD literature. And the authors show that prior attention-based heuristic scores arise as special cases of their more general formulation.

# Weaknesses:
- Applicability to Real-World Inference Scenarios: The proposed eviction strategy appears ill-suited for realistic inference settings, particularly those involving prefix caching or multi-turn dialogue [1-2]. Query-aware eviction methods tend to be myopic; they optimize for the current query but risk discarding tokens that, while currently deemed insignificant, may be crucial for future turns or shared prefixes. This could lead to irreversible information loss and degrade performance in continuous interaction scenarios.
- Limited Experimental Scope: The evaluation relies on a restricted set of baselines and model architectures. The limited variety of models and comparison methods [3] makes it difficult to fully substantiate the generalizability and effectiveness of the proposed approach across different LLM families or broader benchmarks.
- Computational Overhead and Compatibility: The authors do not provide a comprehensive comparison of the computational overhead introduced by their method against other techniques. Furthermore, it remains unclear how this method can be efficiently integrated with standard optimization kernels like FlashAttention. Since FlashAttention avoids materializing the full attention matrix or scores to save memory and I/O, calculating the required gradients or saliency scores would likely require costly re-computation or specialized kernels, which is not adequately addressed.


[1] KVzip: Query-Agnostic KV Cache Compression with Context Reconstruction. NIPS’25
[2] SCBench: A KV Cache-Centric Analysis of Long-Context Methods. ICLR’25
[3] ProtoKV: Long-context Knowledges Are Already Well-Organized Before Your Query. ICLR’26


Minor comments for presentation:
- I recommend replacing $\delta V$ with $\delta_V$, as $\delta V$ could be mistaken for a scalar $\delta$ multiplied by $V$.
- I recommend replacing $\hat{V}|_{\mathbf{e}_p^\top[\hat{V}\ \hat{K}]=\mathbf{0}}$ with $(I-I_p)V$, where the $p$th row and $p$th column of $I_p$ is 1 and all others are 0. The former notation only indicates that the $p$th row of $\hat{V}$ is $\mathbf{0}$; it does not imply that any other rows of $\hat{V}$ equal $V$.
- The left side of Equations (14) and (15) is missing a plus sign.
- Equations (17) and (18), $\mathbf{a}_i$ -> $\hat{\mathbf{a}}_i$.

---

> ### Author Rebuttal · Authors · 2026-03-31
>
> Thank you for providing constructive feedback. We address your concerns below:
>
> > **[W1] Applicability to real-world query-agnostic scenarios.**
>
> We would like to clarify that our main contribution is a scoring framework for improving attention-only scores in KV cache eviction, rather than a specific deployment strategy. OBCache is therefore orthogonal to whether eviction is performed in a query-aware or query-agnostic setting, and can in principle be integrated into both.
>
> In the paper, we evaluate OBCache within existing eviction pipelines (e.g., SnapKV), which are mainly studied in the query-aware setting. Importantly, these methods can also be applied in a query-agnostic manner (as in AdaKV [1]), where the KV cache is compressed without access to the actual query. This setting is more challenging and often yields lower absolute performance, but it still provides a meaningful test of the robustness of eviction strategies.
>
> To show applicability, we add query-agnostic experiments (see W2), where OBCache remains effective under this harder regime. Methods such as KVZip are specifically designed for query-agnostic scenarios and can achieve higher performance there; our work is complementary, since OBCache is a modular scoring mechanism that could potentially be integrated into such frameworks.
>
> [1] AdaKV: Optimizing KV cache eviction by adaptive budget allocation for efficient LLM inference
>
> > **[W2] Limited experimental scope.**
>
> To strengthen the evaluation scope, we integrate OBCache into AdaKV, a SOTA adaptive budget allocation baseline stronger than SnapKV. We also extend evaluation from the NIAH subset to the full RULER benchmark, and additionally consider the query-agnostic setting.
>
> We report results at 10%~40% cache budgets under both query-aware and query-agnostic settings on RULER and LongBench. Due to space constraints, results are provided at the **[anonymized link](https://anonymous.4open.science/r/fig-9003/perfAdaKVSnapKV.png)**. We report OBCache-K since it consistently outperforms OBCache-V and mostly matches OBCache-VK while requiring lower complexity.
>
> Across RULER-4K/32K and LongBench, under both query-aware and query-agnostic settings, OBCache-K consistently improves over SnapKV and AdaKV. Notably, when integrated with AdaKV, it yields substantial gains (e.g., +15% accuracy on RULER-4K query-agnostic at 30% budget and +3.1 average performance on LongBench query-agnostic at 10% budget). These results strengthen the evidence that OBCache generalizes to stronger baselines and broader benchmarks.
>
> > **[W3] Computational overhead and compatibility.**
>
> The incompatibility with FlashAttention is shared by existing score-based methods (e.g., H2O, SnapKV) as they all require access to intermediate attention weights. A common workaround is to use FlashAttention for full prefill, then recompute attention within a recent query window when applying eviction. OBCache operates in this same setting. It does not require computing gradients at inference time: the saliency scores are derived in closed form and computed using forward-only operations.
>
> We clarify that efficiency results are already reported in Appendix D.1. OBCache-V adds only a small linear $O(d_{\text{head}})$ cost, and OBCache-K/VK incur $O(W d_{\text{head}})$ complexity with a small $W$. Empirically, this introduces negligible additional overhead during both prefill (<0.1s for 32K context) and decoding (<15 ms/token).
>
> > **[Q4] Perturbation window sensitivity.**
>
> Under our framework, the accumulation strategies of prior attention-based methods correspond to accumulating output perturbation across different query positions, which leads to the concept of a perturbation window (Sec. 4.3). Thus, this is not a new hyperparameter, but a unifying notion that connects and formalizes the accumulation windows used by prior methods.
>
> Fig. 2 provides empirical insights: using small recent windows aligns better with oracle selections, whereas using larger windows can bias the scores toward early tokens, in which case a fixed recent-retention policy helps. Model performance is not sensitive to window size when it falls within a small range. For example, SnapKV conducts an ablation study on what they call the "observation window" (perturbation window) and reports similar LongBench performance when this window is set to 16, 32, or 64.
>
> > **[Q5] Grouped query attention.**
>
> To examine possible dilution of a key critical to only one head, we compare GQA aggregation by sum vs. max on RULER-4K query-agnostic at 20% budget:
>
> ||sum|max|
> |-|-|-|
> |SnapKV+OBCache-V|32.3|32.7|
> |SnapKV+OBCache-K|41.7|41.2|
> |AdaKV+OBCache-V|50.0|49.8|
> |AdaKV+OBCache-K|51.7|52.7|
>
> The two choices perform similarly, which suggests that sum aggregation does not introduce significant negative effects in practice, and both are reasonable choices.
>
> > **Comments for presentation.**
>
> Thank you for these suggestions. We will incorporate all requested corrections in the revised manuscript.

---

> > ### Author Rebuttal · Reviewer_Wf94 · 2026-04-03
> >
> > I appreciate the authors' extensive effort during the rebuttal, which has partially resolved my concerns. However, I have two remaining concerns that were not directly addressed in the rebuttal:
> >
> > 1. I am not clear about the specific details of the query-agnostic setting in the supplementary experiments. Generally, a query-agnostic setting refers to the compression method rather than the dataset; therefore, the authors' rebuttal is confusing to me.
> >
> > 2. In my original review, I explicitly mentioned two recent papers: [2] SCBench (ICLR'25) and [3] ProtoKV (ICLR'26). The rebuttal entirely overlooked them.

---

> > > ### Author Response · Authors · 2026-04-04
> > >
> > > Thank you for your acknowledgment. We address your follow-up questions below:
> > >
> > > > **Clarification of the query-agnostic setting and its connection to real-world applicability.**
> > >
> > > We apologize for the confusion. Query-agnostic refers to a more challenging compression setting in which the KV cache of the context is compressed without access to the future query (e.g., the question associated with a document). This setting mimics real-world cache-reuse scenarios, such as prefix caching and multi-turn dialogue, where compression must be performed before future queries are known. It has been used in AdaKV [1] and KVPress [2] to evaluate the robustness of standard cache-eviction algorithms when the future query is unknown at compression time.
> > >
> > > Our main claim in the paper is that OBCache is a scoring framework that improves existing attention-based eviction pipelines, and this is already supported by consistent gains over four baselines (H2O, TOVA, SnapKV, AdaKV) in the query-aware setting. The added query-agnostic experiments in the rebuttal are intended to address your concern about real-world applicability more directly: they test whether OBCache remains useful under a harder and more practical compression setting.
> > >
> > > As a concrete example, we report LLaMA-3.1-8B-Instruct performance on NarrativeQA from LongBench in Table R.1. NarrativeQA contains 20 shared documents, each paired with 5–19 related questions. In the query-agnostic setting, the document KV cache is compressed once without seeing the questions and then reused to answer multiple associated questions. This reflects a real-world multi-request scenario, in which the retained KV states of the context must remain useful across multiple unknown questions. As shown, OBCache-K improves AdaKV at 10%, 20%, and 40% budgets, with the largest gains under stronger compression.
> > >
> > > **Table R.1:** Evaluation on NarrativeQA from LongBench (query-agnostic). The reported metric is F1 score.
> > > ||AdaKV|AdaKV+OBCache-K|
> > > |-|-|-|
> > > |Full KV|30.30|30.30|
> > > |10% KV|25.18|27.76|
> > > |20% KV|27.71|30.02|
> > > |30% KV|28.81|28.72|
> > > |40% KV|28.37|28.47|
> > >
> > > Therefore, the supplementary query-agnostic results provide evidence that OBCache remains effective beyond the standard query-aware setting. To further validate its effectiveness in other real-world inference settings, we conduct additional experiments on SCBench (see next section).
> > >
> > > [1] AdaKV: Optimizing KV cache eviction by adaptive budget allocation for efficient LLM inference
> > >
> > > [2] Expected Attention: KV cache compression by estimating attention from future queries distribution
> > >
> > > > **Discussion of SCBench and ProtoKV.**
> > >
> > > Due to space limits, we did not discuss SCBench and ProtoKV in the initial rebuttal. SCBench is a multi-request long-context benchmark designed to evaluate KV cache reuse under shared-context scenarios, and it has two modes: **multi-turn** (context reused within a single session) and **multi-request** (context reused across multiple sessions). Our supplementary query-agnostic setting corresponds directly to the multi-request mode, where the context is compressed before queries are observed and then reused across multiple requests.
> > >
> > > To further align with SCBench, we also report results under the multi-turn mode, where the context grows with the accumulating query-answer history within a session. Since SCBench is substantial in scale (931 contexts, 4853 queries, average input length 227K), we evaluate on a representative subtask (QA.En), and truncate all inputs to 120K to fit within the context window of LLaMA-3.1-8B-Instruct. The results are shown in Table R.2.
> > >
> > > **Table R.2:** Evaluation on QA.En from SCBench; MR denotes multi-request and MT denotes multi-turn. The reported metric is F1 score.
> > > ||AdaKV (MR)|AdaKV+OBCache-K (MR)|AdaKV (MT)|AdaKV+OBCache-K (MT)|
> > > |-|-|-|-|-|
> > > |Full KV|29.31|29.31|36.60|36.60|
> > > |10% KV|22.78|23.16|28.38|30.04|
> > > |20% KV|25.66|25.84|30.97|32.18|
> > > |30% KV|28.28|28.18|32.97|32.79|
> > > |40% KV|28.98|29.53|32.97|33.91|
> > >
> > > These results are consistent with Table R.1, where OBCache-K yields larger improvements over AdaKV at higher compression ratios, and further support the claim that OBCache scores remain effective in broader realistic inference settings.
> > >
> > > ProtoKV is methodologically beyond the scope of our study. First, it performs KV cache compression at the cluster level, whereas our work focuses on a plug-and-play token-level scoring framework. Second, it formulates a clustering problem to identify semantic groups based on key-vector cosine similarity, while our method targets a structured-pruning problem to estimate token-level saliency via output perturbation. These differences mean that OBCache is not directly applicable to ProtoKV. We will clarify this distinction in the revision. Additionally, an open-source implementation of ProtoKV is not currently available, making it difficult for us to reproduce it as a baseline within the rebuttal period.

---

### Official Review · Reviewer_LPnK · 2026-03-10

**Soundness:** 2
**Presentation:** 3
**Significance:** 2
**Originality:** 3
**Overall Recommendation:** 4
**Confidence:** 3

**Summary:**

This paper proposes OBCache, a principled, training-free framework for KV-cache eviction in long-context LLM inference that draws on Optimal Brain Damage (OBD). It formulates eviction as a layer-wise structured pruning problem and derives closed-form, output-aware saliency scores for evicting isolated values, isolated keys, or joint key–value pairs via a second-order Taylor approximation of attention-output perturbations. The authors show that several attention-weight heuristics (H2O, TOVA, SnapKV) arise as special cases and that replacing their scoring rules with OBCache’s scores consistently improves accuracy and perplexity across LLaMA and Qwen models on NIAH, LongBench, and PG19.

**Compliance With Llm Reviewing Policy:**

Affirmed.

**Final Justification:**

The author's rebuttal helps to address most of my core concerns. I will increase my score to 4.

**Key Questions For Authors:**

- The motivation part is not strong. There is a lack of illustrative or empirical study on the deficiencies of existing methods. Nor did the author clearly show (with persuasive data) the rationales behind the solution insight.
- It is hard to judge the novelty of the proposed solution. The key claimed contribution of this paper is to apply OBD in judging token saliency for different pruning units. However, simply applying OBD itself is not substantially novel; the authors need to clearly state why theoretical foundations matters and why it is a non-trivial task (i.e., challenges therein) to apply OBD for the very problem.
- The overall technical contribution is insufficient. Sec. 4.1, 4.3 and 4.4 are essentially not technical contributions but some peripheral descriptions.
- The evaluation is not solid. Some closely-related baselines (e.g., (Feng et al., 2025)) are not included. The models evaluated are relatively small, and it is unclear how the proposed solution would behave on larger models. Moreover, the efficiency/overhead performance is not evaluated in the main paper body; they are very significant performance aspects and shall be more formally presented.

Please make rebuttal to the above deficiencies if there is any misunderstanding.

**Limitations:**

Yes.

**Strengths And Weaknesses:**

Strengths：
+ It is a timely research problem to enhance cache eviction performance for LLMs.
+ The literature survey of existing works is thorough.
+ Casting KV eviction as a layer-wise structured pruning problem grounded in OBD makes sense. The approach is evaluated on two families of models (LLaMA-3.1-8B, Qwen-2.5-7B) and three standard long-context benchmarks (NIAH, LongBench, PG19 perplexity), covering both prefill and decoding phases.

Weaknesses:
- The motivation part is not strong.
- It is hard to judge the novelty of the proposed solution.
- The overall technical contribution is insufficient.
- The evaluation is not solid.

---

> ### Author Rebuttal · Authors · 2026-03-31
>
> Thank you for providing constructive feedback. We address your concerns below:
>
> > **[W1,Q1] Motivation and lack of illustrative evidence for deficiencies of existing methods.**
>
> Our key motivation is that existing methods use accumulated attention weights as an eviction indicator of token importance, which can be noisy and misaligned with a token's true impact on model outputs. We note that illustrative evidence for this deficiency is provided in Sec. 4.4.
>
> Specifically, we construct oracle tokens based on the true eviction error on model outputs, and evaluate how well different scoring rules recover these oracle-important tokens. Fig. 2 shows that attention output perturbation consistently achieves higher recall than accumulated attention weights. This gap directly supports our solution insight: output-aware scores are a more faithful eviction indicator than attention weights alone. This motivates deriving OBCache scores that lead to improved eviction decisions and empirical performance.
>
> To make it clearer, we will incorporate this illustration into the introduction in the revision.
>
> > **[W2-3,Q2-3] Hard to judge the novelty of the proposed method; why is applying OBD non-trivial, and what are the technical contributions of Sec. 4.1, 4.3, and 4.4?**
>
> The novelty of our work is not "simply applying OBD itself", but formulating dynamic KV cache eviction during autoregressive inference as a layer-wise structured pruning problem, deriving efficient closed-form saliency scores for different pruning units, and generalizing prior attention-based heuristics under this framework.
>
> Applying OBD is non-trivial for two main reasons. First, classical OBD targets static weight pruning, whereas cache eviction prunes dynamic cached activations during online inference. Second, eviction occurs in an autoregressive setting, where the true future error from removing a token is not observable at eviction time. These differences require a new formulation and analysis beyond a direct application of standard OBD.
>
> Specifically, Sec. 4.1 introduces a new objective (pruning-induced eviction error) as a proxy for the true eviction error. Sec. 4.2 then shows that, although exact evaluation of this objective is infeasible, a second-order Taylor approximation yields efficient analytical forms (OBCache scores). This derivation is non-trivial because the pruning variables are dynamic KV states coupled through nonlinear attention, requiring substantial new analysis of first- and second-order terms.
>
> Sec. 4.3 is also a technical contribution rather than peripheral description. It formally unifies prior attention-based heuristics as special cases under our framework by varying the objective and perturbation window. Sec. 4.4 provides empirical support for the surrogate objective and shows the superiority of the Taylor-approximated OBCache scores over attention-only scores.
>
> Overall, our novel OBD-based formulation enables principled, output-aware scores that generalize prior heuristics and yield consistent empirical gains.
>
> > **[W4,Q4] Closely related baselines, performance on larger models and presentation of efficiency results.**
>
> We add the most relevant value-aware methods, VATP (Guo et al., 2024) and CriticalKV (Feng et al., 2025). Both are related to OBCache-V: VATP uses the $L_1$-norm counterpart, while CriticalKV further requires a two-stage scoring mechanism. We also strengthen the evaluation by extending from the NIAH subset to the full RULER benchmark, and by considering both the original query-aware setting and a more challenging query-agnostic setting, where the KV cache is compressed without access to the actual query.
>
> We reproduce VATP (denoted as OBCache-VL1) and CriticalKV, and compare them with our three OBCache variants on RULER-4K when integrated with AdaKV [1], a SOTA adaptive budget allocation baseline stronger than SnapKV. Results are at the **[anonymized link](https://anonymous.4open.science/r/fig-9003/compare2VATPCriticalKV.png)**. VATP yields the smallest improvement over AdaKV, CriticalKV underperforms our OBCache variants in most cases, and OBCache-K/VK achieve the best performance–compression trade-offs. These results show that the additional key-aware scores from our broader OBD-based framework provide clear empirical benefits over value-only scoring. We further report OBCache-K results on RULER-32K and LongBench (**[link](https://anonymous.4open.science/r/fig-9003/perfAdaKVSnapKV.png)**), where OBCache-K consistently improves both SnapKV and AdaKV.
>
> To show that our method generalizes to larger models, we evaluate OBCache-K with AdaKV on LLaMA-3.1-70B-Instruct and report RULER-4K results at 20% budget below:
>
> ||q-aware|q-agnostic|
> |-|-|-|
> |Full KV|96.6|96.6|
> |AdaKV|91.3|56.6|
> |AdaKV+OBCache-K|94.0|63.4|
>
> We agree that efficiency is important, and will move the results from Appendix D.1 into the main body.
>
> [1] AdaKV: Optimizing KV cache eviction by adaptive budget allocation for efficient LLM inference

---

> > ### Author Rebuttal · Reviewer_LPnK · 2026-04-04
> >
> > Thanks for your rebuttal. It helps to address most of my core concerns. I will increase my score to 4.

---

> > > ### Author Response · Authors · 2026-04-06
> > >
> > > Thank you for your thoughtful evaluation and for acknowledging our rebuttal. We sincerely appreciate your time and constructive feedback, and are glad that our clarifications addressed your concerns.

---

### Official Review · Reviewer_z8iz · 2026-03-10

**Soundness:** 3
**Presentation:** 3
**Significance:** 3
**Originality:** 3
**Overall Recommendation:** 5
**Confidence:** 4

**Summary:**

This paper casts KV cache eviction as layer-wise structured pruning via Optimal Brain Damage (OBD) theory, deriving closed-form saliency scores (value-pruning, key-pruning, joint) through second-order Taylor expansion. Existing attention-based methods are shown to be special cases. Experiments on LLaMA and Qwen demonstrate consistent improvements when integrated into H2O, TOVA, and SnapKV.

**Compliance With Llm Reviewing Policy:**

Affirmed.

**Final Justification:**

I thank the authors for their detailed and thoughtful rebuttal. The responses have satisfactorily addressed my concerns, particularly regarding the diagonal approximation, the validity of the second-order Taylor expansion under high compression, and the justification of the proxy objective. The additional empirical evidence and clarifications strengthen my confidence in the proposed framework.

Overall, I find the paper technically solid, well-motivated, and practically useful. The unified perspective based on OBD is insightful and provides a principled way to understand and extend prior KV cache eviction methods. I maintain my positive assessment and recommendation.

**Key Questions For Authors:**

1.Can you bound the Taylor truncation error or empirically measure its degradation under high compression?
2.Have you compared diagonal vs. full Hessian scores, even at small scale?
3.Can you formalize when the historical-output proxy faithfully approximates the true future eviction error?

**Limitations:**

yes

**Strengths And Weaknesses:**

**Strengths**
- Clean unifying framework that recovers prior methods as special cases.
- Practical and lightweight—especially OBCache-V adds negligible overhead.
- Consistent empirical gains across models, benchmarks, and compression rates.

**Weaknesses**
- The diagonal assumption is inherited from OBD but never validated for KV cache. Unlike static weights, KV states are coupled through softmax normalization—pruning one key redistributes attention to all others. The resulting off-diagonal interactions may be significant.
- The framework relies on a second-order Taylor approximation (Eq. 2), but the remainder O(||δ||³) is never bounded. Under aggressive compression (e.g., 5% retention), perturbations are far from infinitesimal, and there is no guarantee the approximation holds.
- The true eviction error affects future outputs o_{s+1}, o_{s+2}, …, which are unavailable at eviction time. Using historical output perturbation as a proxy is only justified empirically (Figure 2, one task), with no theoretical bound on the gap between proxy and true objective.
- The value-pruning score S_p^{value} is acknowledged to be equivalent to the score in [1]. The additional key-pruning and joint scores yield only marginal gains over the value-only variant in many settings (Table 1, SnapKV; Figure 4), questioning whether the heavier theoretical machinery is justified.

**References**
[1] Y. Feng, J. Lv, Y. Cao, X. Xie, and S. K. Zhou, “Identify Critical KV Cache in LLM Inference from an Output Perturbation Perspective,” arXiv preprint arXiv:2502.03805, 2025.

---

> ### Author Rebuttal · Authors · 2026-03-31
>
> Thank you for providing constructive feedback. We address your concerns below:
>
> > **[W1,Q2] Diagonal assumption and the effect of attention redistribution from key pruning.**
>
> We would like to clarify that the effective diagonal structure arises from the isolated eviction operation itself. When evicting the $p$-th KV pair, only the $p$-th rows of $\hat V$ and $\hat K$ are zeroed, while all other rows remain unchanged. Accordingly, in Eq. 2, only the $p$-th rows of $\delta V$ and $\delta K$ are nonzero, so only the diagonal parts of $\mathbf{H}^{vv}$ and $\mathbf{H}^{kk}$ contribute to the quadratic form.
>
> If off-diagonal Hessian terms are allowed to contribute, this corresponds to a different setting in which rows other than the $p$-th row are also perturbed when the $p$-th KV pair is removed, i.e., the remaining KV states are jointly modified. This is closer to an OBS-style setting [1], where pruning is coupled with merging, rather than pure cache eviction. It is thus relevant to problems such as cache merging.
>
> Importantly, attention redistribution from key pruning is already captured by the diagonal part of $\mathbf{H}^{kk}$. The off-diagonal terms instead capture cross-effects from jointly perturbing multiple keys. Since Eq. 2 considers single-token eviction, these terms are not required here.
>
> [1] Second Order Derivatives for Network Pruning: Optimal Brain Surgeon
>
> > **[W2,Q1] Second-order Taylor truncation error and its empirical measurement under aggressive compression.**
>
> In the pruning literature, second-order approximations are commonly used, as higher-order terms are often negligible and difficult to derive in efficient closed form.
>
> To address your question regarding Taylor truncation error under high compression, we refer to the empirical study in Sec. 4.4. We construct oracle top-40 tokens (99% compression rate) using the true eviction error, and evaluate how well different saliency indicators recover them. Fig. 2 shows that the three OBCache scores closely match the exact pruning-induced eviction error, indicating that second-order Taylor terms remain an accurate approximation even under aggressive compression.
>
> > **[W3,Q3] Formalize the gap between historical output perturbation and true eviction error.**
>
> We analyze the value-pruning case, considering a perturbation window size of 1 and only the first output $o_{s+1}$. In this case, the pruning-induced error has the closed form in OBCache-V (Eq. 4), which we denote by
> $$||\hat E||=A_{s,p}||v_p||.$$
> Let the true eviction error on $o_{s+1}$ be
> $$E=o_{s+1}^{\text{evict}}-o_{s+1},$$
> where $o_{s+1}^{\text{evict}}=\frac{1}{1-A_{s,p}}\sum_{j\neq p}A_{s,j}v_j$, $o_{s+1}=\sum_j A_{s,j}v_j$. Let $w_p:=\sum_{j\neq p}A_{s,j}v_j$, then
> $$||E||=||\frac{A_{s,p}}{1-A_{s,p}}w_p-A_{s,p}v_p||.$$
> By the reverse triangle inequality,
> $$\bigl|||E||-||\hat E||\bigr|\le||E-\hat E||=||\frac{A_{s,p}}{1-A_{s,p}}w_p||=\frac{A_{s,p}}{1-A_{s,p}}||w_p||.$$
> Moreover,
> $$||w_p||=||\sum_{j\neq p}A_{s,j}v_j||\le\sum_{j\neq p}A_{s,j}||v_j||\le(1-A_{s,p})V_{\max},$$
> where $V_{\max}:=\max_{j=1...s}||v_j||$. Therefore,
> $$\bigl|||E||-||\hat E||\bigr|\le A_{s,p}V_{\max}.$$
>
> Thus, the gap is bounded by the attention weight of the evicted token times the maximum historical value norm. Due to space limits, we will provide analysis for the key-pruning case in the discussion phase.
>
> > **[W4] Key/joint-pruning scores yield marginal gains over the value-pruning score in many settings, questioning if the heavier theoretical machinery is justified.**
>
> To directly evaluate the benefit of key/joint-pruning scores over the value-only variant, we integrate OBCache into AdaKV [2], a SOTA adaptive budget allocation baseline. We also extend our evaluation from the NIAH subset to the full RULER benchmark, and additionally consider a query-agnostic compression setting, where the KV cache is compressed without access to the actual query.
>
> We compare all OBCache variants under both SnapKV and AdaKV, and report results on RULER-4K. The result curves are provided at the **[anonymized link](https://anonymous.4open.science/r/fig-9003/compareOBCache-V-K-VK.png)**. As shown, when integrated with SnapKV, the value-pruning variant is ineffective in many cases, particularly in the query-agnostic setting, while key-pruning scores consistently improve performance. When integrated with AdaKV, our strongest baseline, all OBCache variants achieve substantial gains, and the key-pruning variants consistently outperform the value-pruning variant across all settings (e.g., +3% accuracy at 10% budget in the query-aware setting and +4% accuracy at 30% budget in the query-agnostic setting).
>
> These results show that the additional key-aware scores derived from our broader OBD-based framework provide clear empirical benefits over value-only scoring, supporting the usefulness of the additional theoretical machinery in practice.
>
> [2] AdaKV: Optimizing KV cache eviction by adaptive budget allocation for efficient LLM inference

---

> > ### Author Rebuttal · Reviewer_z8iz · 2026-04-02
> >
> > Thank you for your reply. I will maintain my score.

---

> > > ### Author Response · Authors · 2026-04-06
> > >
> > > Thank you for your thoughtful evaluation and for acknowledging our rebuttal. We sincerely appreciate your time and constructive feedback.

---

### Official Review · Reviewer_6yM3 · 2026-03-12

**Soundness:** 3
**Presentation:** 4
**Significance:** 2
**Originality:** 2
**Overall Recommendation:** 4
**Confidence:** 2

**Summary:**

This paper examines KV cache eviction for long-context LLM inference in attention. The key contribution is an Optimal Brain Damage-inspired formulation of KV eviation as a layer-wise structued pruning problem from which the paper dervies closed form saliency scores for value pruning, key pruning and joint key-value pruning. The paper also argues that prior eviction rules and recent value aware scores can be viewed as special cases under particular objectives.

**Compliance With Llm Reviewing Policy:**

Affirmed.

**Final Justification:**

The paper leans on an established idea in OBD and adapts it for KV cache eviction and does a through empirical validation of the method and the rebuttal added in two relevant baselines which has led me to increase my score.

**Key Questions For Authors:**

This is admitedly not my area of expertise, but I still feel that there are some clear baselines that should be included here in addition to the H20, TOVA and SnapKV that is presented.

1. The most important missing experiments are direct comparisons to the closest newer baselines that the paper itself now cites but does not evaluate. In particular, it would be great to see comparisons against **Guo et al. (2024)** and **Feng et al. (2025)**, since the paper explicitly states that these prior methods are recovered as the isolated-value special case of OBCACHE. Without those comparisons, it is hard to tell whether the proposed OBD formulation and the additional scoring terms provide a meaningful empirical advantage over the closest prior output aware methods, or whether the gains mainly come from rederiving an already known value aware score in a cleaner framework.

2. It also may be useful to include a comparison against at least one strong adaptive budget allocation baseline such as **AdaKV** or **CAKE**, since those methods address another important source of performance gains (non-uniform layer/head budget allocation) that is not captured by H2O/TOVA/SnapKV.

If the authors provide these results, I will consider raising my score.

**Limitations:**

yes

**Strengths And Weaknesses:**

## Strengths

1. The paper is built around a principled and reasonably well motivated idea: it formulates KV cache eviction through an Optimal Brain Damage-style structured pruning lens and derives explicit saliency scores for value-only, key-only, and joint key-value pruning. From my read through, the math and derivations look sound.

2. The paper does a good job unifying prior heuristics within a common framework. In particular, it helps clarify how attention based and value aware eviction rules can be viewed as arising from different perturbation objectives or windows.

3. The empirical section is fairly broad. The paper studies both prefill and decode time under eviction, evaluates on multiple families, and uses several long-context benchmarks.

4. The paper also includes some practical systems analysis, including efficiency and GQA discussion which improves the practical relevance of the work.

## Weaknesses

1. The main weakness is that the comparison set still feels incomplete relative to the scope of the claims. Although the paper discusses several closely related recent methods, the experiments only compare against a limited baseline set, so it remains unclear how the approach fares against the strongest contemporary alternatives. See questions below for further details

2. The practical gains over the strongest reported baselines do not always look decisive. In some settings the improvements over SnapKV appear relatively modest, which weakens the case that the method clearly advances SOTA

---

> ### Author Rebuttal · Authors · 2026-03-31
>
> Thank you for providing constructive feedback. We appreciate your suggestions and address your concerns regarding missing baselines and performance gains with additional experiments. Specifically, we (1) integrate OBCache into AdaKV [1], an adaptive budget allocation baseline; (2) following AdaKV, extend our evaluation from the NIAH subset to the full RULER benchmark and additionally consider a query-agnostic compression setting, where the model must compress the KV cache without access to the actual query; and (3) compare against the closest value-aware baselines, VATP (Guo et al., 2024) and CriticalKV (Feng et al., 2025).
>
> [1] AdaKV: Optimizing KV cache eviction by adaptive budget allocation for efficient LLM inference
>
> > **[W1,W2,Q2] Integration with adaptive budget allocation baselines and whether modest gains over SnapKV weaken the case that OBCache advances SOTA.**
>
> Following your suggestion, we integrate OBCache scores into AdaKV, a SOTA baseline stronger than SnapKV. We report results at 10%~40% cache budgets under both query-aware and query-agnostic settings on RULER and LongBench. Due to space constraints, the results are provided at the **[anonymized link](https://anonymous.4open.science/r/fig-9003/perfAdaKVSnapKV.png)**. We report results for OBCache-K, since it consistently outperforms OBCache-V and mostly matches OBCache-VK's performance while requiring lower complexity.
>
> Across all benchmarks and both query-aware and query-agnostic settings, OBCache-K consistently improves over attention-only methods (SnapKV and AdaKV). Notably, when integrated with AdaKV, it yields substantial additional gains (e.g., +15% accuracy on RULER-4K query-agnostic at 30% budget and +3.1 average performance on LongBench query-agnostic at 10% budget). Since AdaKV is already much stronger than SnapKV, these results provide strong evidence that the proposed OBCache scores advance the state of the art.
>
> We agree that improvements over SnapKV are modest in some settings. This is likely because SnapKV applies a heuristic 1D pooling step tailored to noisy attention-only scores. Since OBCache produces output-aware scores that are already more accurate, the marginal benefit of such smoothing may be reduced. Nevertheless, OBCache consistently improves SnapKV, and the stronger gains over AdaKV indicate that the benefits are substantive.
>
> > **[W1,Q1] Comparison to VATP (Guo et al., 2024) and CriticalKV (Feng et al., 2025), and whether the OBD formulation provide a meaningful empirical advantage.**
>
> We first clarify how these methods relate to our OBCache-V score. Both VATP and CriticalKV use $A||V||_1$ as the pruning metric. Our OBD-based framework uses a smooth objective (the squared $L_2$ norm), as this is required to derive OBCache-K in closed-form; under this objective, the OBCache-V score is $||A||_2^2||V||_2^2$. However, if we restrict the framework to value pruning only and use an $L_1$ objective instead, the resulting score reduces to the same form $A||V||_1$. Thus, VATP differs from OBCache-V only in the choice of norm (and it is proposed empirically rather than derived from a theoretical framework).
>
> Additionally, CriticalKV relies on a two-stage eviction procedure: half of the cache budget is first allocated using an existing method (e.g., SnapKV), and the remaining budget is allocated using $A||V||_1$. This design relies on its theoretical assumption that the top 50% of tokens ranked by attention collectively contribute more than 50% of the total attention mass. In contrast, OBCache does not require such assumptions and can directly replace attention-only scores to improve eviction. Moreover, as our framework flexibly defines pruning units and objectives, it yields two additional key-pruning-oriented scores that these value-aware baselines do not consider.
>
> For direct comparison, we reproduce VATP (denoted as OBCache-VL1) and CriticalKV, and compare them with our three OBCache variants on RULER-4K when integrated with AdaKV. The result curves are provided at the **[anonymized link](https://anonymous.4open.science/r/fig-9003/compare2VATPCriticalKV.png)**. As shown, in the query-aware setting, OBCache-K/VK achieve the strongest performance among all methods. OBCache-V follows and outperforms both CriticalKV and VATP, especially at higher compression ratios. In the query-agnostic setting, CriticalKV is competitive at the lowest compression ratio (40% budget); however, it still underperforms OBCache-K and OBCache-VK at higher compression ratios. Overall, VATP provides the smallest improvement over AdaKV, CriticalKV underperforms our OBCache variants in most cases, and OBCache-K/VK achieve the best performance-compression trade-offs.
>
> These results demonstrate that the gains of OBCache do not come merely from rederiving a known value-aware score in a cleaner framework, but from the broader OBD-based formulation and, importantly, from the additional key-aware scoring terms that capture output sensitivity more effectively.

---

> > ### Author Rebuttal · Reviewer_6yM3 · 2026-04-04
> >
> > I thank the authors for their rebuttal, I have raised my score accordingly.

---

> > > ### Author Response · Authors · 2026-04-06
> > >
> > > Thank you for your thoughtful evaluation and for acknowledging our rebuttal. We sincerely appreciate your time and constructive feedback, and are glad that our clarifications addressed your concerns.

---

### Decision · Program_Chairs · 2026-04-30

**Decision:**

Accept (regular)

**Comment:**

This paper addresses an important and practically relevant problem of efficient KV-cache eviction for long-context LLM inference. The authors propose a principled framework grounded in Optimal Brain Damage (OBD). A central strength is the clean and unifying formulation, which places a range of prior heuristic methods under a single theoretical lens. The empirical results, strengthened during rebuttal, demonstrate consistent improvements across models, benchmarks, and settings, including more challenging regimes such as query-agnostic compression.  Overall, the paper is technically solid, well-motivated, and offers a clear conceptual contribution by reframing cache eviction as a structured pruning problem with output-aware saliency scores. While some concerns remain regarding approximation assumptions and deployment considerations, these do not detract from the overall contribution.